# Protein translation rate determines neocortical neuron fate

Ekaterina Borisova[1], Andrew G. Newman [1], Marta Couce Iglesias [2], Rike Dannenberg[1], Theres Schaub[1], Bo Qin [2], Alexandra Rusanova[1,3], Marisa Brockmann[4], Janina Koch[1], Marieatou Daniels[1], Paul Turko [5], Olaf Jahn [6,7], David R. Kaplan[8], Marta Rosário [1], Takao Iwawaki[9], Christian M. T. Spahn [10], Christian Rosenmund[4], David Meierhofer[2], Matthew L. Kraushar [2], Victor Tarabykin[1,11] ✉ & Mateusz C. Ambrozkiewicz [1,11] ✉

The mammalian neocortex comprises an enormous diversity regarding cell types, morphology, and connectivity. In this work, we discover a post-transcriptional mechanism of gene expression regulation, protein translation, as a determinant of cortical neuron identity. We find specific upregulation of protein synthesis in the progenitors of later-born neurons and show that translation rates and concomitantly protein half-lives are inherent features of cortical neuron subtypes. In a small molecule screening, we identify Ire1α as a regulator of Satb2 expression and neuronal polarity. In the developing brain, Ire1α regulates global translation rates, coordinates ribosome traffic, and the expression of eIF4A1. Furthermore, we demonstrate that the Satb2 mRNA translation requires eIF4A1 helicase activity towards its 5′-untranslated region. Altogether, we show that cortical neuron diversity is generated by mechanisms operating beyond gene transcription, with Ire1α-safeguarded proteostasis serving as an essential regulator of brain development.

The molecular origins of cellular diversity in the cerebral cortex have been the center of attention for developmental neuroscientists for decades. Pyramidal glutamatergic neurons of the neocortex are born from the progenitors outlining the ventricular zone (VZ) of the brain primordium and polarize synchronously with their migration to laminate distinct six cortical layers[1–6].

Classically, pyramidal cortical neuron subtypes are determined by combinatorial expression of transcription factors (TFs). Among other TFs, earlier-born deep layer neurons express FEZ Family Zinc Finger (Fezf) and COUP-TF-Interacting Protein 2 (CTIP2, also known as Bcl11b), whereby loss of either gene disrupts the corticospinal tract[7–11]. Special AT-Rich Sequence-Binding Protein 2 (Satb2) is expressed in

[1]Institute of Cell Biology and Neurobiology, Charité-Universitätsmedizin Berlin, corporate member of Freie Universität Berlin and Humboldt-Universität zu Berlin, Charitéplatz 1, 10117 Berlin, Germany. [2]Max Planck Institute for Molecular Genetics, Ihnestraße 63-73, 14195 Berlin, Germany. [3]Tomsk National Research Medical Center of the Russian Academy of Sciences, Research Institute of Medical Genetics, Tomsk, Russia. [4]Institute of Neurophysiology, Charité-Universitätsmedizin Berlin, 10117 Berlin, Germany. [5]Institute of Integrative Neuroanatomy, Charité-Universitätsmedizin Berlin, 10117 Berlin, Germany. [6]Neuroproteomics Group, Department of Molecular Neurobiology, Max Planck Institute for Multidisciplinary Sciences, Hermann-Rein-Str. 3, 37075 Göttingen, Germany. [7]Translational Neuroproteomics Group, Department of Psychiatry and Psychotherapy, University Medical Center Göttingen, Georg-August-University, Von-Siebold-Str. 5, 37075 Göttingen, Germany. [8]Program in Neurosciences and Mental Health, Hospital for Sick Children and Department of Molecular Genetics, University of Toronto, Toronto, Canada. [9]Medical Research Institute, Kanazawa Medical University, 1-1 Daigaku, Uchinada, Kahoku, Ishikawa 920-0293, Japan. [10]Institute of Medical Physics and Biophysics, Charité-Universitätsmedizin Berlin, 10117 Berlin, Germany. [11]These authors contributed equally: Victor Tarabykin, Mateusz C. Ambrozkiewicz. ✉e-mail: victor.tarabykin@charite.de; mateusz-cyryl.ambrozkiewicz@charite.de

neurons of all layers and is indispensable for the formation of the corpus callosum[12,13]. In the murine brain, Satb2-expressing cells are born during a protracted period of development[14], including both earlier- and later-born neurons, where Satb2 regulates different transcriptional networks[15]. The co-expression of Satb2 and other TFs, as well as developmental timing of Satb2 expression determine the cortical connectome and projection neuron fates in the mammalian brain[14–16]. The temporal progression of ventricular progenitor states requires restriction of multipotency towards generating Satb2-expressing neurons from E14.5 onwards in late cortical lineages[17–19]. Transcriptional priming in progenitors involves translational repression which restricts expression of early lineage genes in the later-born lineage[20]. We recently demonstrated such a mechanism for regulating Satb2 with its repressed mRNA in the neuronal precursors as early as the onset of cortical neuron production at E12.5 and showed translation upregulation for chromatin binding TFs, including Satb2, in a critical developmental window during murine mid-gestation[21]. Translation dynamics in cortical cell subtypes, as well as the translational requirements for key developmental events, such as the temporal succession of cortical progenitor fate, remain unclear.

To date, a plethora of translational regulators and their brain function, particularly regarding the disease context, like neurodegeneration and cancer, have been described. Among them is kinase/RNase Inositol-Requiring Enzyme 1α (Ire1α), also known as endoplasmic reticulum (ER)-to-Nucleus Signaling 1 (Ern1), the main sensor of ER homeostasis and regulator of the Unfolded Protein Response (UPR). Upon ER stress, a translational shift promotes expression of proteins vital for cell survival and restoration of the ER folding capacity[22]. Additionally, Ire1α regulates stress-independent remodeling of actin filaments by its association with filamin A[23], linked to periventricular heterotopia[24]. Another study unveiled a direct interaction of Ire1α with the translocon machinery in vitro[25]. Beyond that, homeostatic functions of Ire1α, especially in the developing neocortex, have remained quite elusive.

In this work, we reveal that protein translation regulates the generation of cortical neuron diversity. Using metabolic labeling, we visualize higher translation rates in the progenitors of later-born Satb2-expressing neurons, driving the synthesis of protein translation regulators in later-born cortical lineages. We reveal fundamentally different dynamics of protein synthesis rates during differentiation of early and late progenitor-to-neuron lineages. We find that Ire1α is essential for the development of Satb2-expressing cortical neurons and axon-dendrite polarity. Notably, we show that conditional deletion of *Ire1α* in the neocortex results in global decrease of translation rates, fewer translation initiation sites, altered ribosome kinetics and decreased expression of initiation factor 4A1 (eIF4A1). Finally, we reveal that Satb2 expression in neurons requires Ire1α-regulated eIF4A1-dependent translational control of 5′UTR of Satb2 itself. Taken together, our study defines post-transcriptional requirements of neuronal progenitors and distinct neuronal subtypes, unveiling the layers of gene expression regulation driving cortical neuron diversity.

## Results

### Transient attenuation of translation results in loss of Satb2 expression in cortical neurons

The overwhelming majority of research on the sources of cortical cell diversification has employed transcriptomic-based analyses to decode neuronal identities. The central dogma of molecular biology states that the protein, and not the transcriptome, ultimately constitutes the gene output[26]. Based on our recent discovery[21], we hypothesized that protein translation is among the molecular determinants of neuronal subtype identity.

First, we established a mixed primary culture system, in which we separately nucleofected E12.5 and E14.5 cortical cells with dsRed- and

EGFP-expressing plasmids, respectively, mixed the now differentially labeled cells, and cultured together to tightly control for the microenvironment. In this nucleofection system, the EGFP-positive late lineage is enriched for Satb2-expressing cells as early as at day-in-vitro one (DIV1) as compared to dsRed-expressing early lineage (Fig. S1a and b, Supplementary Data S1). At DIV1, a similar minor proportion of cells in both early (E12.5) and late (E14.5) progenitor-derived lineages express a mitotic marker Ki67 and Pax6, expressed in radial glia (Fig. S1c). In our mixed culture system at DIV5, dsRed-labeled early lineage enriched for deeper layer marker-expressing neurons and EGFP-expressing later lineage for Satb2 (Fig. S1d and e).

We then asked if pharmacologically attenuating translation early in development alters the type of neurons found in both lineages. We exposed our mixed cultures to either DMSO or cycloheximide (CHX), an inhibitor of protein synthesis[27], immediately after plating transiently for 24 h and maintained the cultures until DIV5 in normal medium (Fig. 1a). Remarkably, transient inhibition of translation attenuated Satb2 expression in both lineages but had discernible effect on the expression of CTIP2 (Fig. 1b).

Transient translation inhibition increased the proportion of Ki67-positive, but not of pHH3 positive cells at the end of the treatment at DIV1 (Fig. S1f and g). Nevertheless, control and CHX-exposed cells expressed neuronal NeuN at DIV5, indicating no differentiation defects (Fig. S1h). The inhibition paradigm did not alter the expression of neuronal fate markers Brn2 or Tbr1, signifying a specific requirement of protein translation for Satb2 in developing cortical neurons (Fig. S1i–k).

Additionally, EGFP-expressing neurons of the late lineage projected a Tau-1-positive axon (Fig. S1l), while most treated neurons failed to break their symmetry and expressed Tau-1 in the soma (Fig. S1l). We quantified a CHX-induced loss of axons (Fig. S1m), overall reduction of neurite branching (Fig. S1n and o) and aberrant polarization of KDEL-labeled ER (Fig. S1p and q). Altogether, transiently inhibiting protein synthesis reveals a critical translational window for development of Satb2 expression and neuronal polarization.

### The progenitors of later-born neurons exhibit higher translation rates

Next, we studied the global translation rates in cycling cortical progenitors, earlier- and later-born neurons using Fluorescent Noncanonical Amino Acid Tagging (FUNCAT)[28,29]. To label cortical cell lineages, we used ex utero electroporation (EUE), a method of in vivo DNA delivery to a spatiotemporally defined subset of cycling progenitors at the VZ of the embryonic cortex[30]. We electroporated VZ progenitors at E12.5 with dsRed- and E14.5 progenitors with EGFP-encoding vectors and cultured them in the presence of L-homopropargylglycine (HPG), an alkyne analog of L-methionine. After a Huisgen alkyne-azide cycloaddition reaction, the fluorescence intensity of azide-coupled fluorophore is proportional to incorporation of HPG into newly synthesized proteins and serves as an estimate of translation rate. To ensure cell specificity, we labeled the DIV1 cultures for progenitor-expressed Ki67[30,31] and found that E14.5 progenitors displayed higher translation rates as compared to their E12.5 predecessors (Fig. 2a and b). Additionally, we found that such significant upregulation of translation rates in E14.5 progenitors dramatically decreased during their differentiation into postmitotic daughter neurons at DIV5. The early cortical lineages displayed more constant translation rates during neuronal differentiation (Fig. 2c). The majority of E12.5 progenitor-derived DIV5 postmitotic cells co-expressed both Satb2 and CTIP2 in culture, and E14.5 progenitor-derived ones expressed Satb2 and NeuN (Fig. S2a and b).

Taken together, the translation rate is a dynamic feature of cortical progenitors and their derived progeny, and likely represents cell- and stage-specific requirements of different protein sets during development.

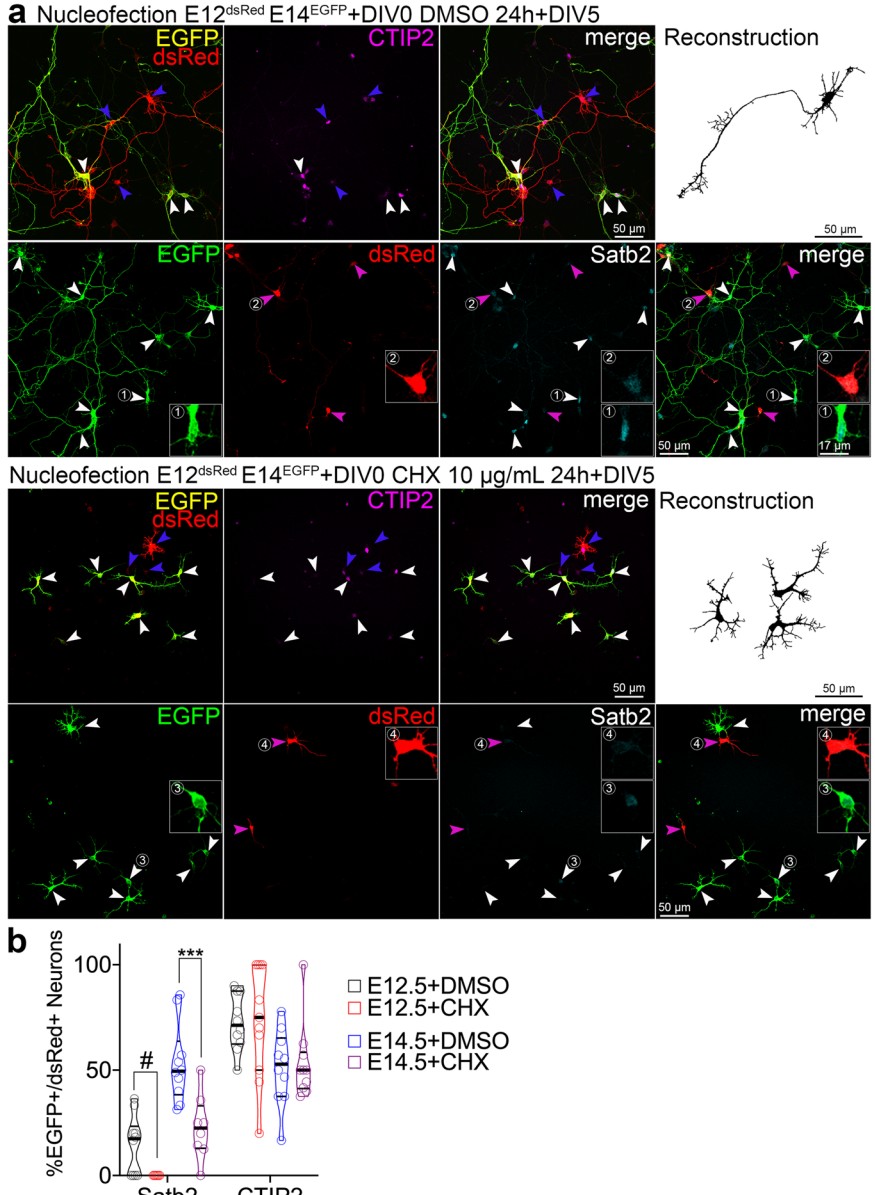

**Fig. 1 | Neuronal Satb2 expression requires a critical window of protein translation in precursor cells. a** Images of immunolabeled primary cells from E12.5 embryos nucleofected to express dsRed, and from E14.5 embryos to express EGFP, mixed and plated on a single glass coverslip. Two hours post-plating, cells were treated with vehicle (DMSO) or cycloheximide (CHX) for 20 hours, followed by medium change, and fixed at day-in-vitro 5 (DIV5). Upper panels show staining using rat anti-CTIP2, goat anti-EGFP, and rabbit anti-RFP, the latter one recognizing both EGFP and dsRed. For this reason, E12.5 cells in this case were recognized as solely expressing dsRed (blue arrowheads), but the E14.5 ones, both EGFP and dsRed (white arrowheads). Lower panels show anti-Satb2, anti-EGFP and anti-dsRed immunostaining with no cross-reacting antibodies; in this case, E12.5-derived cells express dsRed (blue arrowheads) and E14.5 ones EGFP (white arrowheads), as expected. Representative neuronal morphology is demonstrated as a semi-automated, EGFP- or dsRed-based reconstruction. (1-2) Example E14.5 (1,3) or E12.5 (2,4) cortex-derived cells immunolabeled with an antibody anti-Satb2. **b** Quantification of the cell identity markers in DIV5 neurons derived from E12.5 or E14.5 cortex. Lines on violin plot indicate median and quartiles. For statistics, Satb2, one-way ANOVA with Bonferroni's multiple comparisons test; E14.5 + DMSO vs. E14.5 + CHX, $p = 0.0007$; CTIP2, Kruskal-Wallis test with Dunn's multiple comparisons test. # indicates a comparison between fractions of Satb2-positive E12.5 DMSO- and E12.5 CHX-treated group, the latter represented by no positive cells. Data were collected from three independent cultures. *** $p < 0.001$. Refer to Supplementary Data S1 for detailed information on numerical values.

## The late cortical lineages upregulate translation of protein synthesis machinery

Because of the dynamic translation rates in the developing cortical lineages, we next investigated their translatome. To label cycling progenitors and postmitotic neurons in vivo, we took advantage of Fucci2aR reporter mouse line. To visualize cortical lineages, we crossed Cre-sensitive Fucci2aR reporter[32] to the *Emx1*[Cre/+] driver mouse line, whereby forcing cells of *Emx1*-lineage in the S/G$_2$/M phases of mitosis to express mVenus. First, we corroborated that ventricular Venus-expressing progenitors are positive for Pax6 (Fig. S2c and d). Next, puromycin labeling of the nascent polypeptides in cortical slices at E12.5 and E14.5[30,33–35] reinforced our previous findings, demonstrating uniform puromycin labeling throughout the E12.5 cortex as compared to a gradient of puromycin incorporation at E14.5, with its highest level at the VZ (Fig. 3a and b). Next, we prepared cortical primary cultures from bulk E12.5 and E15.5 cortices and pulsed puromycin in the media to label nascent proteins (Fig. 3c and d). Translation inhibition using CHX abrogated puromycin incorporation. We then

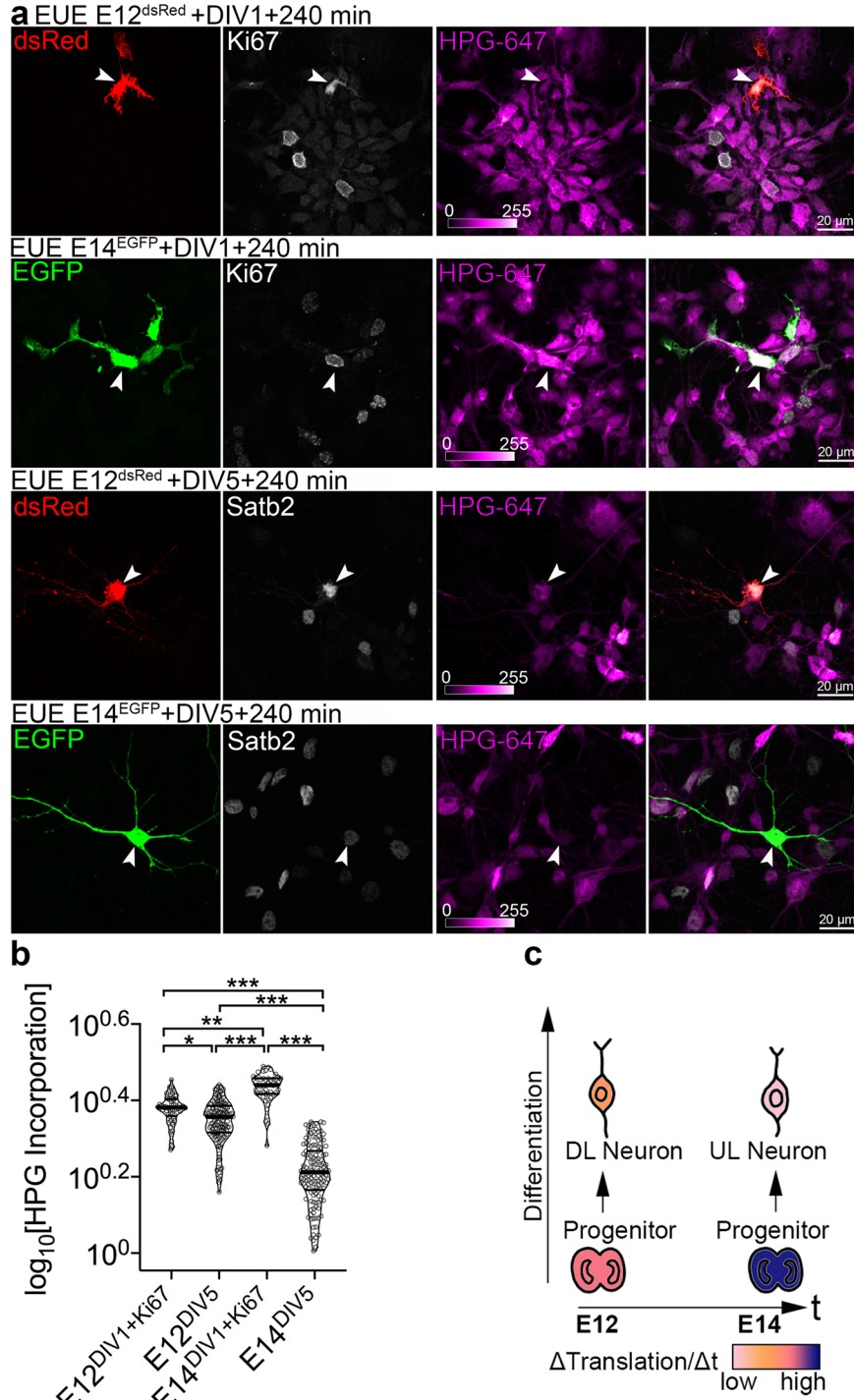

**Fig. 2 | Progenitors of later-born neurons display translation rate upregulation.**
**a** Images of primary DIV1 and DIV5 cortical cells immunolabeled for EGFP, dsRed, Ki67, and Satb2. To target Ki67-positive neuronal progenitors and their derived progeny, cortices of E12 embryos were ex utero electroporated (EUE) to express dsRed, and cortices of E14 embryos to express EGFP. Cells were triturated, mixed, plated together on a glass coverslip and pulsed with L-homopropargylglycine (HPG) for 240 min prior to fixation at DIV1 and DIV5. White arrowheads indicate representative cells. **b** Incorporation of HPG was detected using click reaction with Alexa-647 (HPG-647) and quantified as intensity of fluorescence signal normalized to the cell surface area. **c** Schematic summarizing the translation rates ($\Delta$Translation/$\Delta$t) in different types of cortical cells from E12- and E14-derived lineage. Deeper layers, DL; upper layers, UL. Violin plots on (**b**) represent per cell quantifications, thick line median and thin lines quartiles. Quantifications comprise data from three independent cultures. For statistics, D'Agostino-Pearson normality test and Kruskal-Wallis test with Dunn's multiple comparisons test. Statistical tests were two-sided. For exact p values, please refer to Supplementary Data S1. *** $p < 0.001$; $0.001 <$** $p < 0.01$; $0.01 <$* $p < 0.05$.

immunoprecipitated (IPed) puromyclated proteins and subjected the samples to mass spectrometry. We quantified the level of each protein in the IP normalized to its abundance in the input lysate (Supplementary Data S2). Notably, Gene Ontology enrichment analyses for proteins with high IP:input ratio, implicating their higher translation rates, revealed biological pathways specifically represented in early versus late lineages with glucocorticoid receptor signaling, synaptic translation and lamellipodium assembly identified in E12.5 cultures and

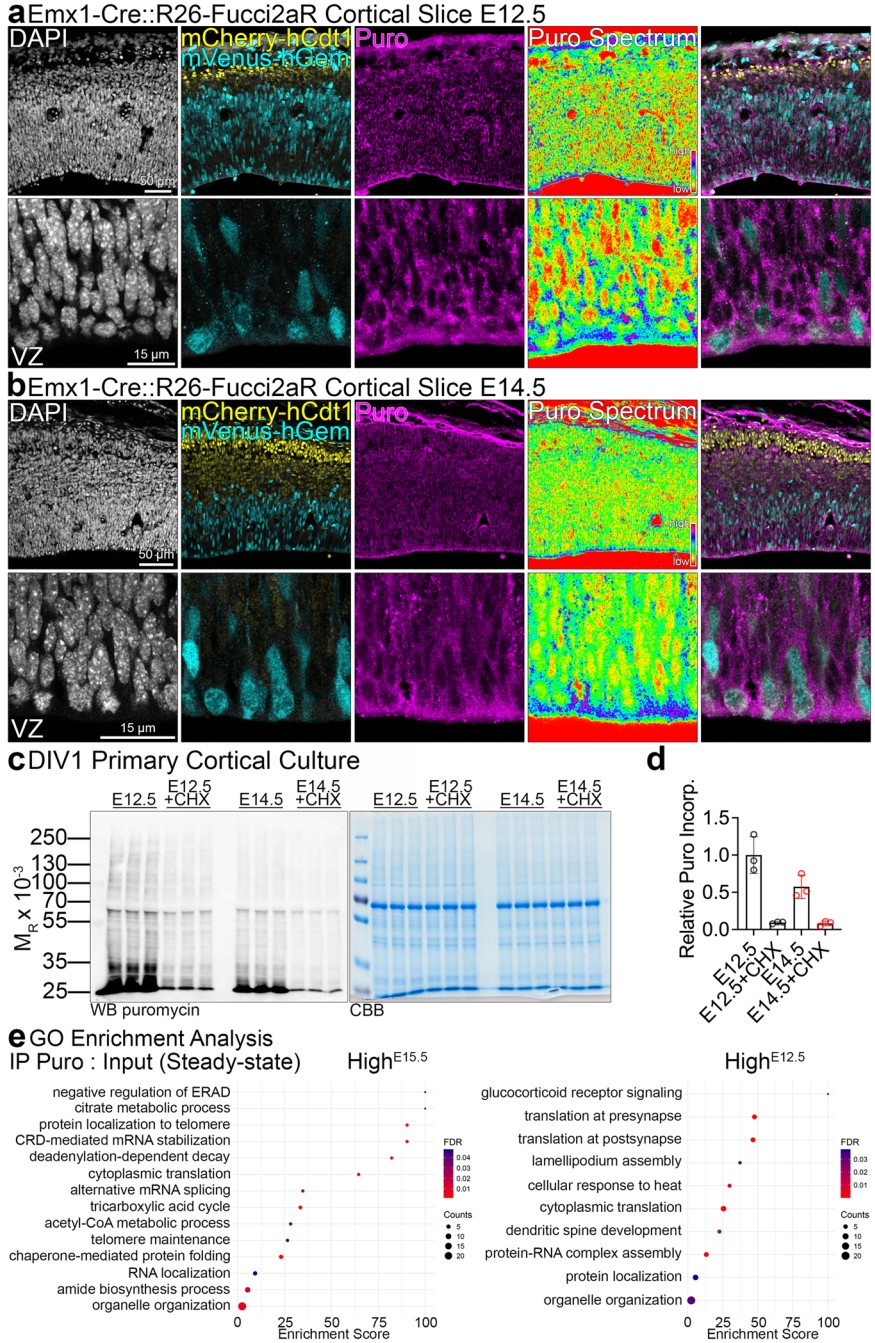

**Fig. 3 | The translatome of early and late cortical lineages. a, b** Representative images of anti-puromycin (Puro) immunolabeled slice cultures from E12.5 and E14.5 cortices. Panels labeled Puro Spectrum are intensity encoding renderings of Puro incorporation. Ventricular zone, VZ. The slice culture experiment was repeated three times with similar Puro labeling pattern. **c, d** Anti-puromycin Western blotting in E12.5 or E14.5 primary cortical cultures at DIV1, pulsed with puromycin and treated with CHX, and Puro incorporation quantification. CBB, Coomassie Brilliant Blue stain to visualize proteins in SDS-PAGE gels. **e** Summary of GO enrichment analysis based on the normalized abundance of identified proteins in the IP fraction in E15.5 and E12.5 cultures. Refer to Supplementary Data S2 for a full dataset and to Data Availability section for the information on data deposition. Bar graphs show individual data points and averages ± S.D. For statistics on (**d**), D'Agostino-Pearson normality test and Kruskal-Wallis test with Dunn's multiple comparisons test, $n = 3$ biological replicates per each condition.

ER-Associated Degradation, mRNA stability, cytoplasmic translation, and metabolic processes, including tricarboxylic acid cycle in E15.5 cultures (Fig. 3e, Supplementary Data S2).

### Translation rate is an inherent molecular feature of cortical neuron subtype

We next asked whether differences in translation rates are an intrinsic feature of postmitotic neurons of different cortical layers.

We took advantage of MetRS* mouse to study translation rates in cortical neurons in vivo. In this transgenic line, L274G mutation in methionyl-tRNA synthetase allows for a Cre-dependent cell-specific labeling of nascent polypeptide chains using a non-canonical L-methionine analog, amino acid L-azidonorleucine (ANL) and click chemistry[36]. Induction of the targeting vector is additionally marked by expression of EGFP in cells expressing Cre. To enable metabolic labeling in dorsal telencephalic progenitors and their

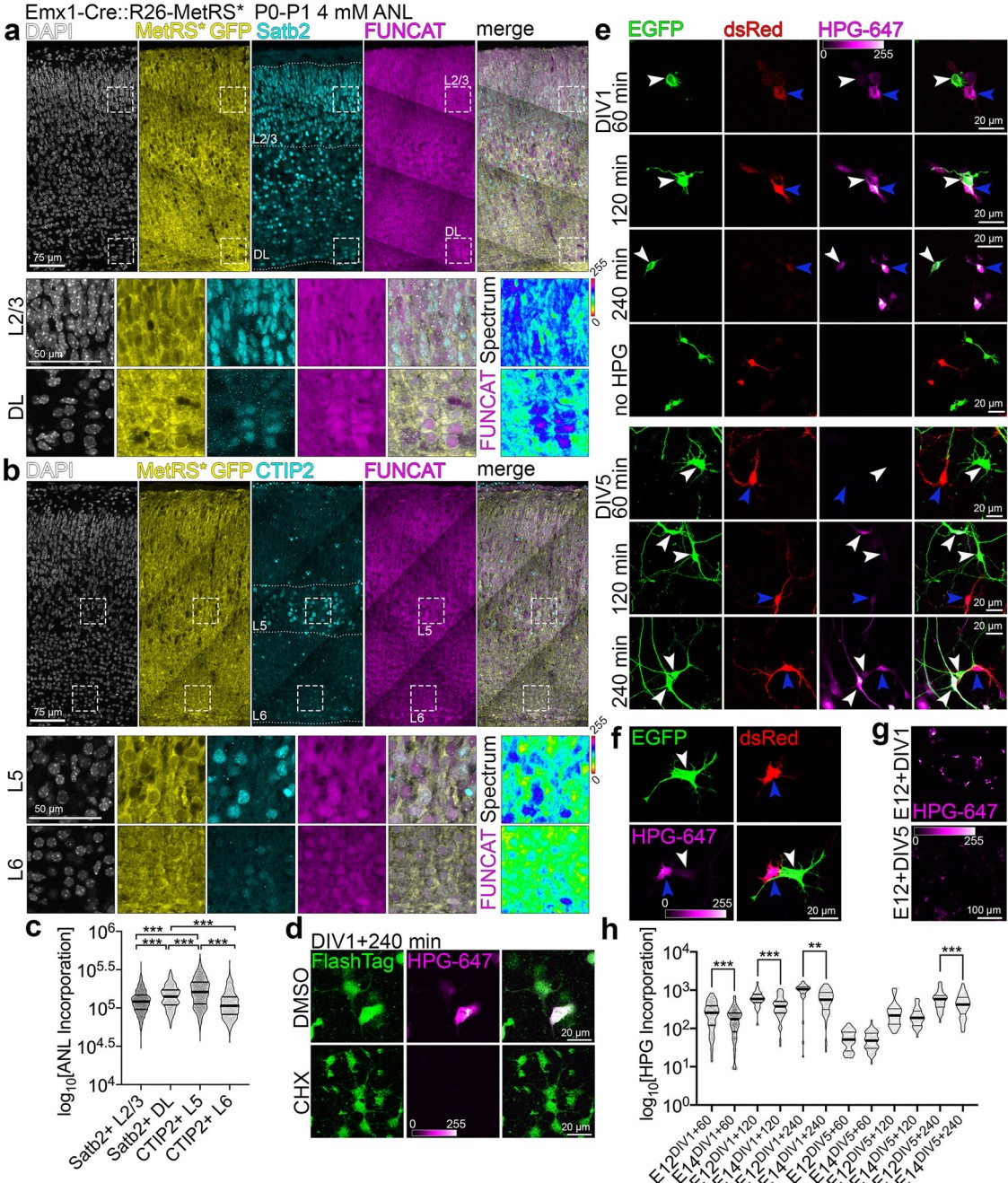

**Fig. 4 | Subtypes of cortical neurons display different protein synthesis rates.**
**a b** Representative images of coronal cortical sections immunolabeled for indicated marker proteins and after click chemistry-based conjugation of ANL and Alexa-647. Squares with dotted outlines represent respective magnified regions. Layer 2/3, L2/3; deep layers, DL. FUNCAT Spectrum represents an intensity encoding rendering of the ANL incorporation signal in each cell, quantified in (**c**). **d** DIV1 primary cortical cells from E14 cortex, fed with HPG for 240 min in presence of DMSO or 10 µg/mL cycloheximide (CHX). Note strongly reduced HPG labeling in cells cultured with CHX. FlashTag was added to the medium to label all cells. **e** Images of primary cortical cells immunolabeled for EGFP (white arrowheads) and dsRed (blue arrowheads), pulsed with HPG for indicated amount of time before fixation at DIV1 and DIV5. HPG was detected with far-red fluorophore-coupled

azide. **f** DIV1 primary cortical cells from **e** fed with HPG for 240 min. Note significantly higher HPG uptake in E12 cortex-derived immature neurons. **g** HPG labeling in primary neurons at DIV1 and DIV5 prepared from E12 cortex. Note significantly higher HPG uptake in primary cultures at DIV1. **h** Incorporation of HPG was quantified as intensity of fluorescence signal normalized to the cell surface in primary neurons. Violin plots on **c** and **h** represent data points, thick lines median and thin lines quartiles. Statistics for **c** and **h** D'Agostino-Pearson normality test; **c** one-way ANOVA with Tukey multiple comparisons; **h** two-tailed Mann-Whitney tests; for DIV1 60 min and 120 min, $p < 0.0001$; DIV1 240 min, $p = 0.002$; DIV5 60 min, $p = 0.4888$; DIV5 120 min, $p = 0.1441$; DIV5 240 min, $p = 0.0009$. For exact p values on **c** please refer to Supplementary Data S1.*** $p < 0.001$; $0.001 <** p < 0.01$; $0.01 <* p < 0.05$.

progeny, we established *Emx1*-Cre::R26-MetRS* mouse line[37] and exposed P0 cortical slices to ANL for 24 hours. Next, we immunostained cortical sections for Satb2 and CTIP2 and "clicked" the ANL to the fluorophore (Fig. 4a and b). This allowed for marker expression- and soma position-based quantification of ANL incorporation in

postmitotic neurons within layer 2/3 (L2/3) and deep layers 5 and 6 (DL, L5, L6). We found that translation rate is an inherent molecular signature of cortical neuron subtypes (Fig. 4c), with Satb2-positive DL and CTIP2-positive L5 neurons translating at higher rates in vivo.

Given that nucleofection of primary cortical cells selects for postmitotic cells as early as DIV1 (Fig. S1c), we then turned to our mixed culture system. We pulsed HPG in the methionine-free cell culture medium for 60, 120 and 240 minutes at DIV1 (for immature neurons) and at DIV5. The addition of CHX attenuated HPG incorporation, reinforcing the translation-specific nature of our findings (Fig. 4d). At DIV1 following HPG pulse, we quantified significantly higher HPG incorporation rates in neurons derived from E12.5 (dsRed +) versus E14.5 (EGFP+) cortices (Fig. 4e and f). We also detected higher HPG incorporation rates for dsRed-positive neurons at DIV5, with mild differences between the two cell populations. Neurons derived from E14.5 cortex were enriched for Satb2 expression at DIV1 (Fig. S1a and b) and DIV5, whilst E12.5 derived neurons expressed deeper layer markers (Fig. S1d and e). Secondly, we observed, that in both E12.5- and E14.5-derived neuronal cultures, HPG incorporation rates were evidently higher at DIV1 when compared to DIV5 (Fig. 4g), with neurons specified by earlier progenitors translating at higher rates in vitro (Fig. 4h).

Altogether, our findings indicate that cortical neuron subtypes in vivo and in culture can be characterized by protein synthesis efficacy.

## Ire1α is a positive regulator of Satb2 in developing cortical neurons

Acute inhibition of translation selectively diminishes Satb2 expression, and a distinct translation rate upregulation takes place in the late progenitors of Satb2-positive neurons (Figs. 1–4). We hypothesized that Satb2 expression is regulated by an upstream signaling modulating protein synthesis.

We have previously developed a method for investigating cell fate acquisition[38], which utilizes the *Satb2*[Cre/+] mouse line[39]. By expressing a Cre-inducible fluorescent reporter loxP-Stop-loxP-tdTomato in E13.5 *Satb2*[Cre/+] primary cortical cells (Fig. 5a), we used cell sorting to quantify the proportion of Satb2-expressing tdTomato-positive cells (*Satb2*[tdTom]). Co-transfection with EGFP-encoding plasmid allowed to normalize for the transfection efficiency.

Using this system, we screened a library of small molecule inhibitors for their ability to alter the proportion of *Satb2*[tdTom] neurons at DIV2[40]. APY69 was among the strongest modulators of Satb2 expression as tested with increasing compound concentration (Fig. 5b–e, Supplementary Data S3). Validation immunolabeling experiments of inhibitor-treated DIV2 cortical cells for tdTomato and CTIP2 corroborated that APY69 decreased the proportion of *Satb2*[tdTom] cells to the benefit of CTIP2-expressing neuron fraction (Fig. 5f and g).

APY69 specifically inhibits Ire1α, an ER-embedded evolutionarily ancient bimodal transmembrane kinase and RNase[41,42]. Recent evidence highlights an UPR-independent developmental function of Ire1α[23]. We hypothesized that stress-independent functions of Ire1α might be essential for expression of Satb2.

Fluorescence in situ hybridization (FISH), Western blotting, and immunolabeling revealed an evident developmental downregulation of Ire1α, expressed homogenously throughout the developing cortex, with its highest levels at E12.5, both at the level of protein, and mRNA (Fig. 5h–j). Notably, we detected robust VZ-enriched immunostaining of S724-phosphorylated Ire1α, one of the critical molecular activity marks of the enzyme[43–47]. Mice harboring S724A knock-in mutation in *Ire1α* exhibit markedly reduced Ire1α autophosphorylation and its blunted RNase activity[47].

We then went on to validate the pharmacological screening results (Fig. 5a–g) using a knock-out (KO) mouse model. To carve out the role of Ire1α in cortical development, we used the *Ire1α*[f/f] line, which enables Cre-dependent deletion of exons 20–21 from the floxed *Ire1α* allele[48]. Upon expression of Cre, the kinase-extension nuclease (KEN) domain including RNase active site is disrupted, altering its oligomerization and Ire1α enzymatic activity[43,45,49]. We conditionally inactivated *Ire1α* in the progenitors of the dorsal telencephalon by establishing a *Ire1α*[f/f]; *Emx1*[Cre/+] mouse line[37], *Ire1α* cKO. Using Western blotting from E12.5 cortical homogenates (Fig. 5k), we confirmed Cre-induced disruption of Ire1α, generation of truncated protein, overall reduction of Ire1α expression (Fig. 5l), and loss of its S724 phosphorylation (Fig. 5m), indicative of the elimination of Ire1α enzymatic activity in the cKO cortex.

To exclude the contribution of the non-*Emx1* lineage to our quantification, we prepared murine embryonic fibroblasts (MEFs) from *Ire1α*[f/f] embryos. We infected cells with control EGFP- and EGFP[p2a]Cre-encoding AAVs (Fig. S3a). At DIV5, the reduction of Ire1α expression in the Cre-infected KO cells was even more evident (Fig. S3b).

Next, we took advantage of in utero electroporation (IUE) to deliver control EGFP or KO-inducing EGFPiCre expression vector[30] to multipotent early cortical progenitors at E12.5 of *Ire1α*[f/f] embryos able to generate diverse neuronal progeny (Fig. 5n and S3c). Such strategy allows to carve out cell-autonomous role of Ire1α in the development and circumvent possible compensatory mechanisms[50]. Quantification revealed fewer Satb2-expressing neurons to the benefit of CTIP2-positive ones in the *Ire1α* KO (Fig. 5o and S3d), without alteration of their laminar positioning within the cortical plate (CP; Fig. S3e and S3f), concordant with our in vitro results (Fig. 5a–g). Silencing of endogenous Ire1α using siRNAs led to a similar neuronal fate switch tendency (Fig. S3g and S3h). Additionally, early progenitor-derived *Ire1α* KO cortical lineages comprised less Cux1-positive neurons and a tendency towards increased proportion of Sox5-expressing cells, as compared to control lineages (Fig. S3i and j).

Overexpression of human IRE1α in wild-type E12.5 cortical progenitors (Fig. 5p) resulted in an increased proportion of Satb2-expressing neurons at E18.5, and remarkably, hardly any CTIP2-positive neurons (Fig. 5q). Neither the forced expression of spliced Xbp1 (Xbp1S), a cellular stress-associated Ire1α substrate, nor S724 phosphorylation-deficient point mutant of human IRE1α altered the fate of neurons specified by E12.5 electroporated wild-type progenitors (Fig. S3k and S3l).

We also noted disrupted, highly branched morphology of *Ire1α* KO neurons, compared to majority of bipolar control cells (Fig. S3m and S3n), in both Satb2- and CTIP2- expressing cells (Fig. S3n), localizing across the entire CP (S3o–S3q).

Additionally, post-mitotic promoter NeuroD1-induced expression of Cre in *Ire1α*[f/f] at E12.5 and E14.5 did not alter the proportion of Satb2 and CTIP2 neurons four days later, indicative of a neuronal progenitor-embedded role of *Ire1α* in the acquisition of neuronal identity (Fig. S3r and S3w). We observed an analogous morphological phenotype in NeuroD1-Cre expressing *Ire1α*[f/f] neurons as in ones transfected with CAG-EGFPiCre (Fig. S3t and S3w).

## Ire1α is required for axon formation in developing cortical neurons of upper layers

In the next experiments, we sought the molecular determinants of the morphological phenotype (Fig. S3m). Given that pyramidal upper layer neurons serve as an established model to study specification of axon-dendrite polarity[51], we expressed EGFP or EGFP and iCre in neuronal progenitors of E14.5 *Ire1α*[f/f] embryos and fixed the brains at E18.5. In line with previously published results[23], loss of *Ire1α* disrupted laminar positioning of L2/3-destined cortical neurons within the CP (Fig. S4a and S4b). Similarly, we detected multiple short processes originating from *Ire1α* KO somata (Fig. S4c), as opposed to bipolar control neurons (Fig. S4d) using expansion microscopy (ExM). We noted no change of Satb2 expression in neurons derived from E14.5 control and KO progenitors (Fig. S4e), indicative of specific temporal context for *Ire1α*-regulated neuronal subtype diversity.

To study the identity of excessive processes specified by KO neurons, we immunostained the EUE-transfected control or iCre-expressing *Ire1α*[f/f] cells (Fig. S4f) for axonal and dendritic markers at DIV4 (Fig. S4g)[30]. *Ire1α* KO neurons either failed to specify an axon or

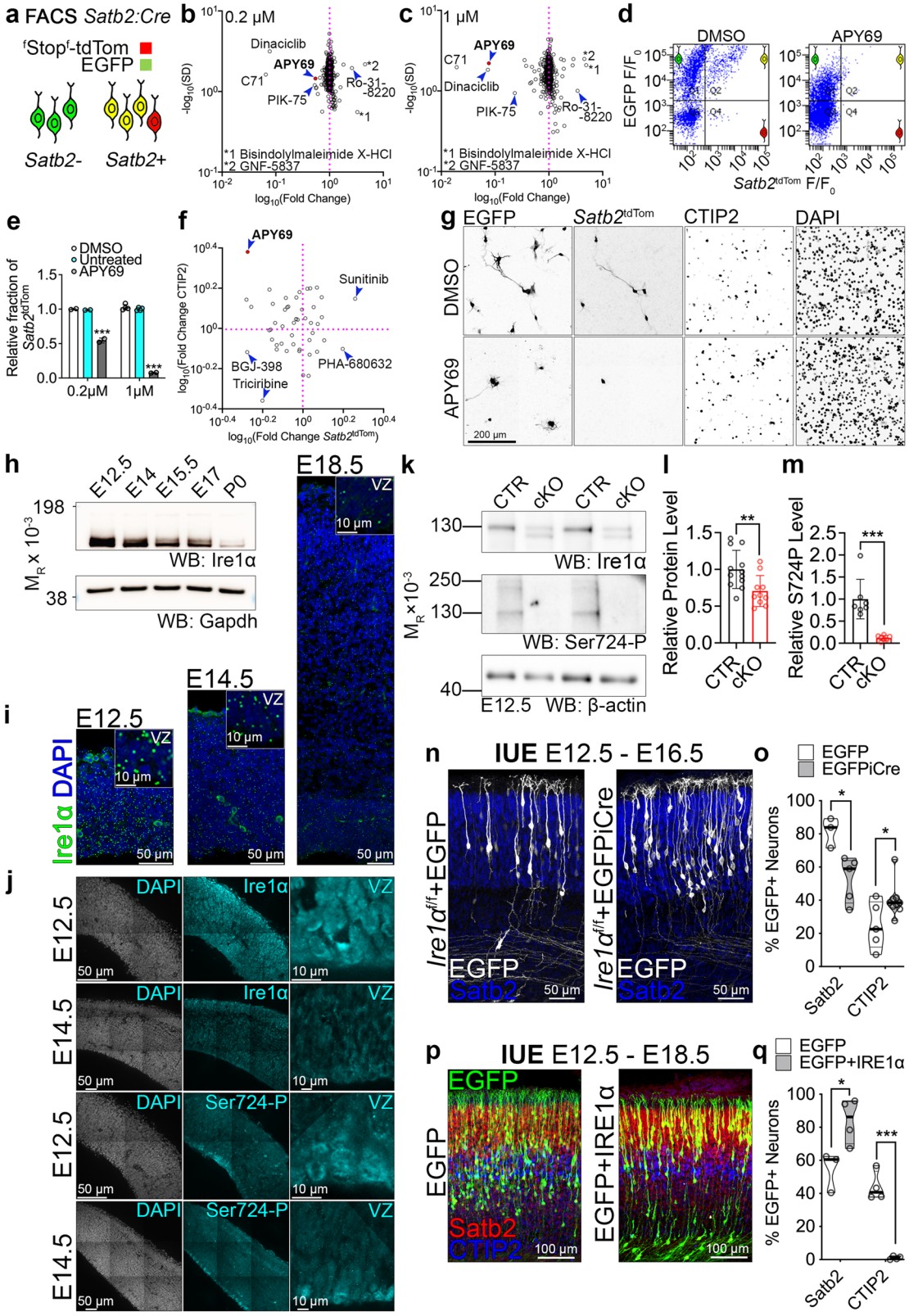

projected multiple axons, as compared to polarized control cells (Fig. S4h). Where present, KO axons were shorter (Fig. S4i) and more branched (Fig. S4j). *Ire1α* KO neurons with no axons exhibited increased dendritic branching, possibly implicating an axon-to-dendrite identity switch in single neurites. Axon-possessing KO neurons exhibited normal dendritic tree (Fig. S4k and l). Morphological defects in the KO were associated with somatic distribution of ER

tubules visualized by EGFP-Sec61β expression as compared to its polarized localization to neurites in control cells (Fig. S4m and n). This indicates a possible involvement of Ire1α in ER-driven neuronal polarization.

At DIV6, axons of control neurons were enriched for Tau-1 (Fig. S5a), Ankyrin G (Fig. S5b), and voltage-gated Na⁺ channels (Na$_v$s; Fig. S5c), in contrast to *Ire1α* KO neurons with perinuclear somatic

**Fig. 5 | Small molecule screening reveals Ire1α activity pivotal for Satb2 expression in cortical neurons. a** Screening workflow to identify signaling pathways upstream of Satb2 neuronal identity. **b, c** The results of the screening using two concentrations of each drug. Each dot represents a tested inhibitor. For full dataset, refer to Supplementary Data S3. Compound 71, C71. **d** Representative results of flow cytometry of DIV2 neurons treated with DMSO and APY69, a selective inhibitor of Ire1α. **e** Quantification of the proportion of *Satb2*-positive neurons. Because of the sheer number of the samples analyzed in the screening, we derived the statistics from two biological replicates per compound per each inhibitor dose. The screening data is further validated on Figs. 5f-5g. **f–g** Immunohistochemical screening validation. **f** DIV2 neurons were immunostained for tdTom and CTIP2 and their proportions were quantified. **g** Representative immunostaining results for DMSO- and APY69-treated cortical neurons. **h** Western blotting in cortical lysates to profile developmental expression of Ire1α. Gapdh, loading control. **i** Fluorescence in situ hybridization for Ire1α in cortical sections at indicated developmental stages. Insets demonstrate enlarged fragments of the ventricular zone (VZ). **j** Representative immunostainings in E12.5 and E14.5 wild-type cortices using indicated antibodies. **k** Representative results of Western blotting in E12.5 cortical lysates from *Ire1α*[f/f] (CTR) or *Ire1α*[f/f]; *Emx1*[Cre/+] (cKO) embryos. **l,m** Quantification of the results from **k**. **n** Representative images of immunostaining against EGFP and Satb2 in E16.5 coronal cortical sections of *Ire1α*[f/f] embryos after in utero electroporation (IUE) at E12.5 with plasmids encoding for EGFP or EGFP and Cre simultaneously. **o** Quantification of neuronal cell identity after IUE described in **n**. **p** Representative images of cortical coronal sections at E18.5 after IUE in wild-type E12.5 embryos to express EGFP, or EGFP and human IRE1α. **q** Quantification of neuronal cell identity after IUE described in **p**. For **n** and **p** compare Fig. S13. For **h** and **k** compare Fig. S14. Bar graphs show individual data points and averages ± S.D. Thick lines on violin plots represent median, thin lines represent quartiles. Statistics for **e** one-way ANOVA with Bonferroni post-hoc test; **l** Mann-Whitney test, $n_{cortices}$ for CTR = 12 and for cKO=10, $p = 0.0082$; **m** unpaired t-test with Welch's correction, $n_{cortices}$ for CTR = 7 and for cKO=7, $p = 0.0006$; **o** D'Agostino-Pearson normality test and unpaired t-test, for Satb2 counts, $n_{brains}$ for EGFP = 3 and for EGFPiCre = 4, $p = 0.0185$, and for CTIP2 counts, $n_{brains}$ for EGFP = 5 and for EGFPiCre = 9, $p = 0.0372$; **q** D'Agostino-Pearson normality test and unpaired t-test, for Satb2 counts, $n_{brains}$ for EGFP = 3 and for IRE1α = 4, $p = 0.0313$, and for CTIP2 counts, $n_{brains}$ for EGFP = 4 and for IRE1α = 4, $p < 0.0001$. Statistical tests were two-sided. *** $p < 0.001$; $0.001 <$** $p < 0.01$; $0.01 <$* $p < 0.05$.

---

localization of these axonal markers. Current clamp recordings from *Ire1α*[f/f] autaptic hippocampal neurons infected with lentiviruses encoding for EGFP or Cre showed a higher number of action potentials (APs) generated upon lower current injections (Fig. S5d and e), implicating more axon initial segments (AIS) specified by KO neurons and/or their altered molecular composition. Importantly, overall membrane integrity and conductance of *Ire1α* KO neurons were unaltered (Fig. S5f and g).

Moreover, the aberrant lamination of E14.5 progenitor-derived cortical neurons (Fig. S6a) was associated with loss of bipolar morphology (Fig. S6b–d). Regarding the molecular mechanism of polarity disruption, we corroborated previously reported altered localization and expression level of Filamin A[23] (Fig. S6e), and found loss of acetylated and tyrosinated microtubules from the soma and the longest neurite (Fig. S6f and S6g), as well as increased fraction of neurons with fragmented Golgi apparatus[30] (Fig. S6h and i), misaligned with the emerging longest neurite at DIV2[52] (Fig. S6j–S6l). Altogether, loss of bipolar morphology in developing *Ire1α* KO neurons is associated with defects in axon-dendrite polarity, increased current sensitivity, altered dynamics of microtubules, and localization of Golgi apparatus.

## Ire1α alters cell cycle dynamics of the cortical progenitors

Because of our findings in multipotent early progenitors, as well as VZ expression of active *Ire1α* (Fig. 5 and S3), we then investigated the dynamics of the cell cycle in the cortical progenitors upon *Ire1α* KO. To label proliferating progenitors, we first pulsed thymidine analog incorporating into the DNA of dividing cells bromodeoxyuridine (BrdU) at E12.5 (Fig. S7a-S7g) and E14.5 (Fig. S7h–m) and quantified the proportions of Satb2 and CTIP2-expressing neurons at P2 (Fig. S7n), when neurogenesis is complete. Among all cells born at E12.5, we counted fewer Satb2+ and more CTIP2+ cells (Fig. S7b and S7c) in the cKO, consistent with our IUE experiments using *Ire1α*[f/f] line (Fig. 5o). Moreover, in E14.5 BrdU-pulsed P2 cKO cortices, we found more BrdU+/CTIP2+ cells than in control (Fig. S7h and S7i), which localized to the top of the CP, at the expense of BrdU+/Satb2+ neurons (Fig. S7j–l).

Next, we analyzed the formation of CTIP2-positive and Tbr1-positive lineages in control and cKO cortices using BrdU pulse at E11.5, E12.5 and E14.5 (Fig. S7o–v). One day after the pulse, we detected a higher proportion of CTIP2+ neurons born at E11.5 and E14.5. Additionally, we found an increased proportion of cells which exited the cell cycle in the E13.5 cortex. To visualize the dynamics of the cell cycle upon disruption of *Ire1α*, we also introduced a Cre-sensitive Fucci2aR (F2aR) transgene[32] to the *Ire1α*[f/f]; *Emx1*[Cre/+] mouse line. This way, we quantified a higher proportion of Venus-labeled S, G2 and M phase

progenitors, as well as mCherry-expressing postmitotic neurons in the E14.5 cortex upon *Ire1α* cKO (Fig. S7s and t), in accordance with our BrdU pulse-chase experiments (Fig. S7a and u). These data altogether indicate a critical requirement of Ire1α for the correct proportion between Satb2- and CTIP2-expressing neurons in a mechanism regulating the cell cycle of cortical progenitors.

## Ire1α controls the synthesis of the ribosomal constituents and proteins involved in protein translation in the developing forebrain

During the UPR, robust cellular reprogramming is driven by Ire1α signaling network[53,54]. Ire1α activation is paralleled by suppression of general translation[55]. We hypothesized that also during development, Ire1α influences protein translation to regulate key developmental milestones. To test this, we purified actively translating ribosomes ex vivo in forebrain-specific *Ire1α* cKO[56]. Sucrose density gradients revealed higher level of polysomes in cKO cortices (Fig. 6a). Importantly, in the CP of *Ire1α* cKO, we corroborated fewer Satb2-expressing neurons to the benefit of CTIP2-positive ones (Fig. S8a and b), as well as morphological abnormalities similar to ones observed in the IUE experiments presented in *Ire1α*[f/f] line on Fig. S4a and d (Fig. S8c and d).

We then investigated the RNAs that localize to polysome fraction in *Ire1α* cKO cortex. RNA sequencing in bulk cortex of the control and cKO did not reveal gross changes in the transcriptome (Fig. S8e). Next, we performed two types of analyses in polysome-bound RNAs of control and cKO cortices. Differential Gene Expression (DGE) detected changes in the abundance of small nucleolar or spliceosomal RNA and transcripts encoding for core components of rRNA processing, among others (Fig. 6b, c, Supplementary Data S4). Given the disruption of the RNase domain in our cKO and the ability of this Ire1α domain to catalyze mRNA decay[49] and splicing[57], we also analyzed and Differential Transcript Expression (DTE) and found differences in the abundance of intron-retained transcripts coding for proteins involved in ribosome assembly, splicing and protein translation initiation and elongation, among others (Fig. 6d, Supplementary Data S4).

Polysome-associated RNAs in *Ire1α* cKO cortices encoded structural components of the ribosome, cytoplasmic translation and ribosomal subunits, revealed by the Gene Set Enrichment analysis (Fig. 6e and f, Supplementary Data S4). These results indicate that *Ire1α* loss in the cortex alters translation of RNAs crucial for ribosome complex function.

Activation of UPR, triggered by Ire1α, leads to translational shutdown in cellular efforts to restore protein homeostasis[58]. We next assayed translational repression upon Ire1α disruption in non-stressed conditions, using in situ run-off assays in primary cortical cultures

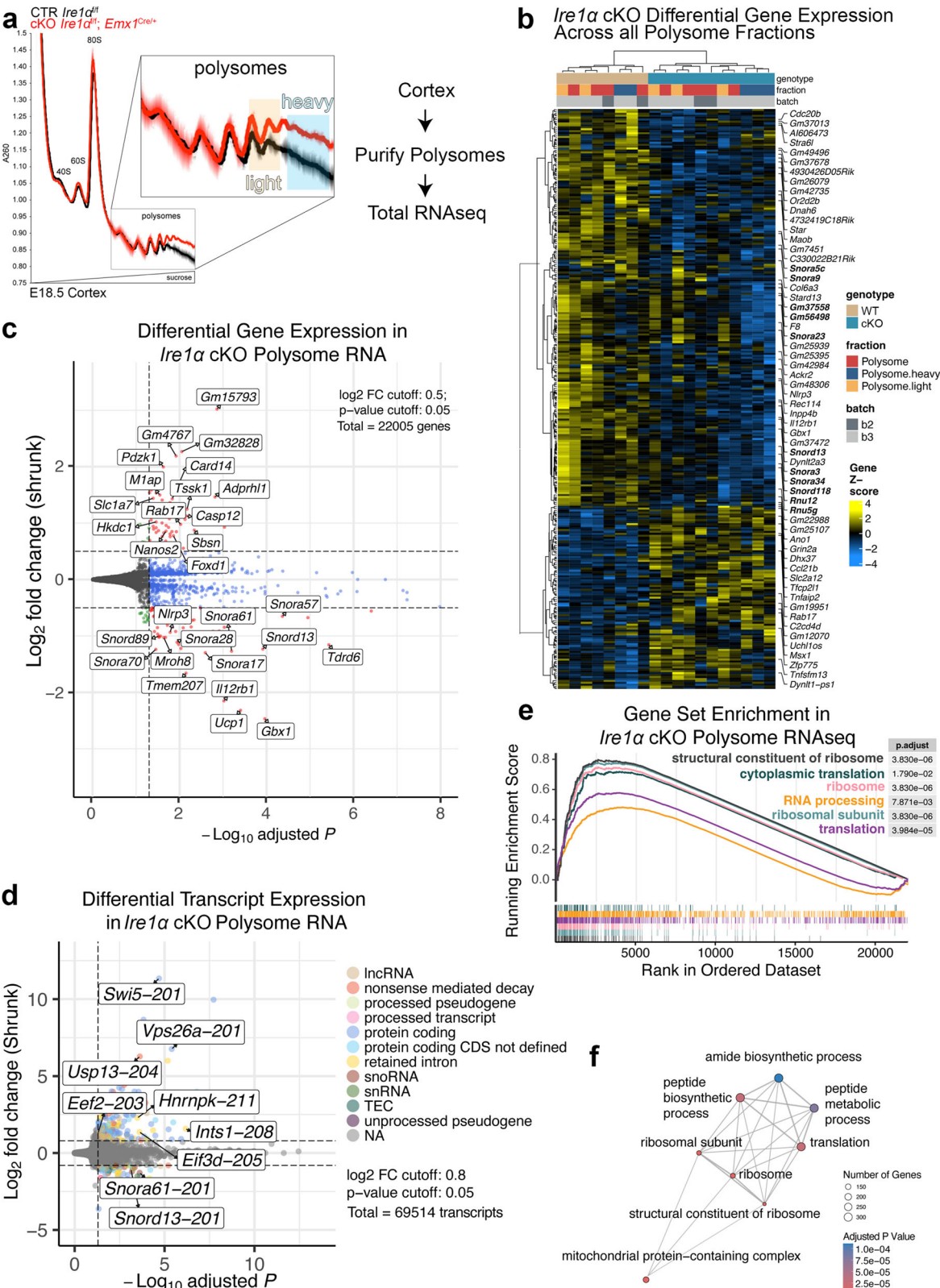

followed by puromycilation[59]. Using this assay, we detected an increase in ribosome stalling in the cKO neurons, indicating Ire1α-mediated regulation of translation in basal conditions (Fig. 7a and b).

Given robust *Ire1α*-driven regulation of polysome-bound RNAs converging on cytoplasmic translation, as well as specific translation of protein synthesis machinery (Fig. 6f), we measured the steady-state expression levels of translation initiation and elongation regulators in

control and cKO cortices (Fig. S9a and b). We identified upregulated level of eukaryotic elongation factor 2 (eEF-2), downregulated level of phosphorylated eEF-2 on Thr56, and decreased level of eukaryotic initiation factor 4A1 (eIF4A1) in the *Ire1α* cKO cortex (Fig. 7c and d).

We then overexpressed eEF-2 using IUE in E12.5 and E14.5 wild-type embryos (Fig. S9c–h). Four days later, we detected no alteration of Satb2 expression in the neuronal progeny after eEF-2 OE

**Fig. 6 | Polysome-enriched transcripts in *Ire1α* cKO encode proteins regulating constituents of the ribosome and protein translation machinery. a** Analytic density gradient fractionation of A260-normalized E18.5 neocortex lysates, measuring the relative abundance of ribosomal subunits, 80S ribosomes, and polysomes. A260 curves plotted as mean ± S.D. across replicate fractionations in one experimental batch, baseline (1.0) centered at onset of 40S peak. **b** Differential Gene Expression across all polysome fractions in *Ire1α*^f/f control (WT) and *Ire1α*^f/f; *Emx1*^Cre/+ (cKO) cortices. In bold are small nucleolar RNAs and components of spliceosome. **c** Volcano plot summarizing the Differential Gene Expression in control and cKO polysome RNAs. Indicated are hits with altered expression levels between genotypes. **d** Volcano plot summarizing Differential Transcript Expression

in control and cKO polysome RNAs. **e** Top GSEA plots for cKO polysome RNA fraction versus control polysome RNA fraction. **f** Enrichment map visualization of cellular processes represented by significantly changed transcripts within the polysome fractions between control and cKO cortices. For statistics, we used DESeq2 which employs the Wald test to test the null hypothesis that gene expression in a generalized linear model fit with a negative binomial distribution is zero, and adjusts p values using the Benjamin Hochberg (BH) procedure. Unique genes with adjusted p values < 0.05 and log2foldchange > 0.5 were determined to be differentially expressed. For detailed datasets, refer to Supplementary Data S4 and to Data Availability section for the information on data deposition.

---

(Fig. S9d and e), but a mild neuronal lamination defect, similar to *Ire1α* KO (Fig. S9f–h), linked to enhanced cell cycle exit one day after IUE (Fig. S9o and p). Next, we inactivated *eIF4A1* using CRISPR-Cas9 technology (Fig. 7e), which allows for simultaneous sgRNA and humanized Cas9 nickase delivery to the developing E12.5 progenitors[60,61]. We corroborated the efficiency of sgRNA targeting eIF4A1 (Fig. S9k and S9l). The loss of *eIF4A1* resembled *Ire1α* KO (Fig. 5) regarding the types of neurons generated, with less Satb2-expressing neurons, at the expense of CTIP2-positive cells (Fig. 7e and f). Notably, the onset of eIF4A1 downregulation in *Ire1α* cKO was detectable in E14.5 cortical bulk homogenate (Fig. 7g–j). Interestingly, simultaneous eEF-2 OE and *eIF4A1* KO resulted in the highest proportion of E14.5 progenitor derived neurons without an axon (Fig. S9i and j) reminiscent of *Ire1α* KO neurons (Fig. S4f).

Both eIF4A1 and eEF-2 were detected throughout the neurogenic stages in cortical homogenates (Fig. S9m) and highly and homogenously expressed in the E12.5 cortex, and to a lesser extent at E14.5 (Fig. S9n). Using fluorescence activated cell sorting, we purified FlashTag-positive (FT+) apical radial glia[19] and verified the expression of cap-dependent translation initiation factors eIF4A1 and eIF4A2, as well as Ire1α in E14.5 cortical progenitors (S9q–t).

Because of Ire1α-dependent sorting of intron-retaining transcripts to polysomes (Fig. 6d), as well as loss of S724 phosphorylation, implicated in regulation of mRNA stability and splicing[47,49] (Fig. 5k), we next hypothesized that Ire1α controls the proteostasis of eIF4A1, leading to its diminished levels (Fig. 7c and d). To mimic the molecular landscape after disruption of *Ire1α*, we inhibited the S724 phosphorylation using APY29[47] (Fig. 7k). CHX pulse chase experiments showed that APY29-mediated inhibition of Ire1α resulted in eIF4A1 destabilization (Fig. 7l and m).

Next, we investigated the developmental stage-specific association of Ire1α with the ribosome. Using co-immunoprecipitation in E12.5 and E14.5 cortical homogenate, we quantified that endogenous Ire1α interaction with ribosomal protein S6, (RPS6), was significantly stronger at E12.5 (Fig. 7n and o). Binding to the large ribosome subunit protein RPL7 and interactor of mRNA poly(A) tail PABPC4 was not developmentally regulated. These results suggest a specific requirement for a stronger interaction between Ire1α and RPS6 at the onset of neurogenesis.

Altogether, these findings demonstrate that ribosome-associated Ire1α controls expression levels of translation regulators and proteostasis of eIF4A1.

## Ire1α is a regulator of protein translation rates

Increased polysome level in the *Ire1α* cKO may represent ribosomes elongating more slowly or stalling, and thus accumulating in the heavy fraction (Fig. 7a and b)[62]. To further explore this hypothesis, we examined translation rates in *Ire1α* cKO using FUNCAT[28,29]. We first investigated protein synthesis rates in early E12.5 multipotent neuronal progenitors able to generate diverse lineages using EUE-mediated Cre delivery to induce *Ire1α* loss. At DIV1, we quantified lower translation rates in Cre-expressing Ki67-positive[63] *Ire1α* KO progenitors as

compared to EGFP-expressing *Ire1α*^f/f ones (Fig. 8a and b). Similarly, we detected lower translation rates in E12.5 mitotic progenitors upon loss of *eIF4A1* (Fig. 8c and d). We quantified approximately 50% lower rates of translation in DIV4 upper layer *Ire1α* cKO E14.5 progenitor-derived neurons as compared to control cells (Fig. 8e and f). To study the rate of translation in vivo, we introduced the MetRS* transgene to the *Ire1α*^f/f; *Emx1*^Cre/+ line. After exposure to ANL and click chemistry with biotin-alkyne, we demonstrated largely reduced ANL incorporation in E14.5 cKO cortices as compared to *Ire1α*^f/+; *Emx1*^Cre/+ heterozygotes (Fig. S10). Taken together, *Ire1α* loss engenders a decrease in the rate of protein synthesis in the developing cortex.

To further investigate the mechanism of such regulation, we took advantage of ribosome run-off experiments with harringtonine[64]. To circumvent confounding effects of varying translation rates in cortical cell types (Figs. 1–4), we infected *Ire1α*^f/f MEFs with control or Cre-encoding AAVs and kept in culture for DIV5 (Fig. S3a and b). We first measured protein synthesis in MEFs using puromycin[33] and found that *Ire1α* KO cells showed its reduced incorporation (Fig. 8g). *Ire1α* KO cells also showed diminished response to the harringtonine treatment, which might be attributable to slowly elongating ribosomes (Fig. 8h).

Next, we used SunTag[65] to label single mRNAs and visualize translation in control and Cre-infected fixed *Ire1α* KO MEFs (Fig. 8i). To study translation dynamics in cells, mCherry-fused PP7 bacteriophage coat protein is used to identify mRNAs and the EGFP-tagged single chain variable fragment recognizing the SunTag allows to detect the synthesis of nascent proteins. As compared to control, *Ire1α* KO MEFs demonstrated fewer colocalizing EGFP and mCherry puncta, reflective of decreased number of mRNA molecules in translation (Fig. 8j–l). Altogether, we conclude that lower translation rates upon *Ire1α* KO are associated with slower ribosome elongation rates and decreased number of active translation sites.

## Non-canonical role of Ire1α in developing cortex

Next, we tested if disruption of *Ire1α* engenders alteration of UPR components expression. Immunostaining for the molecular off-switch for cytoplasmic translation eIF2α and its Ser52 phosphorylation showed homogenous expression across E12.5 cortex and a mild enrichment in the CP neurons at E14.5 (Fig. S11a), an expression pattern different from the one of Ire1α (Fig. 5). Further, quantification of ER stress marker levels[66,67], eIF4A2 and JNK2 immunostaining revealed no gross differences between control and cKO cortices (S11b–d). Finally, using IUE in E13.5 cortices, we mimicked the UPR activation by co-expressing the phosphorylation-deficient S52D variant of eIF2α (S11e) in the cells electroporated with shRNA to knock-down endogenous eIF2α and showed a generalized defect in neuronal differentiation and diminished CP entry (Fig. S11f and g), rather than specific effects on Satb2 and CTIP2 (Fig. 5n and o). These results suggest that cortical cell diversity is regulated by specific translation pathways, including among others, a non-canonical, UPR-independent Ire1α signaling.

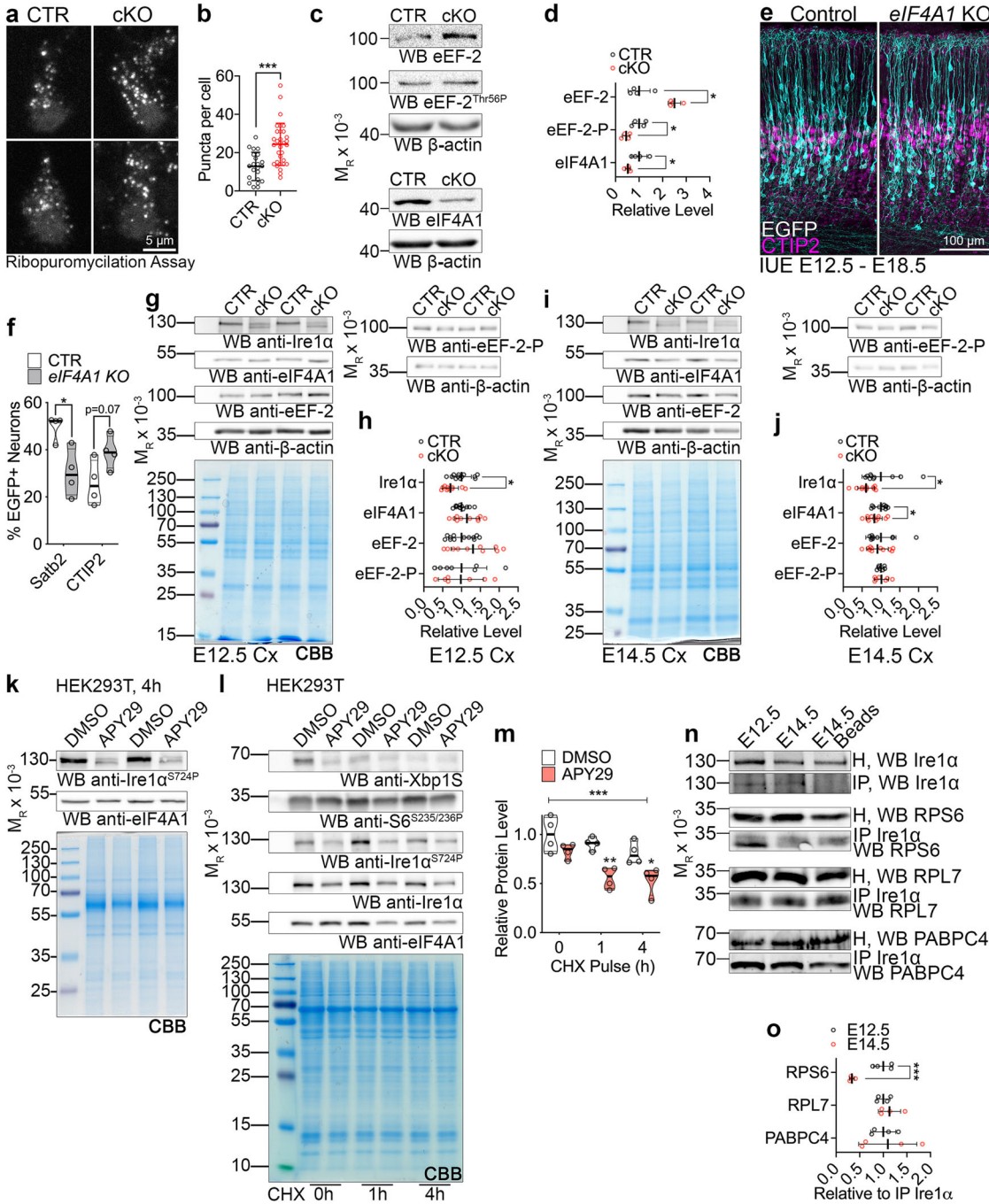

## eIF4A1 helicase activity regulates Satb2 mRNA translation during development

Given the critical requirement for high protein synthesis rates in neuronal progenitors of later lineages (Fig. 1) and the regulation of global mRNA translation by Ire1α and eIF4A1, we investigated the mechanisms of Ire1α-mediated regulation of Satb2 expression.

Translational control of eIF4A1 requires 5′UTR embedded elements, like G-quadruplexes (G4s) in the mRNA[68,69]. We first quantified the highest number of predicted G4s in Satb2 mRNA among classical neuronal fate determinants (Fig. S12a–d). We hypothesized that increased G4s in Satb2 makes its mRNA translation uniquely sensitive to eIF4A1 and thereby also Ire1α level.

We first constructed fluorescence-based translational reporters by fusing 5′UTRs of Satb2 and CTIP2 to EGFP ORF. In these reporters, quantification of EGFP fluorescence intensity is a measure of 5′UTR-dependent translation efficiency. Next, we co-electroporated each

reporter with CRISPR-Cas9 plasmids to induce *eIF4A1* KO in E12.5 cortical progenitors. At E16.5, we observed decreased Satb2 reporter translation efficiency (Fig. 9a and b) and increased CTIP2 reporter translation efficiency (Fig. 9c and d), associated with lower Satb2 expression in CTIP2 reporter-expressing KO neurons (Fig. 9e, f). We detected minor effects using the reporters in *eIF4A1* KO and *Ire1α* KO at E14.5 (Fig. S12e–h).

We observed analogous effects for translational efficiencies for Satb2 and CTIP2 in *Ire1α* KO MEFs transfected with each reporter (Fig. 9g–l), congruent with the nuclear endogenous Satb2 and CTIP2 expression levels (Fig. 9i and l).

Finally, using EUE in early cortical progenitors from control and *Ire1α* cKO, we co-expressed wild-type eIF4A1and its helicase activity-deficient mutant[68,70] (Fig. 9m) with either of the reporters (Fig. 9n–r). We quantified the reporter fluorescence in a Ki67 expression-specific manner (Fig. 9o) and found that wild-type eIF4A1 restored the

**Fig. 7 | Ire1α-mediated regulation of protein translation in the developing cortex. a** Images of representative primary cortical neurons prepared from *Ire1α^f/f* (CTR) or *Ire1α^f/f*; *Emx1^Cre/+* (cKO) cortices after ribopuromycilation assay to label stalled ribosomes. **b** Quantification of ribosome stalling in (**a**). **c** Representative Western blotting results using the control and *Ire1α* cKO E18.5 cortical lysates. **d** Quantifications of the protein level from **c**. Representative images of immunostaining against EGFP and CTIP2 in E18.5 coronal cortical sections of wild-type embryos after IUE at E12.5 with gRNAs and Cas9 nickase to achieve indicated genotypes. **f** Quantification of neuronal cell identity in the experiment in (**e**). **g**–**j** Representative Western blotting results using the control and *Ire1α* cKO cortical lysates at indicated developmental stages and quantification of protein levels. Note that the quantification for Ire1α levels in (**h**) is identical with Fig. 5l. Coomassie brilliant blue, CBB. **k** Representative Western blotting results of HEK293T cells after 4 h pulse with indicated compound. **l**-**m** Representative Western blotting of CHX pulse experiment in HEK293T treated with indicated compounds and quantification. **n** Representative Western blotting results of endogenous Ire1α co-immunoprecipitation (IP) from E12.5 and E14.5 cortical homogenates (H). **j** Interaction between Ire1α and indicated proteins was quantified relative to the amount of immunoprecipitated Ire1α. Graphs represent data points and averages ±

S.D. Thick lines on violin plots represent median, thin lines represent quartiles. CBB, Coomassie Brilliant Blue stain to visualize proteins in SDS-PAGE gels. Statistics for **b**, **d**, **f**, **h**, **j** and **o** D'Agostino-Pearson normality test; for **b** unpaired t-test, $n_{cells}$ for CTR = 22 and for cKO=32 from three independent cultures, $p < 0.0001$; **d** Mann–Whitney test, $n_{cortices}$ for CTR = 4 and for cKO=4, for eEF-2, $p = 0.0286$; for eEF-2-P, $p = 0.0286$; for eIF4A1, $p = 0.0286$; **f** unpaired t-test, for Satb2 counts, $n_{brains}$ for CTR = 4 and for KO = 4, $p = 0.0140$, and for CTIP2 counts, $n_{brains}$ for CTR = 4 and for KO = 4, $p = 0.0691$; **h** Mann–Whitney test, for Ire1α, $n_{cortices}$ for CTR = 12 and for cKO=10, $p = 0.0169$; for eIF4A1, $n_{cortices}$ for CTR = 12 and for cKO=10, $p = 0.3463$; for eEF-2, $n_{cortices}$ for CTR = 12 and for cKO = 10, $p = 0.2276$; for eEF-2-P, $n_{cortices}$ for CTR = 7 and for cKO = 7, $p > 0.9999$; **j** unpaired t-test, for Ire1α, $n_{cortices}$ for CTR = 10 and for cKO = 11, $p = 0.0284$; for eIF4A1, $n_{cortices}$ for CTR = 10 and for cKO = 11, $p = 0.0494$; for eEF-2, $n_{cortices}$ for CTR = 10 and for cKO = 11, $p = 0.5561$; for eEF-2-P, $n_{cortices}$ for CTR = 7 and for cKO=7, $p = 0.8440$; **m** two-way ANOVA with Šidák multiple comparisons test, n = 4 biological replicates, $p < 0.0001$; **o** unpaired t-test, $n_{cortices}$ for E12.5 = 4 and for cKO=4; for RPS6, $p = 0.0007$, for RPL7, $p = 0.3751$, for PABPC4, $p = 0.7826$. Statistical tests were two-sided. $0.01 <* p < 0.05$; $0.001 <** p < 0.01$; $*** p < 0.001$.

decreased translation of Satb2 in E12.5 cKO cortical progenitors (Fig. 9p). We observed analogous effects for CTIP2 (Fig. 9r), with wild-type eIF4A1 reducing the enhanced CTIP2 translation in cKO progenitors, in contrast to its Q195E mutant.

These results indicate that eIF4A1-driven timed regulation of translation downstream of Ire1α is embedded in the 5′UTR of Satb2 (Fig. 9s).

Representative pictures for neuronal fate quantification can be found in Fig. S13 and original pictures of membranes used for representative protein biochemistry in Fig. S14.

## Discussion

Development of the cortex comprises an orchestrated series of sequential, tightly controlled gene expression events with many layers of regulation. We make the surprising finding that protein synthesis rates are intrinsic features of distinct progenitor lineages and differentiated neurons (Figs. 1–4). Our data implicate the regulation of protein synthesis by a mechanism downstream of Ire1α in the specification of Satb2-positive neurons and their polarization. We report a developmental role for Ire1α beyond its canonical role in translation stress pathways.

The demand to synthesize and remodel the proteome during cortical development places homeostatic pathways on the brink of cellular stress signaling. We speculate that such demands in normal development are met by the same molecular players as stress pathways, however, likely leading to unique downstream pathways active in development. During cortical development, Ire1α safeguards the cellular translation flux by driving the expression of translation regulators, essential for the high protein synthesis rate in neuronal progenitors, enabling translation of Satb2 (Figs. 6–9).

Our translation rate measurements in cultured cortical cells enable tight control of microenvironment and identical exposure to the methionine analog (Figs. 2a and 4e). We show that cultured immature neurons at DIV1 derived from E12.5 brain translate at higher rates than ones from E14.5 brain in vitro. In P2 cortex, L5 CTIP2 neurons translate at higher levels than L2/3 Satb2 neurons, while L6 CTIP2 neurons and DL Satb2 neurons exhibit lower translation rates (Fig. 4).

It is becoming increasingly clear that, on top of elegant transcriptional regulation[18,19], generation of cortical neuron diversity relies on specific pathways for translational control. According to our data, the 5′UTR in Satb2 mRNA requires activity of translation initiation complexes, comprising eIF4A1, which unwind cap-proximal regions of mRNA, especially ones with G4s, prior to its ribosomal loading[71]. In line with this are our observations on decreased translation rates upon

eIF4A1 KO (Fig. 8) associated with defects in specification of Satb2-expressing neurons (Fig. 7). Loss of *eIF4A1* or *Ire1α* leads to a decreased translation efficiency of Satb2 reporter. On the other hand, the CTIP2 reporter is translated more efficiently, associated with lower Satb2 expression in neurons (Fig. 9f).

Global attenuation of translation during the UPR is linked to PERK-mediated phosphorylation of eIF2α at Ser52[72]. The stress response alters the cellular translation machinery and its affinity towards ORFs of specific type, position, and secondary structures[73,74]. We report that loss of *Ire1α* results in lower rates of protein synthesis in neuronal progenitors and neurons, independently of eIF2α, or its canonical downstream splicing client Xbp1 (Figs. S3, S9 and S11). *Ire1α* regulates the amount of translating polysomes, with their higher abundance in the cKO. In the context of lower translation flux, this represents ribosome stalling (Figs. 6 and 7) and/or slower elongation (Fig. 8), reminiscent of observations in *Huntingtin* mutants[62]. As Ire1α binds 80S ribosomes directly with high affinity (Fig. 7n, o and[25]), and is in the vicinity of the ribosomes embedded in the ER by the translocon, it is well positioned to impact protein synthesis directly during cortex development. The interaction between Ire1α with small ribosomal subunit is strong at E12.5, when multipotent progenitors able to generate Satb2- and CTIP2-expressing neurons are present in the cortex (Fig. 7n, o).

Apart from being a site of protein synthesis, the ER has been also reported as an organelle crucial for neuronal polarization[75]. Upon *Ire1α* KO, we report a mislocalization of the ER to the polarizing neurite (Fig. S4). Defects in axon formation, and thus axonal targeting of Tau-1, AnkG and Na_vs (Fig. S5) might indicate ER dysfunction upon *Ire1α* loss, implicating its role in ER integrity and neuronal polarization[76]. Disrupted polarity in *Ire1α* KO neurons (Fig. S4) is associated with the loss of ER-dependent microtubule stability and Golgi apparatus (Fig. S6), overall leading to altered current sensitivity (Fig. S5). It is tempting to speculate that the role of Ire1α in regulating the generation of Satb2-expressing neurons is molecularly intertwined with the axon formation, given the conservation of differentiation programs across neuronal lineages[19,77].

Taken together, our study reinforces the powerful impact of post-transcriptional mechanisms in cortex development. Translational regulation of gene expression during neuronal specification is layered on top of transcriptional mechanisms, many of which have been highlighted in excellent recent works[18,19]. Modulation of translation rates may be a particular requirement for cortical neuron diversification, evolving as a mechanism to specify upper layers in the late stages of development. In our comparative proteomics, we identify many cellular pathways, of which the effector proteins are specifically

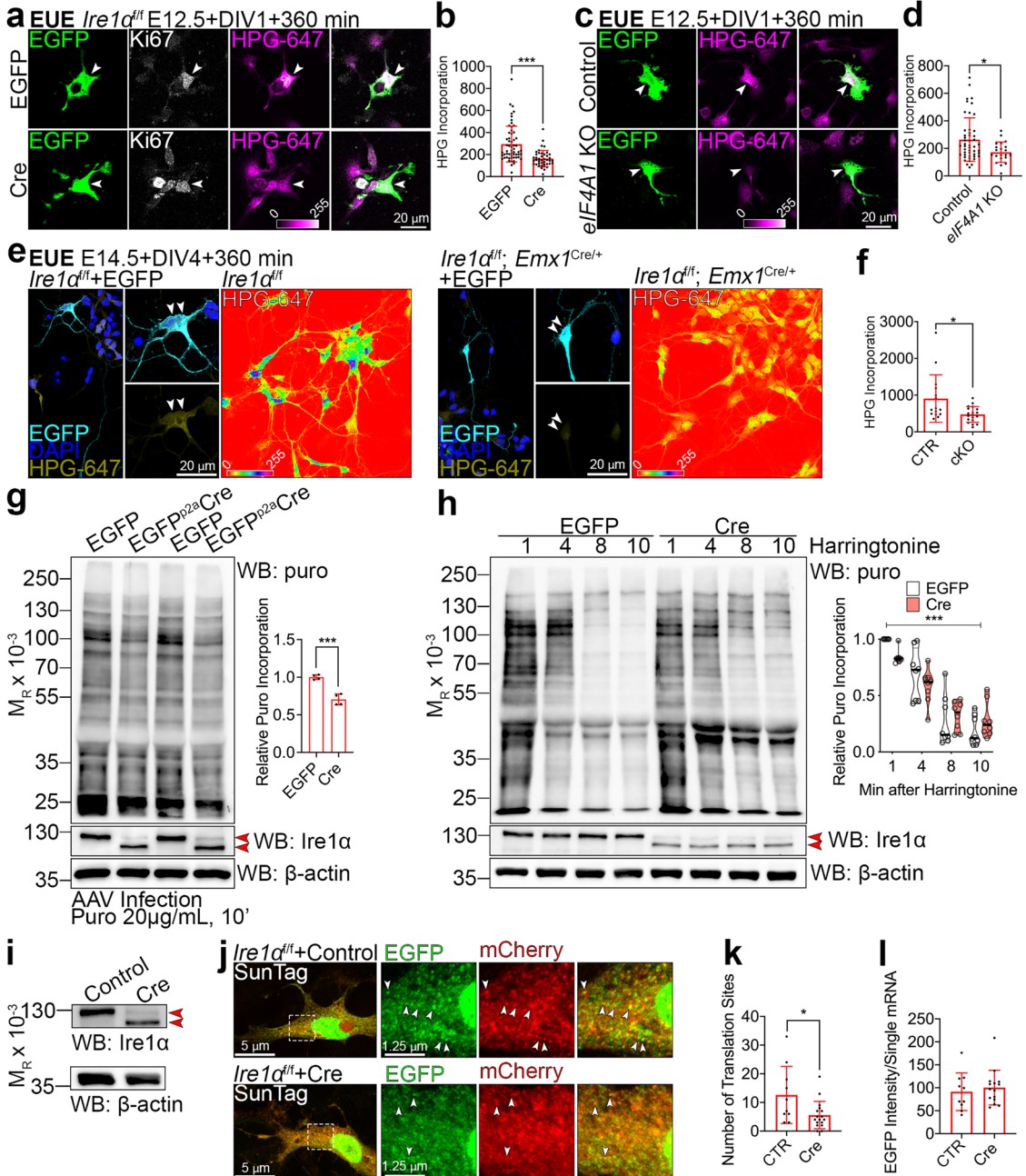

**Fig. 8 | Loss of Ire1α leads to diminished translation rates as an effect of slower elongating ribosomes and fewer translation sites. a, c, e** Images of representative primary cortical neurons prepared from *Ire1α^f/f* **a** wild-type **c** or control and *Ire1α* cKO **e** embryos after ex utero electroporation (EUE) at E12.5 (a and c) or E14.5 **e** with indicated plasmids. Neurons were fed L-homopropargylglycine (HPG) for 360 min prior to fixation at DIV1 or DIV4. HPG was detected with Sulfo-Cyanine5 azide. **e** Right panels: images of HPG incorporation in control and cKO primary DIV4 neurons prepared from E14.5 cortex. White arrowheads point to Ki67-positive progenitors (a and c) or to somata of neurons derived from E14.5 progenitors (**e**). **b, d, f** Quantification of HPG incorporation. **g** Representative Western blotting using DIV5 lysates from *Ire1α^f/f* mouse embryonic fibroblasts (MEFs) after metabolic labeling of protein synthesis using puromycin (puro) and its quantification. MEFs were infected at DIV0 with control or Cre-expressing AAVs. Red arrowheads point to wild-type and KO form of Ire1α. **h** Representative Western blotting results of ribosome run-off assay using puromycin in control and KO MEFs at indicated timepoints after harringtonine treatment and quantification. **i** Western blotting

validation of *Ire1α* KO in AAV-infected MEFs for the SunTag reporter experiment. **j** Representative images of empty and Cre-encoding virus infected MEFs expressing the SunTag24x-BFP-PP7 reporter. **k** Active translation sites were quantified in fixed MEFs. **l** Quantification of the intensity of scFv-GFP at translation sites. Bar graphs represent data points and averages ± S.D. Violin plot on **h** represents individual data points, thick line median and thin lines quartiles. Statistics for **b, d, f, k, l** D'Agostino-Pearson normality test; for **b** Mann-Whitney test, $n_{cells}$ for EGFP = 58 and for Cre = 42 from three independent cultures, $p < 0.0001$; **d** Mann–Whitney test, $n_{cells}$ for Control=54 and for Cre=21 from three independent cultures, $p = 0.0291$; **f** Mann-Whitney test, $n_{cells}$ for EGFP = 15 and for Cre=19 from three independent cultures, $p = 0.0169$; **g** Shapiro-Wilk and unpaired t-test, four independent cultures, $p = 0.0003$; **h** two-way ANOVA with Bonferroni multiple comparisons test, eight independent experiments, $p < 0.0001$; **k** Mann–Whitney test, $n_{cells}$ for CTR = 10 and for Cre=17 from three independent cultures, $p = 0.0438$; **l** unpaired t-test, $n_{cells}$ for CTR = 10 and for Cre=15 from three independent cultures, $p = 0.58$. Statistical tests were two-sided. $0.01 <* p < 0.05$; *** $p < 0.001$.

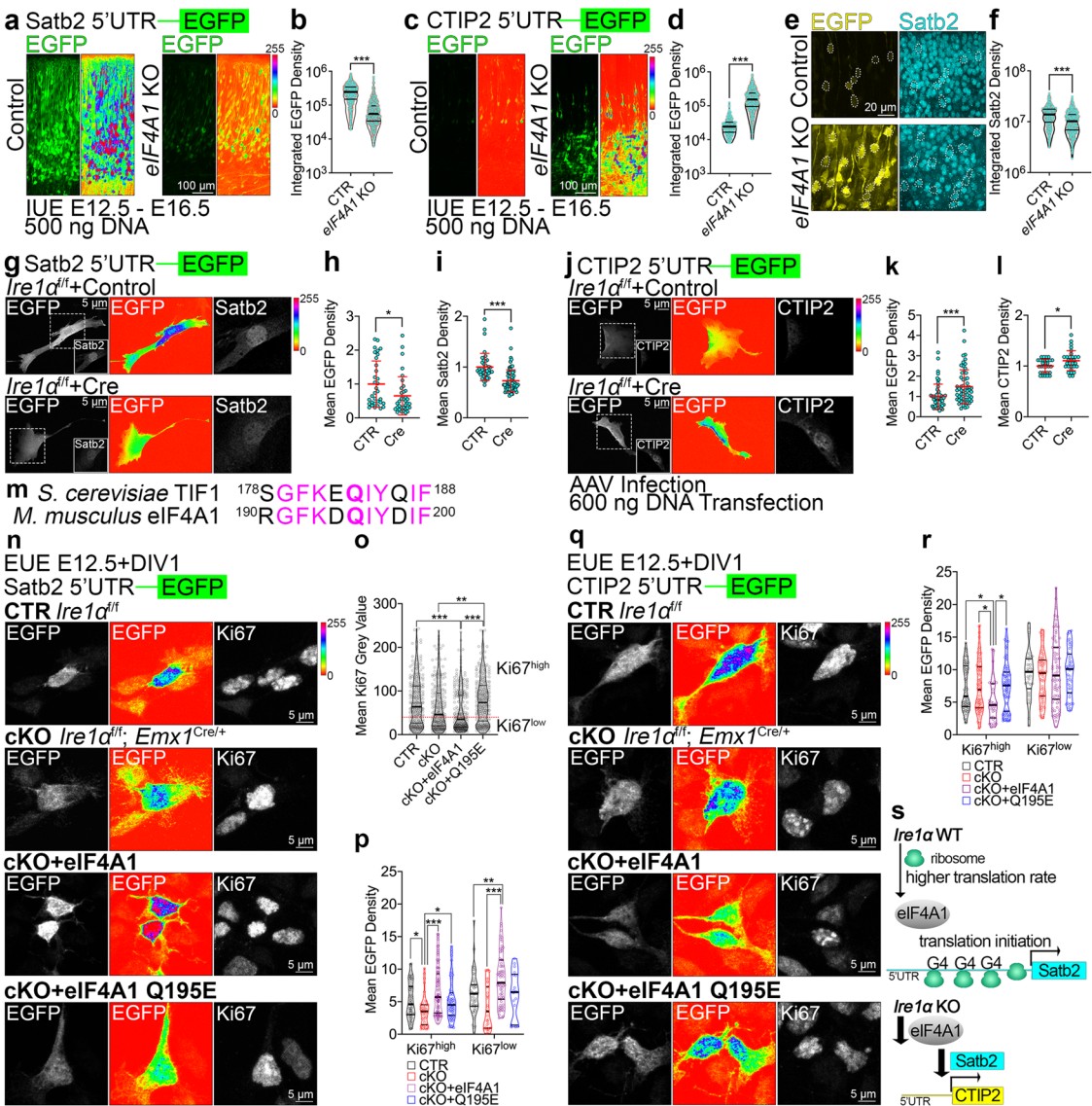

**Fig. 9 | Helicase activity of eIF4A1 and Ire1α are indispensable for translation of Satb2 in cortical lineages. a, c** Representative images of EGFP fluorescence signals in the E16.5 brain sections after IUE at E12.5 with CRISPR-Cas9 vectors to achieve indicated genotypes and Satb2 5′UTR **a** or CTIP2 5′UTR **c** translation reporter construct. Shown are the native signals (left panels) and intensity encoding (right panels). **b, d** Quantification of EGFP fluorescence signals in single cells expressing translation reporters of Satb2 5′UTR **b** or CTIP2 **d. e, f** Representative images and quantification of CTIP2 5′UTR translation reporter construct and anti-Satb2 immunolabeling (compare **c, d**). **g–l** Representative images of EGFP fluorescence signals (left panels: gray scale, middle panels: intensity encoding) and immunostaining for Satb2 **g** or CTIP2 **j** in *Ire1α*^f/f MEFs infected with Control or Cre-encoding AAVs at DIV0. At DIV5, infected MEFs were transfected with indicated reporter constructs, fixed and immunostained at DIV6. **h, k** Quantification of EGFP mean fluorescence signals of 5′UTR Satb2 reporter **h** or CTIP2 (**k**). **i, l** Quantification of mean nuclear fluorescence signals after immunostaining for Satb2 **i** or CTIP2 (**l**). **m** Sequence alignment of yeast TIF1 and murine eIF4A1. In bold the Q residue crucial for helicase activity. **n–r** Early cortical progenitors of indicated genotypes

were transfected with EUE with Satb2 **n** or CTIP2 **q** translational reporter, as well as indicated constructs. At DIV1, cells were fixed and immunolabeled against EGFP and Ki67. **o** Quantification of overall Ki67 fluorescence intensity in analyzed cells. **p, r** Quantification of translational reporter fluorescence in Ki67 expression level-dependent manner. **s** Current model of Ire1α and eIF4A1 interplay in regulation of neuronal cell diversity in the cortex. Violin plots depict median, interquartile range (box) and minimum and maximum value (whiskers). Red line and error bars on **h, i, k, l** indicate mean ± S.D. For statistical analyses, **b, d, f, h,i, k, l** D′Agostino and Pearson normality test and Mann-Whitney test; **o, p, r** D′Agostino and Pearson normality test and Kruskal-Wallis test with Dunn′s correction. For **b, d, f** $p < 0.0001$; for exact p values in **o, p, r** refer to Supplementary Fig. 1. For **h** $n_{cells}$ for CTR = 33 and for Cre = 35, $p = 0.0105$; **i** $n_{cells}$ for CTR = 39 and for Cre = 85, $p < 0.0001$; **k** $n_{cells}$ for CTR = 53 and for Cre = 58, $p = 0.0003$; **l** $n_{cells}$ for CTR = 30 and for Cre = 30, $p = 0.0388$. Results on **h-i** and **k-l** represent quantifications from three independent cultures. Statistical tests were two-sided. $0.01 <* p < 0.05$; $0.001 <** p < 0.01$; $*** p < 0.001$.

translated in early and later-born neuronal lineages (Fig. 3). It is foreseeable that other translational regulators alongside Ire1α drive the increased translation efficiency in progenitors of later-born upper layer neurons. Hence, translational control is deployed in normal development and on a par with protein degradation orchestrates the neuronal diversification and wiring of the mammalian cortex. Importantly, it has become evident that the RNA transcripts are not the

principal determinants of protein abundance and cellular fate in developmental systems[78–80].

## Methods

### Animals

Mouse (*Mus musculus*) lines described in this study have been maintained in the animal facilities of the Charité University Hospital and

Lobachevsky State University. Mice were housed in a 12 h light/dark cycle at 18-23 degrees of Celsius, 40-60% air humidity, with pellet food and water available ad libitum. Mice were used at embryonic (E), or postnatal (P) stages, as reported for each experiment. Each sample included both sexes within litters without distinctions. Wild-type mice were of NMRI strain. Females were housed in groups of up to five animals per cage. Males were single housed from the point of exposure to female animals. In *Ire1α*^f/f mouse line[48], exons 20-21 of *Ire1α* were engineered to be flanked by loxP sites. To inactivate *Ire1α* in the developing cortex, we crossed *Ire1α*^f/f mice with the *Emx1*^Cre/+ line, in which Cre recombinase is expressed from the *Emx1* gene allele (Gorski et al., 2002). As described in our previous work, for the breeding we exclusively used *Ire1α*^f/f males and *Ire1α*^f/f; *Emx1*^Cre/+ females to circumvent leaky expression of Cre[81]. *Ire1α*^f/f; *Emx1*^Cre/+ was further crossed to MetRS* mouse line[36] and to Fucci2aR mouse line[32]. Animals homozygous for the loxP alleles in both lines were viable, fertile, born at the expected Mendelian ratio, and exhibited no overt phenotypic changes in the cage environment. For experiments with tdTomato reporter, *Satb2*^Cre/+[39] males were mated to NMRI wild-type females. The date of vaginal plug was counted as E0.5. All mice were sacrificed by administering a lethal dose of pentobarbital.

## Inclusion and ethics statement

MCA is a signatory of the ALBA Declaration on Equity and Inclusion. All experiments were performed in compliance with the guidelines for the welfare of experimental animals approved by the State Office for Health and Social Affairs, Council in Berlin, Landesamt für Gesundheit und Soziales (LaGeSo), permissions G0079/11, G0206/16, G0184/20, G0054/19, G0055/19, and by the Ethical Committee of the Lobachevsky State University of Nizhny Novgorod.

## Sex and age/developmental stage of animals for in vivo experiments

Littermates of both sexes were randomly assigned to experimental groups during experimental procedures or collection of embryonic tissue. Developmental stages or stages at experimental interventions are listed on the figures or in the figure legends.

## Murine primary cortical cultures

Embryonic brains were dissected in ice-cold Hank's Balanced Salt Solution (HBSS) under stereomicroscope to collect cortices in 5 mL of ice-cold 0.25% trypsin (Gibco) solution. After incubation at 37 °C for 20 min with occasional swirling, tissue digestion was terminated by addition of 1 mL of fetal bovine serum (FBS, Gibco). Further, 1 U of DNaseI (Roche) was added to the tube for 1 min. Next, cortices were carefully washed twice with 5 mL of pre-warmed (37 °C) Complete Neurobasal medium and triturated in 1 mL Complete Neurobasal medium using P1000 and P200 pipette tip. Cells were counted with Naubauer counting chamber and seeded in 24 well-plate formats at 120,000 cells (EUE) or 240,000 cells (after nucleofection) per coverslip in 1 mL of Complete Neurobasal medium. Glass coverslips were coated with poly-L-lysine (70,000–150,000, 0.01%, P4707, Sigma) and laminin (from Engelbreth-Holm-Swarm murine sarcoma basement membrane, L2020, Sigma) dissolved in PBS. The day of neuronal prep was counted as day in vitro 0 (DIV0). Cells were cultivated at 37 °C in the presence of 5% carbon dioxide in HERA-cell240 (Heraeus) incubator. For the experiments with DMSO and CHX, cortical cultures were treated with the assayed compound two hours post-plating. Autaptic hippocampal cultures were prepared from P0-P2 *Ire1α*^f/f mice[82].

**Complete neurobasal.** 500 mL Neurobasal (Gibco, Life Technologies), 10 mL B-27 (Gibco, Life Technologies), 5 mL GlutaMAX (Gibco, Life Technologies), 5 mL penicillin/streptomycin (Life Technologies, Gibco), 1 mL Primocin (Invitrogen).

## Mouse embryonic fibroblasts (MEFs) from *Ire1α*^f/f mice

MEFs were collected from E13.5 *Ire1α*^f/f embryos. Briefly, abdominal wall was dissected, minced with a sterile razor blade and incubated with 1 mL 0.25% trypsin (Gibco) at 37 °C for 30 minutes. Trypsinization reaction was terminated by addition of 4 mL MEF medium and the tissue was triturated 20 times to release single cells. Cellular suspension was transferred to T75 flask with 15 mL of fresh MEF medium and cells were incubated until confluency. Cells were split according to standard cell culture protocols. Cell cultures were expanded and frozen at passage five, flash frozen in liquid nitrogen and stored until use at −80 °C. Before each experiment, correct genotype of the fibroblasts was validated using the DNA isolated from the cell suspension according to the genotyping protocol for *Ire1α*. MEFs were infected using AAVs according to standard infection protocols. Given that the primary fibroblasts come from randomly assigned embryos, these cells contain clones from both sexes.

**MEF medium.** A total of 500 mL DMEM (Life Technologies), 50 mL FCS (Gibco), 5 mL penicillin/streptomycin (Life Technologies, Gibco), 5 mL GlutaMAX (Gibco).

## Cell lines

HEK293T, Neuro-2A, and NIH3T3 cells lines were from DSMZ and were maintained in the standard medium (MEF medium). The cells were routinely tested negative for mycoplasma contamination prior to the experiments.

## Molecular cloning strategies for constructs generated in this study

Cloning of all expression vectors in this study was performed using the NEBuilder system according to the manufacturer's protocol (New England BioLabs). DNA fragments were amplified using GXL Prime Star DNA polymerase (Takara) using cDNA libraries or donor plasmids as templates. Destination vectors were linearized with EcoRI-HF (New England BioLabs).

**pRai-HA-IRE1α.** Human IRE1α cDNA (NM_001433.3) was amplified from plasmid template [Addgene, #13009[83]], using the following oligos: 5'- agattacgctatctgtacaggcATGCCGGCCCGGCGGCTG-3' and 5'-ggccgctagcccgggtaccgCTTGGTTTGGGAAGCCTGGTCTCCCTGC-3' and inserted into the modified pRaichu vector[31].

**pCAGIG-6XHis-eEF2.** Murine eEF2 cDNA (NM_007907.2) was amplified from cDNA library using the following oligos: 5'-gtctcatcatttttggcaaagATGCATCATCATCATCATCATGTGAACTTCACAGTAGATC-3' and 5'-cggccgcgatatcctcgaggCTACAGTTTGTCCAGGAAGTTG-3' and inserted into pCAGIG vector for simultaneous expression of 6XHis-tagged eEF2 and EGFP.

**pCAG-5'UTR-Satb2 and -CTIP2 reporter.** Satb2 and CTIP2 5'UTR sequences were amplified using a mouse embryonic cDNA library and Prime Star GXL polymerase. An existing pCAG-IRES-tdTom vector was linearized using EcoRI digestion and PCR-amplified fragments were inserted and fused together with a GFP coding sequence. Primers for PCR had the following sequences: 5'UTR Satb2_fwd, 5'-gtctcatcatttttggcaaagCGCCCCCATCATCATAAC-3'; 5'UTR Satb2_rev, 5'-ccttgctcacCATGCTGCTCCGATTTGG-3'; GFP_fwd, 5'-gagcagcatgGTGAGCAAGGGCGAGGAG-3'; GFP_rev, 5'-cggccgcgatatcctcgaggTTACTTGTACAGCTCGTCCATG-3'; 5'UTR CTIP2_fwd, 5'-gtctcatcatttttggcaaagAATTTATTTTAGCCTTTTCTCTATTTTAGAGCAAG-3'; 5'UTR CTIP2_rev, 5'-ccttgctcacCATGCCCCGGCATCTATTC-3'; GFP_fwd, 5'-cggggcaatgGTGAGCAAGGGCGAGGAG-3'; GFP_rev, 5'-cggccgcgatatcctcgaggTTACTTGTACAGCTCGTCCATG-3'. The fragments were recombined with the vector backbone using NEBuilder system. After that, the 5'UTR fragments

fused to GFP and IRES sequence were shuttled to a pCAG vector[84] and fused with dsRed encoding sequence also using NEBuilder system.

**eIF2α variant replacement experiment.** eIF2α was knocked down using a pLKO Sigma Mission plasmid system. The genetic replacement was carried out using co-transfection of the pLKO and the human eIF2α, its phosphorylation-deficient S52A or -mimic S52D, which were gifts from David Ron purchased from Addgene (#21807, http://n2t.net/addgene:21807; RRID: Addgene_21807; #21808, http://n2t.net/addgene:21808; RRID: Addgene_21808; #21809, http://n2t.net/addgene:21809; RRID: Addgene_21809).

**eIF4A1 and translational rescue experiment using the 5'UTR Reporters.** The helicase activity-reduced Q195E variant[70] of eIF4A1 were gifts from Hans-Guido Wendel (Addgene plasmid #70044, http://n2t.net/addgene:70044; RRID: Addgene_70044[68]) purchased from Addgene.

The wild type eIF4A1 was extracted from murine cDNA library using the PCR with the following primers: 5'-gtctcatcattttggcaaagATGgactacaaagacgatgacgacaagTCTGCGAGTCAGGATTCTC-3' and 5'-cggccgcgatatcctcgaggTCAAATGAGGTCAGCAACGTTG-3' and cloned into pCAG expression vector using EcoRI restriction digest and NEBuilder system.

**pCAG-Xbp1S.** The wild type Xbp1S was extracted from murine cDNA library using the PCR with the following primers: 5'-atctgtacagaattcggtaccATGGACTACAAAGACGATG-3' and 5'-gcggccgctagcccgggtacTTAGACACTAATCAGCTGG-3' and cloned into pRai expression vector using EcoRI restriction digest and NEBuilder system.

**pCAG-IRE1α S724F.** Mutagenesis of pcDNA3.1-IRE1α was performed using Q5 Site-Directed Mutagenesis Kit and primers 5'-GGGCAGACACgctTTCAGCCGCCG-3' and 5'-ACTGCCAGCTTCTTGCAG-3' and the resulting mutated IRE1α sequence was cloned into pCAG vector using NEBuilder strategy.

### Lentivirus and AAV production and infection of autaptic hippocampal cells

Production of lentiviral particles and infection of autaptic cultures for electrophysiology was performed by the Charité Viral Core Facility. For other experiments in this report, we used pAAV-CAG-EGFP, pAAV-CAG-iCre-p2a-EGFP-WPRE, pAAV-CAG-iCre-WPRE3, and pAAV-CAG-WPRE3 particles.

### Electrophysiological recordings

Voltage and current clamp experiments were performed according to our previously published works[85] using autaptic hippocampal neurons from P0-P2 *Ire1α*^f/f infected with the control and Cre-encoding lentiviral particles[84] produced by the viral core facility of the Charité-Universitätsmedizin Berlin. Whole-cell patch-clamp recordings were performed on the autaptic cultures at room temperature at DIV12-20. Synaptic currents were recorded using a Multiclamp 700B amplifier (Axon Instruments) controlled by Clampex 9.2 software (Molecular Devices). Membrane capacitance and series resistance were compensated by 70%, and data was filtered by a low-pass Bessel filter at 3 kHz and sampled at 10 kHz using an Axon Digidata 1322A digitizer (Molecular Devices). Neurons were perfused (SF-77B, Warner Instruments) with the extracellular solution (in mM): 140 NaCl, 2.4 KCl, 10 HEPES (Merck), 10 glucose (Carl Roth), 2 CaCl$_2$ (Sigma-Aldrich, St. Louis, USA), and 4 MgCl$_2$ (Carl Roth) (-300 mOsm; pH7.4). Somatic patches were carried out using borosilicate glass pipettes, with a tip resistance of 2-3.5 MOhm and filled with the following internal solution (in mM): 136 KCl, 17.8 HEPES, 1 EGTA, 4.6 MgCl$_2$, 4 Na$_2$ATP, 0.3 Na$_2$GTP, 12 creatine phosphate, and 50 U/ml phosphocreatine kinase (-300 mOsm; pH7.4). Data were analyzed using AxoGraph software.

### Nucleofection of primary neurons

Transfection of primary cortical cells was performed using nucleofection (Lonza Bioscience) according to the manufacturer's instructions[38]. Primary cortical cells isolated from murine embryos were transfected using the Amaxa Mouse Neuron Nucleofector Kit (Lonza, VPG-1001). After the trituration, cell concentration was determined using the Neubauer chamber. For the nucleofection, we used 1 μg DNA per 1 million cells. Accounting for cell death, we plated 120,000 cells per well of 24 well plates for immunocytochemistry and 240,000 per well of 24 well plates for biochemical assays.

### Small molecule inhibitor screening

Right after trituration, E13.5 embryonic cortical cells prepared from *Satb2*^Cre/+ mice, were nucleofected with pCAG-EGFP and pCAG-loxP-Stop-loxP-tdTomato plasmids and seeded at 120,000 cells per well of 96-well plate. Two hours post-plating, cultures were treated with compounds at two concentrations, in technical duplicates. Cells were then cultivated until DIV2, when the proportion of *Satb2*^tdTom neurons normalized to EGFP positive cells was determined using FACS[38]. TdTom+ (Q2 + Q4) cell count was divided by the number of EGFP (Q1 + Q2) expressing cells and then normalized to the "Untreated" condition. For the detailed result per inhibitor, refer to Supplementary Data S3.

### Ex utero electroporation (EUE)

DNA solution (final concentration of 500 ng/μL) was microinjected into the lateral embryonic ventricles, followed by electroporation using 6 pulses of 35 V applied using platinum electrodes. Right after the electroporation, the isolated tissue was placed in ice cold HBSS until proceeding with primary cultures.

### Immunocytochemistry

Cells cultured on coverslips were fixed with cold (4 °C) 4% PFA and 4% sucrose (Merck) PBS solution for 20 minutes at room temperature, washed three times with PBS and incubated with Blocking buffer for 30 minutes. Next, cells were incubated with primary antibodies diluted with blocking buffer accordingly for 16–20 h at 4 °C with moderate shaking, followed by washing 3 times with PBS. Further, secondary antibodies coupled to appropriate fluorophore diluted in blocking buffer were applied for 2 hours at room temperature. After washing 3 times with PBS, coverslips were briefly rinsed with ddH$_2$O and mounted on Superfrost Plus glass slides (Thermo Scientific) with Immu-Mount mounting medium (Shandon, Thermo-Scientific).

**Blocking buffer.** 5% horse serum (Gibco, Life Technologies), 0.3% Triton X-100 (Roche), in PBS buffer.

### Puromycin labeling in MEFs

To metabolically label proteins using puromycin, we used day-in-vitro 5 (DIV5) *Ire1α*^f/f mouse embryonic fibroblasts infected with AAV particles (Charité Viral Core Facility) encoding for EGFP (control) and EGFP-p2a-Cre at DIV0. Cells were incubated with 20 μg/mL puromycin (Sigma) for 10 minutes[33]. The cells were washed with PBS and lysed in 1X Laemmli sample buffer at 70 °C for 20 minutes. Proteins were then resolved with SDS-PAGE and analyzed using Western blotting with anti-puromycin antibody (Millipore). The puromycin incorporation was normalized to the protein loading amount, visualized by Coomassie Brilliant Blue (CBB), or an appropriate loading control.

### Puromycilation in primary cortical cultures

For translatome characterization, primary cultures were prepared according to the standard protocol described in "Primary cortical cultures" section[30]. The cerebral cortices of embryos at E12.5 and E15.5 were isolated in ice-cold Hank's Balanced Salt Solution without MgCl$_2$ and CaCl$_2$ (HBSS −/−, Gibco) and harvested in ice-cold HBSS with

$MgCl_2$ and $CaCl_2$ (HBSS +/+, Gibco). The cortices were washed three times with HBSS +/+. The tissue was digested by adding 1 ml of 2.5 % trypsin and 0.02 mg/ml DNase to 5 ml of HBSS +/+ and following incubation for 20 min at 37 °C. The trypsinization was stopped by adding 2 ml of FBS. The tissue was washed three times with Neurobasal media without supplements. The cortices were triturated in 1 ml of full Neurobasal media (500 mL Neurobasal A (Gibco), 10 mL B-27 (Gibco), 5 mL GlutaMAX (Gibco), 5 mL penicillin/streptomycin (Gibco)). The cortices were then gently triturated in 1 mL of full Neurobasal media. In the following step, cell counting was performed using the Neubauer counting chamber. The cells were plated in 6-well plates at a density of one million cells per well. Cells were cultivated at 37 °C in the presence of 5% CO2. At DIV1 neurons were incubated with Puromycin (Sigma) at a final concentration of 10 μg/mL for 10 minutes. Afterward, cells were lysed in E1A buffer (50 mM HEPES, 150 mM NaCl, 0.1% NP40, pH 7.0) supplemented with protein inhibitors cocktail, 2.5 mM $MgCl_2$, and 0.025 U/μl benzonase and sonicated. The lysate was cleared by centrifugation at maximum speed (20 kg) for 10 minutes at 4 °C. The supernatants were transferred to a new tube. Immunoprecipitation was performed using Sepharose beads, that were washed three times with E1A buffer before the experiment to get rid of traces of ethanol. The beads were incubated for one hour with an anti-puromycin antibody (Millipore) in E1A buffer with rotation at 4 °C. The residual unbound antibodies were removed by washing with E1A buffer. Then, the lysates were incubated with beads for 3 hours at 4 °C with rotation. After incubation, the beads were washed twice with E1A buffer. Then, the samples were analyzed with the standard SDS-PAGE and Western blotting to verify puromycin incorporation and then for mass spectrometry to identify enriched peptides in the IP as well as to quantify steady-state expression in the input (Supplementary Data S2).

### Puromycin immunoprecipitation mass spectrometry

To remove NP-40 from the input samples, the following were added with vortexing after each step: 4 volumes of methanol, 1 volume of chloroform, and 3 volumes of ultrapure water. The samples were centrifuged at 15000xg for 2 minutes to remove the aqueous layer, and 4 volumes of methanol were added followed by vortexing. Samples were centrifuged again at 15000xg for 2 minutes. After the removal of the supernatant, the samples were dried with a SpeedVac. Precipitates were resuspended in 50 μl of fresh 100 mM ammonium bicarbonate. This was followed by a tryptic digest including reduction and alkylation of cysteines. Reduction was performed by adding tris(2-carboxyethyl)phosphine to a final concentration of 5.5 mM at 37 °C on a rocking platform (800 rpm) for 30 minutes. For alkylation, chloroacetamide was added to a final concentration of 24 mM at room temperature on a rocking platform (800 rpm) for 30 minutes. Proteins were then digested with 200 ng trypsin (Roche, Basel, Switzerland) per sample by shaking at 800 rpm for 3 hours at 37 °C. Samples were acidified by adding 1 μL of 100% formic acid (2% final concentration), and centrifuged briefly to spin down the beads. The supernatants containing the digested peptides were transferred to a new low-protein binding tube. Peptide desalting was performed on C18 columns (Pierce). The eluates were lyophilized and reconstituted in 20 μL of 5% acetonitrile and 2% formic acid in water, vortexed briefly, and sonicated in a water bath for 30 seconds before injecting 2 μL into the nano-LC-MS/MS. IP samples in bead slurry proceeded directly to reduction, alkylation, and digestion as described above.

LC-MS/MS was carried out by nanoflow reverse-phase liquid chromatography (Dionex Ultimate 3000, Thermo Scientific) coupled online to a Q-Exactive HF Orbitrap mass spectrometer (Thermo Scientific), as reported previously[86]. Briefly, the LC separation was performed using a PicoFrit analytical column (75 μm ID × 50 cm long, 15 μm Tip ID; New Objectives, Woburn, MA) in-house packed with 3-μm C18 resin (Reprosil-AQ Pur, Dr. Maisch, Ammerbuch, Germany). Peptides were eluted using a gradient from 3.8 to 38% solvent B in solvent

A over 120 min at 266 nL per minute flow rate. Solvent A was 0.1 % formic acid and solvent B was 79.9% acetonitrile, 20% $H_2O$, and 0.1% formic acid. Nanoelectrospray was generated by applying 3.5 kV. A cycle of one full Fourier transformation scan mass spectrum (300 – 1750 m/z, resolution of 60,000 at m/z 200, automatic gain control (AGC) target $1 \times 10^6$) was followed by 12 data-dependent MS/MS scans (resolution of 30,000, AGC target $5 \times 10^5$) with a normalized collision energy of 25 eV. To avoid repeated sequencing of the same peptides, a dynamic exclusion window of 30 sec was used.

Raw MS data were processed with MaxQuant software (v2.0.1.0) and searched against the murine proteome database UniProtKB UP000000589 with 55,366 protein entries, released in March 2021. MaxQuant database search parameters were a false discovery rate (FDR) of 0.01 for proteins and peptides, a minimum peptide length of seven amino acids, a first search mass tolerance for peptides of 20 ppm, and a main search tolerance of 4.5 ppm. A maximum of two missed cleavages were allowed for the tryptic digest. Cysteine carbamidomethylation was set as a fixed modification, while N-terminal acetylation and methionine oxidation were set as variable modifications. The mass spectrometry data have been deposited at the ProteomeXchange Consortium (http://proteomecentral.proteomexchange.org) via the PRIDE partner repository[87] with the dataset identifier PXD048919.

### Gene Ontology Analysis in Puromycin-IP Mass Spec

Based on the normalized abundance of identified proteins in the IP fraction, a rank for each protein was quantified, where smallest rank was assigned to proteins of highest intensity. Further, based on the spread of the ranks across all identified peptides, we defined top 30% on the list as ones preferentially regulated at E12.5 or E15.5. Further, we analyzed the list of proteins lowly translated at E12 and highly translated at E15 with PANTHER Classification System 18.0, over-representation test (released 2024-02-26), GO biological process (Fisher's Exact Test, FDR, $p < 0.05$), GO Ontology database DOI: 10.5281/zenodo.10536401 Released 2024-01-17; Supplementary Data S2), and data plotted in RStudio with the package ggplot2.

### Ribosome run-off assay

To quantify ribosome run-off, we used DIV5 *Ire1α*^f/f mouse embryonic fibroblasts infected with AAV particles (Charité Viral Core Facility) encoding for EGFP (control) and EGFP-p2a-Cre at DIV0. Cells were incubated with 2 μg/mL harringtonine for indicated time and immediately after with 20 μg/mL puromycin for 10 minutes. Next, cells were washed with PBS and analyzed as described for "Puromycin metabolic labeling". Puromycin incorporation was normalized to the protein loading amount and to the incorporation in Control at 1 minute.

### SunTag

To study translation dynamics using the SunTag system, we used DIV5 mouse embryonic fibroblasts from *Ire1α*^f/f infected with AAV particles (Charité Viral Core Facility): empty virus (control) and ones encoding for Cre. At DIV4, MEFs were transfected with SunTag labeling plasmids including pcDNA4TO-24xGCN4_v4-BFP-24xPP7 (Addgene 74929), pHR-scFv-GCN4-sfGFP-GB1-NLS (Addgene 60906), and pHR-tdPP7-3xmCherry (Addgene 74926). After 20 hours, the reporter expression was induced with 1 μg/mL doxycycline (Sigma) for 1 hour[65]. Next, cells were fixed, mounted on SuperFrost slides and fluorescence was detected using spinning disc microscopy.

### Ribopuromycilation assay

To visualize translating ribosomes in neurons, we used a previously published protocol with slight modifications[59,88]. To inhibit translation initiation neurons were treated with 5 μM Harringtonine for 10 min. Next, neurons were incubated with Neurobasal medium containing 5 μM Harringtonine (Sigma), 100 μM Puromycin (Sigma), and 200 μM

Emetine (Sigma) for 5 minutes. Cells were washed then with HBSS+/+ with 0.0003% digitonin (Sigma) for 2 minutes, and once with ice-cold HBSS+/+. Next, neurons were fixed with 4% PFA and 3% sucrose in PBS. Cells were stained with anti-puromycin antibody (Millipore). On the fluorescence images obtained on a Zeiss Spinning Disk microscope, the number of dots per cell was quantified using a cell counter plugin in the Fiji program.

### Fluorescent noncanonical amino acid tagging (FUNCAT)

For the fluorescent labeling of newly synthesized proteins, we modified previously published methods[89]. In brief, neuronal cultures were fed with 1 mM HPG-supplemented Methionine-free Neurobasal Medium, at DIV1 and DIV5, for the time indicated on the figures and in the figure legends. Cells were then extensively washed in PBS prior to fixation with 4%PFA/4% sucrose for 20 min. Control and HPG fed neurons were then fluorescently labeled via reaction with Sulfo-Cyanine5 azide (sCy5az, 1 μM, Lumiprobe). To facilitate azide-alkyne binding we applied sCy5az in a PBS based "click solution" containing 0.2 mM Tris(3-hydroxypropyltriazolylmethyl) amine (THPTA), 20 mM sodium L-ascorbate, and 0.2 mM copper (II) sulfate pentahydrate (all from Sigma-Aldrich). Cells were incubated at room temperature for 20 minutes, before PBS washing and immunocytochemistry (described above).

**Met-free neurobasal medium.** 500 mL Met-, Lys-, Arg-free Neurobasal (Gibco, Life Technologies), 10 mL B-27 (Gibco, Life Technologies), 5 mL GlutaMAX (Gibco, Life Technologies), supplemented with final 1 mM L-HPG (Jena), 0.8 mM Lys, and 0.4 mM Arg.

### Methionine analog labeling using MetRS* (FUNCAT and BONCAT)

The previously published method[36] was modified in this study to label newly synthesized proteins. P0 brains of $Emx1^{Cre/+}$; MetRS*; $Ire1\alpha^{f/f}$ mouse line were used to prepare 450-500 μm live slices, which were subsequently incubated in artificial cerebrospinal fluid (NaCl 125 mM; NaHCO₃ 25 mM; NaH₂PO₄ 1.25 mM; CaCl₂ 2 mM; glucose 25 mM; MgCl₂ 1 mM) supplemented with 4 mM ANL for 24 hours at 37 °C in the presence of 5% CO₂. Next, slices were fixed in 4% PFA in PBS for 30 minutes at room temperature (RT) and passed through a sucrose gradient (until 30% sucrose / PBS). Slices were then mounted in OCT compound (Tissue Tek) and transferred to dry ice to freeze. Sections at 16 μm thickness were then collected with cryotome (Leica) and directly processed for fluorescent non-canonical amino-acid tagging (FUN-CAT). Sections were washed three times in PBS to remove the residue of the OCT compound. DakoPen was applied on the borders of the slide to establish a hydrophobic barrier. Firstly, sections were incubated in 0.5% Triton X-100 in PBS for 30 minutes at RT. Next, the incorporated ANL in the brain sections was "clicked" for two hours at RT with alkyne-TAMRA reaction solution (0.1 mM copper sulfate; 0.5 mM THPTA, 5 mM sodium ascorbate, 10 μM alkyne-TAMRA, 5 mM aminoguanidine). For a detailed description of the reagents, refer to Supplementary Data S5. Sections were washed three times with PBS. After washing, samples were incubated with a blocking solution consisting of 0.5% Triton X-100, 10% goat serum, 5% sucrose, and 2% fish skin gelatine (Sigma) in PBS. Sections were then incubated overnight at 4 °C with primary antibodies (anti-GFP to visualize MetRS* expression, anti-Satb2, anti-CTIP2 to identify cell fate) and DAPI as a nuclear marker dissolved in blocking solution. Further, samples were washed three times in PBS and incubated for four hours at RT with secondary antibodies. After incubation, sections were washed in PBS and mounted with ImmoMount medium and a cover glass. Fluorescence images were obtained on a Zeiss Spinning Disk microscope with a 40x objective. For TAMRA signal quantification, we used single-plane images. First, we identified Satb2- or CTIP2-positive cells in respective cortical layer using Cell Counter (Fiji). Next, the somata were outlined,

and the integrated signal intensity from such obtained region of interest (ROI) was measured. Raw integrated densities normalized to the area of each cell were plotted individually on a graph.

For BONCAT, we used a modified version of the previously published protocol[90]. Cortices were lysed in 300 μl Buffer A (1% SDS, 50 mM Tris-Cl pH 7.5, 180 mM NaCl) supplemented with 2.5 mM MgCl₂, 0.025 U/μl benzonase, and protein inhibitors. Samples were dissociated by careful pipetting up and down approximately 20 times avoiding foaming and incubated for 2 minutes at RT. Next, the samples were incubated for 20 minutes at 70 °C with the following centrifugation for 15 minutes at 14000 rpm and RT. 30 μl of the supernatant were taken for input #1 and mixed with 10 μl of 4X Lämmli. 200 μl of the supernatant was transferred to a new tube and diluted with 200 μl of Buffer B (1% SDS, 50 mM Tris-Cl pH 7.5, 180 mM NaCl, 1% Triton X-100). The click reaction was performed by adding DBCO-PEG4-biotin (Jena Bioscience, final concentration 100 μM) and incubating for one hour at RT with rotation. After incubation, 50 μl of samples were taken for input number #2 and mixed with 20 μl of 4X Lämmli. The rest 350 μl was diluted with 550 μl Buffer B and subsequently used for the streptavidin immunoprecipitation using magnetic beads (Thermo Fisher Scientific). After incubation, samples were eluted with excess biotin and diluted with a 4X Lämmli buffer prior to western blotting.

### Cryosectioning

For all histological procedures, brain sections were prepared on Leica CM3050S and RWD Minux FS800 cryostat. Prior to cryosectioning, brains were passed through a sucrose gradient to cryoprotect the tissue. Next, brains were frozen in −38 to −40 °C isopentane (Roth), or on dry ice. For standard histological purposes, free-floating coronal cryosections of 50 μm thickness were collected in PBS/0.01% sodium azide solution. For in situ hybridization 16 μm thickness sections were collected.

### Organotypic slice culture

For the slice culture, we used a previously published protocol with slight modifications[30]. After placing the slices on the membrane, they were left in an incubator for one hour to settle. Puromycin (Sigma, final concentration 10 μg/mL) was then added to the medium and the slices were incubated for 30 minutes at 37 °C in the presence of 5% CO₂. Sections were fixed in 4% PFA and 3% sucrose in PBS overnight. After fixation, slices were washed with PBS and further subjected to a standard staining protocol with an anti-puromycin antibody (Millipore) and DAPI (as a nuclear marker). Fluorescence images were obtained on a Zeiss Spinning Disk microscope with a 40x objective.

### Immunohistochemistry

Fixed brain sections were washed with PBS three times at room temperature prior to the procedure to remove the sucrose and freezing compound residue. The sections were then incubated with Blocking solution for one hour at room temperature, then with the primary antibody and DAPI diluted in blocking buffer for 16–20 h at 4 °C, washed in PBS three times for 30 minutes and incubated with secondary antibody diluted in the blocking buffer for up to four hours at room temperature. Next, sections were incubated with PBS for 30 minutes three times and mounted with cover glass (Menzel-Gläser) and Immu-Mount mounting medium (Shandon, Thermo-Scientific).

**Blocking solution.** 5% horse serum, 0.5% (v/v) Triton X-100, PBS.

### Fluorescence in situ hybridization (FISH)

We used RNAScope Technology to detect mRNA of Ire1α (RNAScope Probe-Mm-Ern1, 438031) according to the manufacturer's protocol for RNAscope Multiplex Fluorescent V2 Assay (ACDBio, 323100). Prior to hybridization, embryonic brains at E12.5, E14.5 and E18.5 were collected

in DEPC-PBS and incubated in 4%PFA/PBS/DEPC for 16-20 hours at 4 °C. Brains were then passed through a series of sucrose solutions (10%-20%-30%/PBS) until they reach osmotic equilibrium, embedded in OCT. Compound (Tissue-Tek) in a plastic cryoblock mold and frozen on dry ice. Coronal sections of 16 μm thickness were collected using the cryostat.

## BrdU Pulse and Immunohistochemistry

BrdU was pulsed according to standard laboratory practice[6]. Brains of both genotypes (Control: *Ire1α*^f/f and cKO: *Ire1α*^f/f; *Emx1*^Cre/+) were fixed in 4% PFA, passed through sucrose gradient and frozen in isopentane bath. Cryosections of 50 μm (P2) or 16 μm (embryonic stages) thickness were placed on the SuperFrost Plus glass slides and let dry overnight at +4 °C. Antigen retrieval was performed in boiling citrate-based Antigen Unmasking Solution (Vector, pH 6.0) for ten minutes in a microwave, followed by 20 min of cooling on ice. Next, sections on slides were further processed for immunohistochemistry.

## In utero electroporation (IUE)

Prior to IUE, DNA was diluted in TE buffer and mixed with 0.1% Fast Green FCF (Sigma-Aldrich). Final concentration of DNA used for transfecting cortical progenitors was 500 ng/μL. For the experiments with the reporters, the DNA concentration is specified on the figures. Silencing RNAs (ON-TARGETplus Non-targeting Control Pool, D-001810-10-05; ON-TARGETplus Mouse Ern1 78943 siRNA, SMARTpool, L-041030-00-0005, Horizon Discovery) were used at final concentration of 50 nM. For the experiments described in this paper, 6 pulses of 37 V were applied using platinum electrodes. Molecular cloning strategies for the plasmids can be found under the "Molecular cloning" section. The origin of plasmids used for the IUE can be found in Supplementary Data S5. For experiments in *Ire1α*^f/f, we used EGFPiCre plasmid, to simultaneously express EGFP and Cre, using bicistronic construct with IRES sequence. In this plasmid, Cre sequence is modified to reduce the high CpG content of the prokaryotic coding sequence (improved Cre, iCre) to reduce a chance for epigenetic silencing. Plasmids for IUE contained beta-actin promoter (pCAG), or NeuroD1 promoter.

## FlashTag labeling and cortical progenitor isolation

To prepare the working solution of FlashTag Cell Trace Yellow (Thermo Fisher Scientific) the lyophilized powder was dissolved in 50 μL of DMSO. The Fast Green (Sigma) dye was added to allow easier visualization of the fluid when injected. The procedure was performed under sterile conditions. The lateral ventricles of embryonic brains were injected with a glass capillary with 0.5 μl of FlashTag solution[19]. The embryonic cerebral cortices were isolated and placed in ice-cold HBSS with MgCl$_2$ and CaCl$_2$ (HBSS+/+, Gibco). Before trypsinization, brains were washed by centrifuging at 600 rpm for 1 minute. Then, the supernatant was removed and replaced with 5 ml of fresh HBSS+/+. The tissue was digested by adding 500 μl of 2.5 % trypsin, 0.02 mg/ml DNase, and 1 μl of 25U/μl benzonase to 5 ml of HBSS +/+ and following incubation for 20 min at 37 °C. The reaction was stopped by adding 2 ml of FBS. The samples were washed three times with FACS buffer (2% FCS, 0.02 mg/ml DNase, 1 μl of 25U/μl benzonase). Next, the cortices were carefully triturated in 300 μl of FACS buffer. Subsequently, samples were transported on ice in FACS buffer to the cell sorting facility. Cell sorting was performed by the BIH Cytometry Core Facility on a BD FACSAria Fusion (BD Biosciences, San Jose, CA, USA), configured with 5 lasers (UV, violet, blue, yellow-green, red). Samples were collected after sorting in 1.5 ml of FACS buffer. The cells were then centrifuged for 15 minutes at 1000 rpm. The samples were subsequently frozen and stored at −80°C until further use.

## Expansion microscopy

For the purpose of expansion of coronal brain sections to visualize morphology of EGFP-expressing neurons, we followed previously published protocol[91]. For immunohistochemistry, we used anti-EGFP antibody (Rockland) at 1:200 and the secondary Alexa488-coupled one at 1:100. After PBS washes, the sections were incubated with AcX crosslinker (10 mg/mL) in 150 mM NaHCO$_3$ for 20 hours at room temperature, washed with PBS, incubated for one hour with monomer solution (19% sodium acrylate, 10% acrylamide, 0.1% bis-N',N'-methylene-bisacrylamide, 0.01% 4-hydroxy-TEMPO in PBS) and gel was polymerized by addition of 0.2% APS/0.2% TEMED for two hours at 37 °C. The section embedded in the gel was then incubated with a proteinase K in a digestion buffer (50 mM Tris-Cl, 800 mM guanidine-Cl, 2 mM CaCl$_2$, 0.5% Triton X-100 in pH 8.0) for 20 hours at 37 °C with gentle agitation. Next, the section was transferred to a Petri dish with ddH$_2$O, which was changed at least five times for two days. The expansion factor was determined before mounting of the specimen. Sections were then transferred onto the glass-bottom chambers (Ibidi) and imaged using Leica Sp8 confocal.

## Biochemical experiments

Details of the protocols used for sodium dodecyl sulfate polyacrylamide gel electrophoresis (SDS-PAGE) and Western blotting were performed according to standard lab practice and are thoroughly described in our previous works[30,31]. For all experiments in this manuscript, we used nitrocellulose membranes. Unless explicitly stated otherwise, all cellular assays were performed by directly lysing the cells in culture with Lämmli buffer, supplemented with 2.5 mM MgCl$_2$, 0.025 U/μl benzonase, and protein inhibitors (0.1 mM PMSF, 1 mg/L aprotinin, 0.5 mg/L leupeptin). Acquisition of the chemiluminescence for the biochemical experiments in this manuscript was performed using BioRad Molecular Imager, ChemiDoc XRS+ and Azure Biosystems 600. For the image analysis, we used ImageLab, ImageStudioLite and Fiji software.

## Preparation of cortical lysates

For quantitative Western blotting using embryonic tissue, dissected and frozen cortices were lysed in an appropriate volume of lysis buffer (1% SDS, 50 mM Tris-Cl pH 7.5, 180 mM NaCl) supplemented with 2.5 mM MgCl$_2$, 0.025 U/μl benzonase, protein inhibitors (0.1 mM PMSF, 1 mg/L aprotinin, 0.5 mg/L leupeptin) and phosphatase inhibitors (Roche). After homogenization, cortices were incubated for 20 minutes at 70 °C with the following centrifugation for 15 minutes at 12,000 g. The supernatant was then transferred to a separate tube, and the pellet was discarded. Protein concentration was determined using the BCA method (Thermo, Pierce).

## Quantification of protein levels

Quantitative Western blotting was performed using ImageJ and Image Studio Lite (LI-COR). Protein levels were normalized to the total protein amount in the sample visualized by a Coomassie staining (NB-45-00078, Neo-Biotech) or to a loading control, Gapdh, vinculin, or β-actin.

## Co-Immunoprecipitation (Co-IP)

Mouse cortices were lysed in E1A lysis buffer (150 mM NaCl, 50 mM HEPES, pH 7.0, 0.1% NP-40), supplemented with protease inhibitor cocktail (Roche), 2.5 mM sodium pyrophosphate, 1 mM beta-glycerophosphate, 10 mM sodium fluoride, 1 mM Na$_3$VO$_4$, and 1 mM PMSF (Millipore). For a single assay point, four cortices of E12.5 and two of E14.5 were pooled and lysed together. The lysate was centrifuged at 12,000 g for 15 min and the supernatant with 0.25 mg total protein was incubated with 0.5 μg anti-Ire1α (Cell Signaling) antibody for 4 hours at 4 °C. Next, the supernatant was incubated with 25 μL Sepharose A beads (Roche) for 1 hour at 4 °C. As a control, the supernatant was incubated with 0.5 μg rabbit IgG (Cell Signaling). After extensive washing with E1A buffer, bound proteins were eluted in 1X Lämmli sample buffer and incubated at 70 °C for

20 minutes, prior to SDS-PAGE and Western blotting. To quantify the interaction between Ire1α and respective translational regulators at E12.5 and E14.5, the density of indicated proteins were normalized to the density of Co-IPed Ire1α and to the strength of interaction at E12.5.

## Antibodies

For Western blotting, all antibodies were diluted 1:750, apart from anti-β-actin (used at 1:2000), and anti-puromycin (used at 1:10000), in 5% non-fat dry milk in TBS-T buffer (TBS buffer with 0.1% Tween-20). Antibodies against phospho-proteins, anti-puromycin and anti-Ire1α were diluted in 3% BSA in TBS-T buffer. Secondary antibodies coupled with HRP were diluted 1:10000 in the same buffer as the primary ones. For immunocytochemistry, all primary antibodies were diluted 1:300 in the blocking buffer, and the appropriate fluorophore-conjugated secondary antibodies were used at 1:750. For immunohistochemistry, primary antibodies and the respective fluorophore-linked secondary ones were diluted 1:500 in the blocking buffer. A detailed description of antibodies used in this study can be found in Supplementary Data S5.

## Polysome purification

Ribosome fractionation and polysome purification from E18.5 cortex was performed according to the published protocol[56]. Seven neocortices (14 hemispheres) of each genotype were pooled per biological replicate. Neocortices from animals of both sexes were dissected on ice in ice-cold PBS, snap-frozen and stored at −80 °C. Frozen tissue pellets were lysed by cryogenic grinding in the lysis buffer: 20 mM HEPES, 100 mM KCl, 10 mM MgCl$_2$, pH 7.4, supplemented with 20 mM Dithiothreitol (DTT), 0.04 mM Spermine, 0.5 mM Spermidine, 1x Protease Inhibitor cOmplete EDTA-free (Roche, 05056489001), 200 U/mL SUPERase-In RNase inhibitor (ThermoFisher, AM2694), 0.3% v/v IGE-PAL CA-630 detergent (Sigma, I8896). Tissue lysates were clarified by centrifugation at 16,000 $g$ for 10 minutes at 4 °C. Sucrose density gradients were prepared in Beckman Coulter Ultra-Clear Tubes (344060 for 14 mL 5–45% gradients). Base buffer consisted of 20 mM HEPES, 100 mM KCl, 10 mM MgCl$_2$, 20 mM DTT, 0.04 mM Spermine, 0.5 mM Spermidine, 1x Protease Inhibitor cOmplete EDTA-free (Roche, 05056489001), 20 U/mL SUPERase-In RNase inhibitor (ThermoFisher, AM2694), pH 7.4, prepared with either 5 & 45% or 10 & 50% sucrose w/v. Overlaid sucrose-buffer solutions were mixed to linearized gradients with a BioComp Gradient Master 107ip. Neocortical lysates were overlaid on gradients pre-cooled to 4 °C. The gradients were centrifuged in a SW40 rotor (Beckman Coulter) for 5 h, 4 °C, 25,000 rpm and fractionated using a BioComp Piston Gradient Fractionator and Pharmacia LKB SuperFrac, with real-time A260 measurement by an LKB 22238 Uvicord SII UV detector recorded using an ADC-16 Pico-Logger and associated PicoLogger software. Collected samples were stored at −80 °C for downstream analysis and RNA isolation and sequencing.

## Isolation of RNA, cDNA library preparation and RNA sequencing

Tools and work surfaces used for RNA work were thoroughly cleaned with 70% ethanol (Sigma) and rinsed with RNA-Zap (Thermo Fisher) before the procedure. Embryonic brains at E18.5 were dissected in DEPC-PBS and flash-frozen in liquid nitrogen until the purification procedure. RNA was isolated using RNeasy columns (RNeasy Mini Kit, Qiagen). Further steps were performed according to the manufacturer's protocols. RNA was eluted from the silica membranes using 30 μL of molecular biology-grade (MB)- H$_2$O. Quality and concentration of the prepared RNA were determined using an Agilent Bioanalyzer. Single-end (total cortex) or paired-end (polysome) TruSeq Stranded total RNA libraries were made with Ribo-Zero Gold rRNA depletion and sequenced using Illumina Nextseq 500/550, and in later runs Novaseq 6000.

## Analysis of RNAseq Experiments

All code used in analysis of RNAseq can be found at https://github.com/qoldt/IRE1aKO_Polysome_RNAseq. Computation has been performed on the HPC for Research cluster of the Berlin Institute of Health. Briefly, reads were quantified with Salmon quant --validateMappings[92] using mouse genome assembly GRCm39 and gencode comprehensive gene annotation release M33. Technical replicates were collapsed, and transcripts were quantified for Differential Transcript Usage using DRIMseq, Differential Transcript Expression and Differential Gene Expression were performed using DESeq2 with batch incorporated in the design matrix. Unique genes found to be differentially expressed in Polysome, Heavy Polysome and Light Polysome were used to generate a heatmap using complex-Heatmap. GSEA analysis was performed using gseGO on a ranked gene list with minGSSize = 20, maxGSSize = 1000, against all categories (BP, CC, MF), and notable GSEA categories were plotted using gseaplot2 from the clusterProfiler package in R. Enrichment Map was created using the first 8 significant categories using emapplot from the enrichPlot package[92].

## RNA structure prediction

The sequences of murine neuronal fate determinants mRNA were identified using the UCSC Genome Browser (Mouse Assembly GRCm39/mm39). The structures of 5'UTRs were modeled using the RNA Web Suite http://rna.tbi.univie.ac.at/cgi-bin/RNAWebSuite/RNAfold.cgi.

## G-quadruplexes estimation

The identification and quantification of G-quadruplexes (G4s) within the 5' untranslated regions (5'UTRs) of the mouse genome were conducted employing pqsfinder. The genomic coordinates of the 5'UTR were extracted from the Genecode Release M12 (GRCm38.p5) reference genome obtained from the National Center for Biotechnology Information (NCBI). The computational algorithm embedded in the pqsfinder package was employed for a deep search of overlapping G4 motifs within the 5'UTRs[93]. Default parameters were utilized. The quantification of G4s was performed at both the transcript and gene levels from the output of the pqsfinder function. For the latter, the sum of all G4 instances across transcripts associated with a particular gene was calculated, along with the mean of the corresponding scores. Gene annotations were obtained from the Ensembl database. To assess the accuracy of G4 predictions, the base call accuracy was computed as a surrogate measure. This metric was derived from the pqsfinder scores using the formula:

$$base\ call\ accuracy = 1 - 10^{-\frac{Score}{10}}$$

This approach provides a quantitative representation of the reliability of G4 predictions, with higher base call accuracy values indicative of more robust predictions.

## Confocal imaging of immunostaining signals and analysis

Images of primary cell cultures and brain sections after immunostaining were acquired using a confocal Leica SL, Leica Sp8 or Zeiss Spinning Disc Microscope. Images of the brain slices exposed to puromycin were acquired using Nikon Scanning Confocal A1Rsi+.

## Colocalization with identity markers

Neuronal identity was quantified as a fraction of all EGFP-expressing neurons. The identity of primary cells on coverslips was determined as a fraction of fluorescent reporter-expressing cells. Based on the numbers of dsRed- or EGFP-expressing cells on a coverslip, we either quantified at least 4 fields of view for high-density cultures, or imaged a given number of cells (indicated on the data plots and Supplementary

Data S1) and determined the number of cells expressing a marker protein for cultures with sparse labeling, i.e. after EUE.

### Laminar positioning (% of CP)

We based our positioning analysis on previous published reports[94]. To determine the position of neurons in the cortical plate, confocal images of EGFP signals were first transformed so that the pia is perpendicularly oriented to the horizontal axis. Next, the positions of neurons were marked in Fiji using the Cell Counter plugin. Using the y-coordinate, we then expressed the position of a given cell relative to the size of the CP and plotted it as % CP, with 0% being the bottom of the cortical plate / the subplate (SP) and a 100% being the pial surface / the marginal zone (MZ). Positions of all cells were then plotted individually on the graph. Only brains with comparable electroporation efficiencies were analyzed. We also provided the normalized quantification using cortical bins[30]. For this, the number of neurons in each of the five cortical bins was normalized by the total number of electroporated neurons within a given cortical section.

### Quantification of fluorescence intensity

For quantifications of HPG-647 fluorescent intensity, we defined a region of interest based on the fluorescent marker, i.e. dsRed or EGFP signals. Using Fiji, the raw integrated density was then normalized to the cell surface area of the outlined cell compartment. For our quantifications, we imaged the primary cell cultures as Z-stacks, and for the analysis, we used maximum intensity projections. For quantification of EGFP fluorescence intensity in cells electroporated with 5'UTR reporter, the integrated intensity was quantified, given homogenous cell size. We sampled from not less than three electroporated cortices. Only cortices with similar electroporation efficiencies were used for the analysis. For the experiments in fibroblasts transfected with the reporter constructs, because of highly variable cell sizes, mean fluorescence in a defined cell area normalized to the average of control was plotted. For the SunTag analysis, mean EGFP fluorescence intensity in a given mCherry puncta was quantified using the Cell Counter plugin.

### Quantification of Venus- and mCherry-expressing cells in Fucci2aR mouse line

We used the *Emx1*[Cre/+]; Fucci2aR*; *Ire1α*[f/f] mouse line to investigate cell cycle dynamics upon loss of *Ire1α*[32]. Brains were isolated at stage E14.5 and fixed in 4% PFA in PBS. Next, for the cryoprotection brains were passed through a sucrose gradient starting from 10% sucrose and ending in 30% sucrose in PBS. For coronal sections with 16 μl thickness, brains were embedded in OCT block. After cutting, the samples were immunostained using a standard protocol with the following primary antibodies: anti-GFP (Rockland), anti-RFP (Chromotek) and DAPI as a nuclear marker. Fluorescence images were obtained on a Zeiss Spinning Disk microscope with a 40x objective. The proportions of cells positive for GFP and RFP were counted using Fiji software. Cells were marked with the Cell Counter plugin. Cell proportions were normalized to the total number of cells in the selected area (i.e. number of cells positive for DAPI).

### Microtubule stability measurements

To characterize the distribution of acetylated and tyrosinated tubulin in neurons, fixed cells were immunostained with anti-acetyl-alpha-tubulin and anti-tyrosinated-alpha-tubulin antibodies. Using Fiji software, the soma and longest neurite were manually outlined, and then the integrated signal intensity in these regions was measured.

### Morphometry, axon counting in vitro, Golgi apparatus positioning, polarity classification in vivo

Details concerning neuronal morphometry, axon counting assays in vitro, and quantification of polarity were performed according to

our published works[30,31]. Dendritic complexity was measured using Sholl analysis, computed in Fiji, using EGFP signal in single neurons after nucleofection or EUE. We plotted the number of intersections with concentric circles (1 μm increment) as a function of the distance from the center of the soma. We also plotted the total number of intersections per cell. Axon length and branching were counted manually and plotted as density, expressing the number of primary branches per axon length per cell. For the morphometry, we only analyzed single non-overlapping neurons. Axon counting was performed in cortical cultures after staining with axonal Tau-1 and dendritic MAP-2. We plotted the proportion of neurons with no, single and multiple axons analyzed per group. Polarity classification in vivo was performed using neuronal reconstructions in Z-stacks based on EGFP fluorescence after in utero electroporation using neuronal segmentation plug-in Simple Neurite Tracer in Fiji. Neurons with a single primary leading process and a single trailing process were classified as bipolar. Neurons with additional primary neurites emanating from the soma, or with a deviated primary dendrite (>45-degree angle) were classified as non-bipolar. Golgi apparatus was visualized using anti-GM130 immunolabeling. We classified each neuron based on the distribution and the number of Golgi apparatus found in the cell. Additionally, we classified the cells based on the angle between the Golgi apparatus, the center of the soma and the base of the longest neurite using the angle tool in Fiji.

### Statistical analyses and reproducibility

All statistics were computed using Prism Graph Pad software (version 10.1.0, 264, released Oct 18, 2023 and previous versions). Detailed information on statistics can be found in the Supplementary Data S1. Description of statistical tests, definition of center, dispersion, precision, and definition of significance are also listed where appropriate in the figure legends. Briefly, the distribution of data points was determined using normality tests (D'Agostino-Pearson, Anderson-Darling, Shapiro-Wilk, or Kolmogorov-Smirnov). We used unpaired two-tailed t-test, or one/two-way ANOVA with respective post hoc test; Mann-Whitney test to compare between two groups, or Kruskal-Wallis test with Dunn's multiple comparison test for comparisons between multiple groups. For detailed information on all numerical values and sample size see Supplementary Data S1. Culture experiments, Western blotting as well as immunohistochemistry or FISH presented in this manuscript were repeated at least three times. Representative micrographs or blots are derived from the analyzed datasets. Each inhibitor (per dose) was tested on biological replicates (each from collapsed two technical replicates) because of the sheer magnitude of the tested substances. The flow cytometry data for the putative hits was further validated using immunocytochemistry (Fig. 5f and g). For detailed information on the reagents and critical assays, please refer to Supplementary Data S5.

### Reporting summary

Further information on research design is available in the Nature Portfolio Reporting Summary linked to this article.

## Data availability

The sequencing data generated in this study have been deposited in the GEO database under accession code GSE172489. The mass spectrometry proteomics data generated in this study have been deposited to the ProteomeXchange Consortium via the PRIDE[95] partner repository with the dataset identifier PXD048919. Source data file is provided with this paper as well as in Fig. S13 and S14. Source data are provided with this paper.

## Code availability

All code used in analysis of RNAseq can be found at https://github.com/qoldt/IRE1aKO_Polysome_RNAseq.

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

## Acknowledgements

This project was funded by the German Research Foundation, Deutsche Forschungsgemeinschaft (project number 455114826, to V.T.), and the Studienstiftung des Deutschen Volkes (to M.D.). Work in the lab was supported by the Fritz Thyssen Foundation (10.23.2.003MN, to M.C.A.) and the German Research Foundation, Deutsche Forschungsgemeinschaft (project numbers 515247130 and 536563141, to M.C.A.). M.C.I., B.Q., O.J., D.M., and M.L.K. were supported by the Max Planck Society. A.R. was supported by the Russian Science Foundation (project number 21-65-00017). M.C.A. is a Scholar of the FENS-Kavli Network of Excellence. We thank Mengfei Gao, Helge Ewers, the CAJAL Advanced Imaging Methods Course for Cellular Neuroscience, and Boehringer Ingelheim Fonds (to M.C.A.) for the introduction to expansion microscopy. We thank Rima Al-Awar (Ontario Cancer Institute) for the kinase inhibitor library, Paraskevi Bessa-Newman and Ekaterina Epifanova for sharing their experimental expertize, Hiroshi Kawabe for the pRai plasmid, Frederick Rehfeld for dsRed expression vector, Thorsten Trimbuch, Anke Schönherr, and Nadine Albrecht-Koepke from the Viral Core Facility of the Charité– Universitätsmedizin Berlin (vcf.charite.de) for providing AAV stocks, the BIH Cytometry Core Facility and the Advanced Medical Bioimaging Core Facility of the Charité. Additionally, we thank Erin Schuman, Susanne tom Dieck, and Beatriz Alvarez-Castelao for the MetRS* mouse line, and Nils Brose for his continuous mentorship and advice.

## Author contributions

M.C.A. and E.B. carried out the bulk of the experiments. M.C.A. conceptualized the work, assembled the figures, and wrote the manuscript, with the input of the co-authors. V.T. designed the inhibitor screening, initiated the project, and supervised the work in the lab. A.G.N. performed the sequencing experiments and the bioinformatic analyses. M.L.K., M.C.I., B.Q. and D.M. purified the ribosomes ex-vivo and carried out the puro-IP mass spectrometry and analysis. A.R. carried out the eIF2α experiments. T.S. performed the molecular cloning. R.D. carried out the bulk of biochemical assays, histological and cell culture experiments. M.B. recorded from autaptic cultures and analyzed the results. J.K. and M.D. curated data. P.T. shared his expertise and protocols for metabolic labelling and click chemistry. D.K. helped with the screening design and shared the molecule library. O.J. performed proteomic analyses. T.I. shared the *Ire1α^{f/f}* mouse line. M.R., C.M.T.S., and C.R. supervised works in their labs.

## Funding

## Competing interests

The authors declare no competing interests.
