## [Peer Review File · Nature Communications]

Protein translation rate determines neocortical neuron fateREVIEWER COMMENTS

Reviewer #1 (Remarks to the Author):

In this paper the authors sought upstream regulators of *Satb2*, a marker of upper layer neurons, using a library of small molecules. They were able to identify Inositol-Requiring Enzyme 1 α (Ire1 α), also known as ER-to-Nucleus Signaling 1 (Ern1), the main sensor of ER lumen homeostasis and regulator of the Unfolded Protein Response (UPR), involved in these processes. They reveal that the development of *Satb2*⁺ neurons requires high rates of protein translation. Translational control is aided by 5'UTR-embedded structural elements in fate determinant genes (e.g. *Satb2*). These post-translational mechanisms hence contribute to neuronal diversity during corticogenesis. There is important and novel data in this manuscript. It is interesting that pathways known to be involved in cellular stress signalling can also contribute to correct cortical layer formation, as revealed here.

We are told that Ire1 α is a bimodal transmembrane kinase and RNase. Additionally, Ire1 α regulates stress-independent remodeling of actin filaments by its association with filamin A. There has also been shown a direct interaction of Ire1 α with ribosomes in vitro. This latter function, described here as non-canonical, is the focus of this work. Can the authors more definitively rule out that the other possible functions of Ire1 α are involved in fate determination? The indications for a UPR-independent role of Ire1 α in the regulation of ribosome function and polysome level are rather weakly covered and this ought to be strengthened. Are all other potential functions of Ire1 α ruled out? Ire1 α cKO removes the kinase-extension nuclease (KEN) domain including the RNase active site (exon 20-21..) There is also a truncated protein still remaining, not shown specifically in this manuscript until Fig S4, and also not well explained. Is the truncated protein still expected to have functions?

There is a dynamic interaction of Ire1 α with the ribosome and regulation of eIF4A1 and eEF-2 expression. In one of the last conclusion sentences in the discussion the authors say: 'Given developmental downregulation of Ire1 α , it is plausible that other translational regulators alongside Ire1 α drive the increased translation efficiency in upper layer progenitors.' This sentence emphasizes a major timing question not well addressed in the manuscript. Throughout, the emphasis is on the role of Ire1 α for *Satb2* expression strongly associated with the production of upper layer neurons. However, is Ire1 α also important for the earlier-born deep layer neurons with different fates (ref 23)? When it acts in early progenitors, is it just influencing the number of early-born *Satb2*⁺ neurons produced or other early-born neurons as well? Or is it mainly acting early to later influence *Satb2*⁺ neuron production? Highest levels of Ire1 α are at E12.5 and upper layer neurons are produced mainly later – why is it required so early? There seem to be less effects on E14.5 born cells (Fig S2). Urra and colleagues show an impact on deep layer neurons with a reduced thickness of the *Tbr1* layer. This is not mentioned sufficiently that we understand layering issues when studying Figs 1 and 2. Is the effect on *Tbr1* solely a lamination problem? At E18.5 Ire1 α is beginning to be downregulated but we still see polysome accumulation in mutant cortices at this age. Polysomes are deregulated in which cell type? It seems important to determine this to know how much this phenotype is relevant. Does changed fate influence final position, this is also not so clear? Further explanations are required to explain all these results and to justify and clarify that Ire1 α is a determinant of 'upper layer fate'.

This manuscript also raises the question of translation rates in progenitors over time and this is all very interesting. Was such data predicted from previous single cell data? In the mutant situation, were proliferation / cell cycle rates changed? How much might this influence the phenotypes and changed fate observed?

Axon specification / formation data seem to dilute the progenitor message.

Further more specific points are as follows:

Fig 1 e /S1a / S1d: the branching phenotype is not obvious in the photos – which cells did the authors focus on? Were these cells mis-positioned? In general, it would be nice to know more about the final positions of the electroporated cells. Is their position normally related to their fate? Is their position related to their altered fate?

Cell position (S2): The graphs in d and h are not well explained. We are not told if there are BrdU +ve cells below the CP? In h the position of Satb2+ cells is changed although their numbers are unchanged, and this is not mentioned until the later section. How much does the impact on lower layers affect these results? Double-labeled CTIP+ Satb2+ cells are not commented on? These seem like interesting points to further discuss. The fate change could also be emphasized since overall BrdU + cells are not changed in number.

Axon specification: before doing quantifications, the authors talk of a 'prominent morphological phenotype'. This may be the case but it is not obvious until after the data presented in Fig 2. The title of several sections would be more accurate saying axon formation instead of specification? Indeed, Fig S3 shows an accumulation of axon markers in the soma or changed AISs (although this last part is not quantified). Why is axon formation inhibited? This even occurs after CHX treatment transiently at E14.5. What is the leading cause? Is the ER also affected in this case? Is there a role for Filamin A in these neurons?

Fig S4h: eEF-2 and eIF4A1 expression – is the downregulation only in VZ progenitors or in neurons as well? If eIF4A1 is downregulated in the cKO compared to control in E18.5 neocortices, the fact that it is already downregulated in control at E14.5 could mean that this is less relevant for the fate changes discussed? Potentially the E18.5 result would concern a downregulation in neurons?

Fig 5 Co-cultures: translation rates appear higher in DIV1 immature neurons compared to DIV5. Are there markers at this stage to distinguish future FoxP2+ or Satb2+ neurons? This data seems missing to be able to correlate higher translation in E12 DIV1 immature neurons with a deep-layer fate.

Many times in the manuscript the authors refer to upper layer neurons (or deep layer neurons) – even when analysing primary cultures. Since position in the cortex is not analysed in these experiments, this seems misleading and it would be better to be more precise – e.g. mentioning the timepoints (E12 or E14 derived cells) from which cells are derived, or their expression of markers (e.g. Satb2+) when known. Indeed, in Fig 6d, it appears that many E12 derived DIV5 cells are Satb2+ and not all E14 derived DIV5 cells are Satb2+? What are the remaining cells in the E14 DIV5 cultures if they are neither Satb2+ nor CTIP2+? Also in Fig 6c, we are told that white arrowheads indicate neurons expressing high levels of Satb2, but this is not obvious for the EGFP+ cells.

Fig 7: CHX is used for 24 h to generate transient attenuation of translation inhibition, however in Fig 5 experiments it is clearly very active after only 240 min. Why did the authors choose 24 h for their experiments and what was the state of the cells after 24 h? Presumably this treatment greatly affected proliferation?

Fig 8: It is unclear why CTIP2 reporter expression increases in relation with lower expression of eIF4A1? The authors talk of ribosome preference redistribution – how can they be sure of this? It would be good to understand why CTIP2 expression goes up.

Line 655-658: 'During cortical development, Ire1 α supervises the cellular translation flux (Fig. 4) by driving the expression of translation regulators and RNA binding proteins, essential for the high protein synthesis rate in upper layer progenitors, enabling translation of Satb2 in their postmitotic progeny (Fig. 3).' Why upper layer progenitors? Ire1 α is highly expressed from E12.5 and translation rates are still high in E12.5 progenitors.

Line 663: 'Loss of Ire1 α leads to a decreased number of translation sites (Fig. 4k) and slower translocating ribosomes (Fig. 4h), which is associated with a global reduction of HPG incorporation, indicative of lower rate of protein synthesis.' If this is the case, how can it be explained that Ire1 α activation is paralleled by suppression of general translation (ref 35) (line 265)? It would be good to discuss this point?

Line 670: 'In line with this are our observations on decreased translation rates upon eIF4A1 KO (Fig. 4c and 4d) associated with defects in upper layer neurogenesis (Fig. 3g and 3h)'. The eIF4A1

KO work was performed in E12.5 progenitors (+ 1 DIV) – are we talking about upper layer neurogenesis?

Minor

Fig 1 Dose-dependent effect of APY69 is a bit weak with just two doses.

Fig 1 legend: plasmids encoding EGFP or EGFP and Cre simultaneously, please provide more information on the origin and nature of the Cre plasmids (NeuroD1 included). EGFPiCre: is there an IRES signal? Are the same number of plasmids electroporated in control and mutant conditions?

Fig 2c Where were these neurons and are they expected to already have finished migration? Is this a phenomenon which occurs during migration?

Fig 2e 'Disruption of bipolar morphology and aberrant laminar positioning within the CP was independent of Satb2 expression (Fig 2e)'. The phrasing is unclear to me since the vast majority of the neurons were Satb2 positive. Why is there little fate change when causing the mutation at E14? Are there also axonal defects in CTIP2+ neurons? This is not clearly stated. Are phenotypes for Satb2+ neurons more severe in a particular cortical layer?

Fig S3: Were the analyses performed from mixed morphologies? Why more AIS?

Fig 3d: Enrichment plots need further explanation

Line 299: instead of upper layer progenitors, say E14.5 electroporated progenitors did not affect...? There seem to be many cells between 25-75% CP in S4g?

How was the eIF4A1 KO by Crispr-Cas9 verified?

Line 304-305 – this is too strongly written since not all individual gene manipulations are significant.

Line 320 – is this really true (c.f. NeuroD1 data)?

Fig 4j – control merge appears identical to EGFP (green not yellow).

Lines 436 – 440 – it's unclear to me why it's necessary to say 'e' and not 'E' – why complicate? It should just be made clear in the figures that these are WT experiments?

Line 445 – should be 5d?

Line 447 – lower HPG incorporation at DIV5, same point then repeated on line 450?

Fig S6 – why inconspicuous?

Line 585 – 'neuronal fate determinants' - don't the authors mean more specifically Satb2-fate ?

eIF4A1 expression – does it change over corticogenesis?

Mechanism involving eEF-2 and eIF4A1 downstream of Ire1α – should it be in interaction with?

Surprising finding that different protein synthesis rates are intrinsic features of distinct progenitor and differentiated neuron lineages – was this already implied previously by the single cell RNA sequencing analyses across corticogenesis? Is it indeed surprising?

Line 658: was it shown here that Ire1α drives the expression of RNA binding proteins in general?

Line 676: 'The Ire1α cKO partially mimics the Satb2 KO (8), with the loss of upper layer type neurons at the expense of CTIP2-expressing deeper layer cells' – wouldn't it be to the benefit of CTIP cells not at their expense?

Line 681: 'Such progressive restriction of neuronal progenitor potency is abolished' – is it abolished, or is it just that CTIP2 is increasingly expressed?

Abstract: Is there really a unique sensitivity of upper layer fate to translation rates?

Reviewer #2 (Remarks to the Author):

The manuscript from Ambrozkiwicz et al., describes the functions of Ire1a in regulating the polarity of cortical neurons and neuronal identity presumably under the control of Stab2. The authors find Ire1a as a regulator of stab2 in a pharmacological screening performed with in vitro cultured cortical cells. Then they go ahead and characterize the functions of Ire1a during cortical differentiation using elegant transgenic and shRNA approaches. They explore several mechanisms and utilize many different experimental approaches.

This manuscript contains interesting data and accumulates results from an incredible amount of experimental work. It contains beautiful images and figures. However, the manuscript is complex, not focused, and not all the conclusions are fully supported. Particularly, the role of Ire1a in regulating neuronal identity is preliminary and possibly the data could offer other interpretations. An alternative interpretation is that the phenotypes could be reflecting changes in the time of transition from early fates to late phases, rather than changes per se. The role in neuronal polarity and migration is more clear, and although it has been explored by a previous group, it has not been investigated as deep and as in this manuscript.

The work has great potential, but in its present form, I consider lacks focus and is not conclusive. I suggest rewriting with a sharper focus on migration, not necessarily including all experiments, and revising the conclusions.

Major issues:

Issues about the role of Ire1a in regulating identity:

The argument that the expression of Satb2 defines a particular neuronal identity is oversimplifying. The authors consider Satb2 as a determinant of the identity of the intra-telencephalic neurons and callosal neurons. This is an outdated concept that simplifies the literature. Satb2 is expressed in neurons of all layers and Satb2 positive cells are born during a protracted period of development that includes the time early-born neurons are generated (Paolino et al., Proc Natl Acad Sci U S A. 2020). If there is no confusion, the authors themselves show expression in early-born deep layer neurons, although they do not clarify this in the text. In early and late-born neurons Stab2 regulates different transcriptional networks. It is not only involved in callosals in upper layers but also in regulating subcortical branching.

Thus, it should be best to revise the introduction and describe work posterior to 2008, such as work from McConnell, Chen, Suarez, Cremisi, and other labs, that clarifies the functions and expression of Stab2. It is increasingly clear that neurons differentiate along postmitotic life and that neuronal subtypes are best defined by subsets of transcription factors.

The data shows that Ire1a regulates the expression of Satb2 and Ctip2 in some early-born neurons (not in all by looking to cortical sections of the mutants), in late E14.5 neurons the effects of Ire1a are minor. Thus it cannot be assumed that these losses of Satb2 and gain of Ctip2 represent a switch of upper layer fate to early fates. Many WT early-born neurons express Satb2 or Ctip2 only. I would not agree that the data in the manuscript informs about the control of upper layer identity. Moreover, the results with the Emx-Satb2 do not support the conclusion that Ire1a is required to induce Sab2 and repress Ctip2. This result is observed only when BrdU cells are analyzed but it seems to reflect asynchrony in the generation of successive waves. The images show clearly that there is not a marked reduction in the total number of Satb2 cells or a marked increase in Ctip2 in the KO. Some of these interpretations could be related to the screening. The screening is very elegant and good but the number of Satb2 cells within the selected gates in controls is not very high, which raises the possibility that Ire1a regulates levels of Stab2.

Issues about the role in polarity and migration:

I consider that a role in neuronal polarity and migration is best supported by the data. However, there are a lot of confounding arguments in the manuscript. The role of Ire1a as a regulator of the rates of translation during cortical differentiation is well supported and interesting but it is mixing

the regulation of translation of Satb2 and Ctip2 mRNAs with defects in neuronal polarity and migration makes everything unnecessarily complex. Other issues:

1-It is not fully clear to me that the phenotypes of aberrant polarity and aberrant migration are not equally observed in electroporations targeting early-born and late-born neurons. If the expression of ire1a is higher in early precursors, the effects of knockin-out ire1a on early-born E12-E13 neurons should be more marked than the effects on e14 neurons.

2- If possible, clarify if cells with the abnormal polarity in reconstructed neurons are the ones that fail to migrate.

3-Represent the data from individual mice separately and analyze statistical differences between individual animals within conditions, do not pull all neurons from all animals for each condition.

5-In the western blots, it is unclear if the controls and bands are from the same gels. In Figure S4a, it seems that the bands are from different gels. If so, please make it obvious and show control bands for each of them.

6-In figure 6, it is shown that late progenitors have a higher translation rate than early progenitors, but it is my understanding that the roles in Ire1a dependent translation that are investigated by this work are in neurons. There is not sufficient data supporting the role specifically in precursors. The in vivo data shown in Figure 8 supports very well a role in postmitotic neurons.

7- Images showing Satb2 expression in vitro cells show very low expression levels. In figure 7, there are very few red cells and the morphology is not purely neuronal. Please revise.

8-Have the authors attempted to overexpress ire1a at later stages and analyze the translation rates?

9- Increased translation rate is a hallmark of early-born, deeper layer postmitotic neurons in the developing cortex and then the reverse in postmitotic neurons, but it is unclear to me how this occurs and if it is related to ire1a.

10-the authors conclude that the loss of Ire1 α results in suppression of translation rates as an effect of slower elongating ribosomes and decreased translation sites, but the effects are the same at E12 or E14. It is unclear to me if it should be expected that the data is not the same in the two different stages.

11-There are some editing mistakes, for example in the intro: is required to from the corpus callosum.

Reviewer #3 (Remarks to the Author):

In this manuscript Ambrozkiwicz and colleagues identify, using a small molecule screen, Ire1 α as an essential determinant of SatB2+ upper layer neuron identity. The authors then go on to characterize the expression pattern of Ire1 α and the mechanism by which it drives cortical upper layer neurogenesis. The experiments presented suggest that Ire1 α is a determinant of global translation rates by dynamically interacting with the ribosome and regulating the levels of eIF4A1 and eEF-2 expression. The global decrease in translation appears to be caused by increased ribosome stalling and reduced numbers of translation initiation sites. Intriguingly, as the levels of alternatively spliced Xbp1, a hallmark of canonical UPR, do not change, a non-canonical UPR-independent role of Ire1 α in the regulation of polysome levels is proposed. The authors put forward the idea that whilst eEF-2 is involved in cortical lamination, eIF4A1 regulates acquisition of upper layer identity downstream of Ire1 α in a mechanism of translational control dependent on 5'UTR-embedded structural elements in the Ctip2 and SatB2 mRNAs. The finding that developmental regulation of ribosome dynamics may be responsible for establishment of neuronal diversity and cortical layering is interesting and novel, as the developmental neuroscience field so far has focused predominantly on genomic and transcriptional events. The authors present a compelling body of work and the quality of the experiments is high. However, owing to the large amount of data presented and the non-uniform colour scheme in the figures and labels, the text is often very difficult to follow, especially for a reader not well acquainted with developmental neuroscience. We provide questions and suggestions for improvement below.

MAJOR

1. The manuscript starts with the observation that the Ire1 α inhibitor APY69 reduces Satb2 expression. The authors should specify (in the main text and with new supplementary figures) how many and which inhibitors were tested and how APY69 compares to all of them.
2. Fig. 3c (polysome RNA-seq):
 - Line 286-288: "Given no gross changes on the RNA level (Fig. 3b), we conclude that the expression level changes for these factors are of translational nature". Genes that are regulated at the translational level should come up in their polysome RNA-seq differential expression analysis. Therefore, to back up the claim in line 286-288, the authors should show where those factors lay on the volcano plot in fig.3c. Additionally, the authors should clarify why they think the regulation of a post-translational modification (like eEF2 phosphorylation) is "of translational nature".
 - Considering the observation in fig.3a (=increased polysomes in the cKO), one would expect that RNA sequenced from the polysome fractions of the cKO would be more than in the CTR. In this case, the volcano plot should be skewed towards the cKO. Instead, from fig.3c one would infer that the two distributions are equally centered. Is this the case? The authors should expand on this and explain better their analysis pipeline in the methods section.
 - Genes displaying a significant fold-change have different colors. It's not clear why. They should explain this in the figure legend.
3. Fig. 3i-j: The authors claim the interaction of Ire1 α with the small but not with the large ribosomal subunit, based only on RPS6 and RPL7. The authors should test other RPS and RPL to confirm that the observation is subunit-specific, and not protein-specific.
4. The authors could comment on their interpretation as to why they observe a phenotype for eIF4A1, but not of any other members of the initiation complex (fig.3-S4). In particular, the authors should also test eIF4A2 levels. It is in fact known (Williams-Hill et al. 1996) that eIF4A1 mRNA is translated less efficiently in post-mitotic cells, while eIF4A2 mRNA show the opposite trend. As progenitors differentiate into post-mitotic neurons, the authors should test whether the observed reduction in eIF4A1 is accompanied by an increase in eIF4A2 levels.
5. To better support their claim of a UPR-independent mechanism, the authors should expand on their observation, which is based exclusively on spliced Xbp1 levels, and systematically check in their omics data all the known UPR-target genes and by WB at least some more candidates (like Atf4, ...).
6. Line 452-453: "Altogether, these data indicate that deep layer-fated neurons have inherently higher translation rates compared to upper layer neurons". Increased translation in deep layer neurons may not be an intrinsic feature of upper layer neurons, rather a developmental artefact in their in vitro experiments, as the two populations will be doing different things at different times. To support their claims the authors should measure protein synthesis in Ctip2+ and Satb2+ neurons in intact tissues of different developmental stages. We suggest an ex vivo puromycylation assay (e.g. Biever et al 2019) or a FUNCAT assay on brain slices across developmental time-points. The authors could then perform immunostaining against Puro, Ctip2 and Satb2 and quantify the protein synthesis signal in each cell type.
7. The authors could comment on why CTIP2 neurons don't show a translational dependent phenotype in fig.7, when at E12 the treatment is applied. Perhaps if they anticipated the time of treatment, CHX would also affect Ctip2 identity specification. With regards to the effect that CHX has on Satb2 cell identity, the authors should clarify what is the alternative fate gained, as it is not Ctip2 according to their data. The authors should perform staining for the main neuronal identity markers (ie. Fezf2, Cux1, Cux2, FoxP2 and Tbr1) to tackle this point.
8. The model of differential 5'UTR recruitment is interesting. However, it needs further validation. For example, if the model is correct, one should see CTIP2 and Satb2 mRNA differentially regulated in their polysome RNAseq data set. Is that the case? Additionally, the authors should cluster the genes detected in their polysome RNAseq data set according to G-quadruplex occurrence and test whether they are differentially regulated.
9. Line 293: the authors should comment on whether the effect of eEF2 is only on the distribution of neurons across the cortical wall or it also increases neurogenesis, as suggested by the overall increase in EGFP+ cells seen in figure S4e.
10. Fig. S4: Nickases are more likely to result in small in frame indels. The authors must show that they effectively KO the gene by immunostaining.
11. Line 316-320: The authors should rephrase this paragraph to tone down their conclusions, the experiments are not so black and white.
12. The authors make the claim that eIF4A1 and eEF-2 have very distinct roles, controlling upper

layer specification and lamination, respectively. However, from their data (Fig. S4) it seems as though eEF-2 might also affect progenitor divisions and neuron production. The authors should perform a pH3 stain in CTRL and eEF-2 OE samples to address this.

13. The authors should use color palettes more consistently across (similar) experiments and prefer color-blind friendly tones.

MINOR

14. Fig.1b: the levels are normalized over DMSO for 0.2 μ M samples, but over the untreated for 1 μ M samples. The authors should use the same control across conditions.

15. Fig.1b: the authors should shortly describe how they calculate the "relative Satb2tdTom fraction". Is it tdTomato+ (Q2+Q4) / EGFP+ (Q1+Q2) and then further normalized to DMSO (or Untreated)? Citing in the methods section ref.26 is not enough.

16. Fig.1d: the authors should show where VZ is in the zoom out

17. Fig. S2: from the images it seems that there are more CTIP2+ cells (especially in the deeper layers) in the cKO than in the CTR. This is not reflected by the quantification. The authors should clarify this.

18. Fig. 1e: In the images shown the transfection efficiency is very different and this could explain the skew in the distribution of the neurons seen. More similar images should be shown the new panel should include single channel images of the EGFP signal, so that reader can better appreciate the increased branching.

19. Fig. 1g should include representative images.

20. Fig. 4h: The western blot and the quantifications in 4h do not mirror each other. In the blot it seems that the decay curve between Cre and control are different.

Suggested Changes to text.

21. Line 113: the authors should add a line to explain the aim of the BrdU experiment (something like "to label proliferating cells").

22. Line 118: the authors should spell out cortical plate "CP"

23. Line 184: "This contrasted with the majority of control neurons characterized with a single TP and lack of additional neurites emanating from the soma at this stage (Fig. 2d)." Perhaps the authors were referring to Fig. 2f instead of 2d? Or are they using "bipolar neurons" as a proxy for the described traits?

24. Fig. S4e: eIF4A1 KO has the wrong color palette (green instead of yellow?)

25. Fig.3i legend: missing explanation for "H" (compared to "IP")

26. The "y-axes" of Fig.3e and 3i look misplaced/unclear. They authors should clarify

27. Line 447: The sentence is a bit misleading. Maybe it's worth editing it to something like "We detected higher HPG incorporation rates in e14 than e12 cells also at DIV5."

28. Figure 1g should include representative images

29. In the text Figures S1D, E, F are not referenced in the order they appear in the figures

30. Line 651-2: sentence is obscure. Authors should rephrase it.

31. Line 35: typo "determine", not "determines"

32. Line 112: "of THE dorsal telencephalon"

33. Line 289: authors should cite fig. S4a-b.

34. Line 296: "to THE developing cortex"

35. Line 740: typo "by in part".

Rebuttal Letter, Borisova et al., "Protein translation rate determines neocortical neuron fate".

Reviewer Comments (**Our responses in bold**).

Reviewer #1 (Remarks to the Author):

In this paper the authors sought upstream regulators of *Satb2*, a marker of upper layer neurons, using a library of small molecules. They were able to identify Inositol-Requiring Enzyme 1 α (Ire1 α), also known as ER-to-Nucleus Signaling 1 (Ern1), the main sensor of ER lumen homeostasis and regulator of the Unfolded Protein Response (UPR), involved in these processes. They reveal that the development of *Satb2*⁺ neurons requires high rates of protein translation. Translational control is aided by 5'UTR-embedded structural elements in fate determinant genes (e.g. *Satb2*). These post-translational mechanisms hence contribute to neuronal diversity during corticogenesis. There is important and novel data in this manuscript. It is interesting that pathways known to be involved in cellular stress signalling can also contribute to correct cortical layer formation, as revealed here.

We are told that Ire1 α is a bimodal transmembrane kinase and RNase. Additionally, Ire1 α regulates stress-independent remodeling of actin filaments by its association with filamin A. There has also been shown a direct interaction of Ire1 α with ribosomes in vitro. This latter function, described here as non-canonical, is the focus of this work. Can the authors more definitively rule out that the other possible functions of Ire1 α are involved in fate determination?

We thank the Reviewer for raising this important point. We now provide new data on the expression pattern of Ire1 α and its activity mark, S724-P form in the developing cortex (Fig. 5j). Additionally, we show that this phosphorylation is virtually gone in the cortex of Ire1 α cKO animals (Fig. 5k-5m). Importantly, when we overexpress a S724-P-deficient form of IRE1 α (Fig. S3k and S3l) in E12.5 embryonic cortex, we do not observe the fate phenotype noted after a wild-type IRE1 α overexpression (S5p and S5q). This indicates that enzymatic activity associated with S724-P of Ire1 α is indispensable for fate specification. Moreover, we now provide evidence that overexpressing IRE1 α splicing client *Xbp1S* in E12.5 progenitors does not influence the neuronal progeny type (Fig. S3k-S3l).

The indications for a UPR-independent role of Ire1 α in the regulation of ribosome function and polysome level are rather weakly covered and this ought to be strengthened. Are all other potential functions of Ire1 α ruled out?

We thank the reviewer for raising this point. We now demonstrate that UPR activation in the cortex globally affects neuronal differentiation, abrogating the

cortical plate entry (Fig. S11e-S11g), rather than specifically affecting *Satb2* lineage. Moreover, we show that UPR markers as well as JNK2 and its phosphorylated form are not regulated in the *Ire1α* cKO cortex (Fig. S11b-S11d). Importantly, we show that mimicking the UPR by overexpressing Xbp1S using IUE (Fig. S3k and S3l), or by genetic replacement of eIF2α in vivo (Fig. S11e-S11g) does not specifically affect *Satb2* lineage. Regarding the ribosome function, we provide the data using the ribopuromycylation assay on ribosome stalling in the cKO neurons (Fig. 7a and 7b). Finally, to quantify ribosome function directly in the cortex, we generated a MetRS* X *Ire1α* cKO mouse line and show diminished translation rates upon disruption of *Ire1α* (Fig. S10a-S10c).

Ire1α cKO removes the kinase-extension nuclease (KEN) domain including the RNase active site (exon 20-21.) There is also a truncated protein still remaining, not shown specifically in this manuscript until Fig S4, and also not well explained. Is the truncated protein still expected to have functions?

We rephrased the description of the mutant (page 7, line 204-210). We now show that in our cKO, the overall level of *Ire1α* is reduced (Fig 5k-5l), as early as E12.5 (Fig.7g-7j), as well as in the *Ire1α^{fl/fl}* murine embryonic fibroblast model, infected with AAVs (Fig. S3a-S3b). More importantly, we show that as early as E12.5, the S724 phosphorylation of the enzyme is lost, as compared to control cortex. Additionally, we now show that silencing of the endogenous *Ire1α* using siRNA engenders a similar fate switch tendency as the genetic disruption of *Ire1α*, we describe (Fig. S3g-S3h).

There is a dynamic interaction of *Ire1α* with the ribosome and regulation of eIF4A1 and eEF-2 expression. In one of the last conclusion sentences in the discussion the authors say: 'Given developmental downregulation of *Ire1α*, it is plausible that other translational regulators alongside *Ire1α* drive the increased translation efficiency in upper layer progenitors.' This sentence emphasizes a major timing question not well addressed in the manuscript. Throughout, the emphasis is on the role of *Ire1α* for *Satb2* expression strongly associated with the production of upper layer neurons. However, is *Ire1α* also important for the earlier-born deep layer neurons with different fates (ref 23)? When it acts in early progenitors, is it just influencing the number of early-born *Satb2*+ neurons produced or other early-born neurons as well? Or is it mainly acting early to later influence *Satb2*+ neuron production? Highest levels of *Ire1α* are at E12.5 and upper layer neurons are produced mainly later – why is it required so early? There seem to be less effects on E14.5 born cells (Fig S2).

We have addressed the timing aspect by a series of BrdU experiments in the cKO, as well as generating a new cell cycle reporter cell line Fucci2aR X *Ire1α* cKO (Fig. S7). Moreover, we heavily edited our conclusions and rephrased the description of the phenotypes, given the expression of *Satb2* also in neurons of deeper cortical layers. Finally, in our rescue experiments with the translational

reporter, we can show that the regulation of *Satb2* by *Ire1α* takes place in E12.5 cortical progenitors as well (Fig. 9n-9p).

Urrea and colleagues show an impact on deep layer neurons with a reduced thickness of the *Tbr1* layer. This is not mentioned sufficiently that we understand layering issues when studying Figs 1 and 2. Is the effect on *Tbr1* solely a lamination problem?

We have addressed the *Tbr1* fate in our mutant using BrdU pulse experiments (Fig. S7o and S7p), as well as using IUE (Fig. S3i-S3j).

At E18.5 *Ire1α* is beginning to be downregulated but we still see polysome accumulation in mutant cortices at this age. Polysomes are deregulated in which cell type? It seems important to determine this to know how much this phenotype is relevant.

We now show that in the cKO, the overall expression of *Ire1α* is reduced as early as E12.5 (Fig. 5k-5l and Fig. 7g-j). We also show that cKO neurons display more ribosome stalling than the controls in ribopuromycylation assay (Fig. 7a-7b). Polysome purifications require a great amount of starting input. Even though progenitor sorting from control and cKO cortex is possible, the amount of material we would need to carry out such experiment in a controlled environment (given that our pulldowns are performed in non CHX-treated conditions to avoid artifacts, and all sorting would have to be performed in 4degC, compare Kraushar et al., 2021, Mol Cell) greatly exceeds our capabilities.

Does changed fate influence final position, this is also not so clear? Further explanations are required to explain all these results and to justify and clarify that *Ire1α* is a determinant of 'upper layer fate'.

We apologise for not specifying this aspect more clearly. We now edited the manuscript to describe the role of *Ire1α* more precisely in regulating *Satb2*-positive neuronal lineage. To address this point of the Reviewer, however, we now present data on *Cux1*, *Tbr1* and *Sox5*, when *Ire1α* is disrupted in the early cortical progenitors using IUE (Fig. S3i-S3j). To answer the Reviewer's question about the positioning of the neurons within the cortical plate after *Ire1α* disruption, we quantified the that final position of *Satb2* and CTIP2 neurons in E12.5- and E14.5-derived lineage (Fig. S3e and S3f, S3o-S3q, S6a-S6d), as well as in P2 cKO (Fig. S7n).

This manuscript also raises the question of translation rates in progenitors over time and this is all very interesting. Was such data predicted from previous single cell data? In the mutant situation, were proliferation / cell cycle rates changed? How much might this influence the phenotypes and changed fate observed?

As we explain in the manuscript text, in our previous work (Harnett and Ambrozkiwicz et al., 2022, Nat Struct Mol Biol), we have shown that translation efficiency specifically for chromatin-binding proteins, including TFs, is up-regulated mid-gestation. In this work, we also demonstrate dynamic modulation of ribosome number in progenitor cells in the developing cortex. To further address the question of this reviewer, we now show data on the proliferation (Fig. S7s and S7t, Fig. 9o) and cell cycle rates (Fig. S7a-S7m, S7o-S7r and S7u-S7v) in our mutant. Congruent with our observations using NeuroD promoter-driven Cre electroporations (Fig. S3r-S3w), we show that *Ire1α* regulates cortical cell diversification in a progenitor-embedded mechanism linked to modulating the dynamics of cell cycle exit.

Axon specification / formation data seem to dilute the progenitor message.

We thank the reviewer for this comment. We restructured the entire manuscript quite dramatically during the revision process. We have now moved the data on axon specification completely into the supplement (Fig. S3m-q, S3r, S3t, S3u, S3w, S4-S6) and significantly reduced the text (page 8, line 242-page 9, line 285).

Further more specific points are as follows:

Fig 1 e /S1a / S1d: the branching phenotype is not obvious in the photos – which cells did the authors focus on? Were these cells mis-positioned? In general, it would be nice to know more about the final positions of the electroporated cells. Is their position normally related to their fate? Is their position related to their altered fate?

In the representations, we selected the cells from the cortical plate. We now provide detailed quantifications that the morphological phenotype equally affects *Satb2* and *CTIP2*-positive neurons (Fig. S3n). We also tested if the morphological phenotype severity was associated with their overall mis-positioning. In general, in our E12.5 electroporations, we did not detect any lamination defect (Fig. S3e and S3f). We also did not observe the morphologically aberrant cells to be specifically localizing to a given cortical bin (Fig. S3o-S3p). Interestingly, in our E14.5 electroporations (Fig. S4a-S4b), we observed that the cells with disrupted polarity did indeed localize to deeper cortical bins as bipolar neurons (Fig. S6a-S6d). In that sense, we believe that at E14.5, the fate of a neurons is intertwined with its morphology to regulate the final position of a cell within the cortical plate. This might be embedded in the transcriptional programs orchestrated by the fate marker, e.g. *Satb2* (Bessa et al., 2023, Biorxiv).

Cell position (S2): The graphs in d and h are not well explained. We are not told if there are BrdU +ve cells below the CP? In h the position of *Satb2*+ cells is changed although their numbers are unchanged, and this is not mentioned until the later section. How much does the impact on lower layers affect these results? Double-

labeled CTIP+ Satb2+ cells are not commented on? These seem like interesting points to further discuss. The fate change could also be emphasized since overall BrdU + cells are not changed in number.

We now present a classical binning graphs and normalized quantifications instead of the representation of individual cells (Fig. S7b-S7m). We did not observe BrdU-positive cells below the CP. Additionally, given the Reviewer's question on deeper layers, we now present the analysis of the generation CTIP2- and Tbr1-positive neurons in control and cKO cortices at E11.5 (Fig. S7o-S7p), and of cell cycle exit at E12.5 (Fig. S7q-S7r) and at E14.5 (S7u-S7v). We would like to point out, that we have generated a mouse line, crossing our *Emx1^{Cre/+}*, *Ire1α^{ff}* line with Rosa26-Fucci2aR reporter line (Mort et al., 2014, Cell Cycle) to visualize the dynamics of the cell cycle in E14.5 cortex (Fig. S7s and S7t). Regarding the BrdU-, Satb2- and CTIP2- triple positive neurons, we did not detect a significant difference in their number at P2 (Fig. S7b-S7c, S7h-S7i). We also did not detect a significant difference in the number of Satb2- and CTIP2- double positive neurons in E16.5 cortex after the electroporation at E12.5 (Fig. S3d).

Axon specification: before doing quantifications, the authors talk of a 'prominent morphological phenotype'. This may be the case but it is not obvious until after the data presented in Fig 2. The title of several sections would be more accurate saying axon formation instead of specification? Indeed, Fig S3 shows an accumulation of axon markers in the soma or changed AISs (although this last part is not quantified). Why is axon formation inhibited? This even occurs after CHX treatment transiently at E14.5. What is the leading cause? Is the ER also affected in this case? Is there a role for Filamin A in these neurons?

As we mentioned before in the rebuttal letter, in the light of the previous comment of this Reviewer, we have restructured the manuscript, and reduced the space for the axon formation defects, to focus the manuscript a bit more. We amended the titles of the sections proposed by the author (page 8, line 242; page 15, line 521). We find the question of the Reviewer on what the leading cause of disturbed axon formation quite tricky to answer. Let us explain why this is. Ire1α is an ER resident protein with a plethora of housekeeping functions. Its role in maintaining the cellular homeostasis is quite remarkable. As the Reviewer correctly points out, apart from its well-established role in stress-induced proteostasis, Ire1α has an important role in regulating cellular migration through anchoring of Filamin A (Urrea et al., 2018, Nat Cell Biol). Additionally, Ire1α regulates the mitochondria bioenergetics and ER receptor compositions (Carreras-Sureda et al., 2019, Nat Cell Biol). It is quite evident that Ire1α is critical for ER function also under the normal, non-stressed conditions. In turn, ER-mediated microtubule organization has been shown as a neuronal polarization principle. For this reason, we believe that ER disruption, linked to

***Ire1α* KO, is one of the main reasons for disruption of axon formation in our study. To further support this hypothesis, we now corroborate expected Filamin A mislocalization and higher expression level in the cKO (Fig. S6e), as well as disrupted ER morphology in transiently CHX treated neurons (Fig. S1p-S1q). Additionally, we now provide data on the loss of stabilized (acetylated) and dynamic (tyrosinated) microtubules in the soma and the longest neurite in DIV2 *Ire1α* KO neurons (Fig. S6f and S6g), fragmented and misaligned Golgi apparatus (Fig. S6h-S6l).**

Regarding the CHX-induced loss of axons, we believe that a contributing cause might be also loss of translation of polarity proteins. Axon formation has been found particularly sensitive to protein synthesis inhibition (Jareb and Banker, 1997, J Neurosci), likely due to enrichment of mTOR complexes in the axonal growth cone (Poulopoulos et al., 2019, Nature). It appears that the CHX treatment might be locally preventing the translation of proteins necessary for growth cone expansion.

While we think that in both cases, in our cKO and after CHX treatment, the ER and the protein synthesis pathways play an important role in regulating the morphology, we would also argue that loss of neuronal identity marker *Satb2* can be accountable for this phenotype to a certain extent as well (Bessa et al., 2023, Biorxiv). According to recent elegant work, the genetic programs of polarity were shown to be conserved for neuronal differentiation between early and late progenitor-derived lineages (Telley et al., 2019, Science).

Fig S4h: eEF-2 and eIF4A1 expression – is the downregulation only in VZ progenitors or in neurons as well? If eIF4A1 is downregulated in the cKO compared to control in E18.5 neocortices, the fact that it is already downregulated in control at E14.5 could mean that this is less relevant for the fate changes discussed? Potentially the E18.5 result would concern a downregulation in neurons?

We now include the data on the level of eIF4A1 and eEF-2 in E12.5 and E14.5 control and cKO cortical homogenates (Fig. 7g-7j), as well as developmental expression patterns for eIF4A1 and eEF-2 across the neurogenic stages of the developing cortex (Fig. S9m). Moreover, using FlashTag injections into the developing lateral ventricle, we labeled the apical progenitors using FACS to demonstrate progenitor-embedded expression of *Ire1α* and eIF4A1 (Fig. S9q-S9t). More importantly, we now also provide a functional rescue experiment in E12.5 *Ire1α* cKO progenitors using a wild-type and helicase-activity deficient eIF4A1 regarding the translational reporters of *Satb2* and CTIP2 (Fig. 9m-9r).

Fig 5 Co-cultures: translation rates appear higher in DIV1 immature neurons compared to DIV5. Are there markers at this stage to distinguish future FoxP2+ or *Satb2*+ neurons? This data seems missing to be able to correlate higher translation in E12 DIV1 immature neurons with a deep-layer fate.

To answer this request of the Reviewer, we have tried immunostaining in DIV1 nucleofected co-cultures for Satb2, CTIP2 and FoxP2. Overall, we saw a very weak labeling using CTIP2 and Foxp2. As we have shown in the first version of the manuscript, nucleofected cultures at DIV1 are mainly composed of postmitotic neurons (Fig. S1c). Additionally, we now show that at DIV1, the neurons derived from E14 cortex express Satb2, in contrast to neurons from E12 cortex (Fig. S1a-S1b).

Many times in the manuscript the authors refer to upper layer neurons (or deep layer neurons) – even when analysing primary cultures. Since position in the cortex is not analysed in these experiments, this seems misleading and it would be better to be more precise – e.g. mentioning the timepoints (E12 or E14 derived cells) from which cells are derived, or their expression of markers (e.g. Satb2+) when known.

We absolutely agree with the Reviewer here and we changed the phrasing across the entire manuscript.

Indeed, in Fig 6d, it appears that many E12 derived DIV5 cells are Satb2+ and not all E14 derived DIV5 cells are Satb2+? What are the remaining cells in the E14 DIV5 cultures if they are neither Satb2+ nor CTIP2+?

We now provide a new quantification that majority of these cells express NeuN, indicative of their neuronal differentiation (Fig. S2).

Also in Fig 6c, we are told that white arrowheads indicate neurons expressing high levels of Satb2, but this is not obvious for the EGFP+ cells.

This figure is now Fig. S2a. Here, we counted the proportion of cells expressing Satb2, regardless of the level (Fig. S2b). We removed the word "high", as it was misleading, as correctly noted by the Reviewer.

Fig 7: CHX is used for 24 h to generate transient attenuation of translation inhibition, however in Fig 5 experiments it is clearly very active after only 240 min. Why did the authors choose 24 h for their experiments and what was the state of the cells after 24 h? Presumably this treatment greatly affected proliferation?

We would like to explain the rationale. On Fig. 5 (now, Fig. 4), the CHX was only used to show the translational nature of HPG incorporation. As the Reviewer can also appreciate across this manuscript, CHX is an active inhibitor of translation also in very short time pulses (Fig. 3c, 10 minutes CHX incubation). Our reasoning for applying it for the entire 24 hours was the reported effects on polarity being in the range of 20-22 hours (Fig. 5 in Jareb, Banker, 1997, J Neurosci). Here, it is important to note, that we replace the medium after 24 hours and the cell has a chance to restore its translation, but it is the critical

period right after plating that seems to be vital for *Satb2* expression and the axon (Fig. 1).

In fact, in our nucleofected cultures (Fig. S1c), as opposed to ex utero electroporation (Fig. 2a), we observe a small population of Ki67- and Pax6-positive neurons at DIV1. To address the Reviewer comment, we fixed the DMSO and CHX-treated neurons at the end of treatment (24 hours post-plating) and stained the cultures for mitotic markers Ki67 and PHH3. In line with our previous data, overall, the proportion of PHH3 positive cells at DIV1 after nucleofection is small. We show, however, that at DIV1, the CHX does not change the proportion of PHH3 positive cells in our co-cultures. We demonstrate a strong effect on Ki67 expression (Fig. S1g). This indicates a likely prolonged cell cycle exit of these treated cultures. That said, after changing the medium to normal and maintaining the treated cells until DIV5, we show a full differentiation into neurons, as quantified by the expression of NeuN, Tbr1 and Brn2 (Fig. S1h-S1k). We do not believe the defects can be ascribed to decreased time window to achieve differentiation, as the phenotype is quite strong regarding axon specification and expression of *Satb2*.

Fig 8: It is unclear why CTIP2 reporter expression increases in relation with lower expression of *eIF4A1*? The authors talk of ribosome preference redistribution – how can they be sure of this? It would be good to understand why CTIP2 expression goes up.

We thank the Reviewer for bringing this up. We now removed the "ribosome preference distribution" from our manuscript. We hypothesized that the loss of *Satb2* in *eIF4A1* KO (Fig. 7e-7f), and/or its lower expression levels, may be associated with enhanced CTIP2 expression in the reporter-targeted neurons. We now provide the new data on decreased *Satb2* expression level in *eIF4A1* KO neurons expressing the reporter (Fig. 9e-9f). We have tried performing RiboSeq from the developing cortices to address this question, however, due to technical issues, this experiment failed.

Line 655-658: 'During cortical development, *Ire1α* supervises the cellular translation flux (Fig. 4) by driving the expression of translation regulators and RNA binding proteins, essential for the high protein synthesis rate in upper layer progenitors, enabling translation of *Satb2* in their postmitotic progeny (Fig. 3).' Why upper layer progenitors? *Ire1α* is highly expressed from E12.5 and translation rates are still high in E12.5 progenitors.

In the new line 507, we replaced "upper layer" with "neuronal".

Line 663: 'Loss of *Ire1α* leads to a decreased number of translation sites (Fig. 4k) and slower translocating ribosomes (Fig. 4h), which is associated with a global reduction of HPG incorporation, indicative of lower rate of protein synthesis.' If this is the case,

how can it be explained that Ire1a activation is paralleled by suppression of general translation (ref 35) (line 265)? It would be good to discuss this point?

The Reviewer is correct in stating that the activation Ire1 α during the stress response, involving the activation of PERK and phosphorylation of Ser52 on eIF2 α leads to global shutdown of translation in the cell (Sonenberg and Hinnebusch, 2009, Cell). We show new data that inactivation of Ire1 α does not affect eIF2 α and its phosphorylation status (Fig. S9a-S9b), as well as other UPR effectors (Fig. S11b-S11c), indicative of a different molecular effectors leading to translational silencing. Additionally, we now show that mimicking UPR using an in utero electroporation-based strategy to replace endogenous *eIF2 α* with phospho-deficient (S52A) or phospho-mimic (S52D) eIF2 α , leads to severe cortical plate entry defects associated with decreased expression of postmitotic markers of neuronal identity (Fig. S11e-S11g; page 12, lines 422-436). These data indicate that eIF2 α and its phosphorylation regulate neuronal differentiation per se, rather than intricate neuronal cell fate acquisition.

Our manuscript reveals another non-canonical role of Ire1 α , which is shown to be positioned at the translocon and interacts with 80S ribosomes (Acosta-Alvear, 2018, eLife), primed to safeguard cytoplasmic translation in a non-stressed condition. We show that in this process, Ire1 α uses a different set of cellular translation regulation, as compared to the ones it co-acts with during UPR (page 14, lines 482-489).

Line 670: 'In line with this are our observations on decreased translation rates upon eIF4A1 KO (Fig. 4c and 4d) associated with defects in upper layer neurogenesis (Fig. 3g and 3h)'. The eIF4A1 KO work was performed in E12.5 progenitors (+ 1 DIV) – are we talking about upper layer neurogenesis?

In now line 522, we replaced “upper layer neurogenesis” with “Satb2-positive neurons”.

Minor (*if no comment, then edits*).

We thank the Reviewer for all the edits. We would like to mention that we have introduced all the suggested edits to the text, however, also heavily restructured the whole manuscript, to improve its flow and focus.

Fig 1 Dose-dependent effect of APY69 is a bit weak with just two doses.

We removed “dose-dependent”.

Fig 1 legend: plasmids encoding EGFP or EGFP and Cre simultaneously, please provide more information on the origin and nature of the Cre plasmids (NeuroD1 included). EGFPiCre: is there an IRES signal? Are the same number of plasmids electroporated in control and mutant conditions?

We have included the information on plasmids and their origin in the Supplementary Table S5. In all of our experiments, the number of plasmids used for electroporations is the same, importantly, the final concentration and the injected amount of DNA into the later ventricle is also the same. Regarding the strategy for NeuroD1-driven KO, please refer to the figure legend. To induce a KO in *Ire1a^{fl}*, we used EGFPiCre plasmid, to simultaneously express EGFP and Cre, using bicistronic construct with IRES sequence. In this plasmid, Cre sequence is modified to reduce the high CpG content of the prokaryotic coding sequence (improved Cre, iCre) to reduce a chance for epigenetic silencing. As a control, we used a plasmid expressing EGFP. We have used this strategy in our previously published works (Ambrozkiewicz et al., 2018, *Neuron*; Ambrozkiewicz et al., 2021, *Mol Psychiatry*). We added the requested information in page 33, line 1343).

Fig 2c Where were these neurons and are they expected to already have finished migration? Is this a phenomenon which occurs during migration?

The neurons in this figure were present in the cortical plate, in the layer II/III. Here, we illustrate the example of the morphological phenotype using expansion microscopy, to compare the morphologies on neurons of the same type. Given the nature of the sample preparation, we unfortunately could not show the expanded section overview, due to its dimension (we achieve 4X expansion using the standard protocols, Gao et al., *ACS Nano*, 2018).

Fig 2e 'Disruption of bipolar morphology and aberrant laminar positioning within the CP was independent of *Satb2* expression (Fig 2e)'. The phrasing is unclear to me since the vast majority of the neurons were *Satb2* positive. Why is there little fate change when causing the mutation at E14? Are there also axonal defects in CTIP2+ neurons? This is not clearly stated. Are phenotypes for *Satb2*+ neurons more severe in a particular cortical layer?

We replaced this sentence, now in line 252 with "We noted no change of *Satb2* expression in control and KO neurons derived from E14.5 progenitors (Fig. S4e)". As we have explained before, we now added new data on the morphology and show that both *Satb2*- and CTIP2-expressing neurons were affected to a similar extent (Fig. S3n). Regarding the severity of morphological phenotype and cell positioning, we have addressed this with new experiments on Fig. S3e, S3m-S3q, S6a-S6d). As explained in our reply above, we specifically see the correlation between the severity of morphological phenotype and the position of the neuron within the cortical plate in E14.5 progenitor-derived cortical lineage.

Regarding the question "why there little *Satb2* fate change..." when we induce the KO at E14.5, please refer to Fig. 9 and Fig. S12. We demonstrate a

critical requirement of translational control in early, but not in late progenitors. We believe that Ire1 α -mediated regulation of Satb2-expressing neurons happens in multipotent early progenitors, able to generate all cortical neuron types. E14 progenitors exhibit reduced potency in specification of neuronal fate, and according to our findings (Fig. 9 and S12), the regulation Ire1 α is restricted to the earlier progenitor type. It may be linked to a developmental downregulation of Ire1a (Fig. 5h-5j).

Fig S3: Were the analyses performed from mixed morphologies? Why more AIS?

The Reviewer is referring to the electrophysiological measurements now presented on Fig. S5. These recordings were performed from an autaptic cultures (compare Ripamonti et al., 2017, eLife). Unfortunately, the polarity analysis in such cultures is impossible, given the robust connections each neuron makes on itself. Given our findings on increased proportion of neurons with multiple Tau-1 processes (Fig. S4g-S4h), we believe the increased number of AIS is a direct consequence of disrupted polarity.

Fig 3d: Enrichment plots need further explanation

We thank the Reviewer for the suggestion. We now present a bulk of new data that cover some of the milestones of cortical development. Due to editorial policy, we unfortunately cannot explain every graph to the level of detail that we would prefer. We hope that GSEA is now a standard procedure used for mining of omics data. Given new analysis we performed for the polysomes, we rephrased the paragraph (page 10, line 331-335).

Line 299: instead of upper layer progenitors, say E14.5 electroporated progenitors did not affect...? There seem to be many cells between 25-75% CP in S4g?

We edited the sentence as requested.

How was the eIF4A1 KO by Crispr-Cas9 verified?

As outlined in the Table S5, we used an Addgene plasmid, validated and used before (Ochiai, 2020, Sci Adv). We also now provide a NIH3T3 transfection based validation of eIF4A1 KO using western blotting (Fig. S9k-S9l).

Line 304-305 – this is too strongly written since not all individual gene manipulations are significant.

We softened the conclusion here (now Fig. S9i-S9j). We would like to point out that given comparison of categorical datasets in this experiment, we used Chi-square test. This statistical non-parametrical test is used to examine the

differences between categorized variables and permits evaluation of multiple groups (Ambrozkiwicz et al., 2018). In our case, we show that there is a relationship between the dosage of our tested genes and the number of axon specified by a single neuron.

Line 320 – is this really true (c.f. NeuroD1 data)?

We removed this sentence.

Fig 4j – control merge appears identical to EGFP (green not yellow).

We have corrected this figure (now Fig. 8j).

Lines 436 – 440 – it's unclear to me why it's necessary to say 'e' and not 'E' – why complicate? It should just be made clear in the figures that these are WT experiments?

We now stick to "E" across the entire manuscript.

Line 445 – should be 5d?

Thank you for pointing out the typo.

Line 447 – lower HPG incorporation at DIV5, same point then repeated on line 450?

Of course. Thank you for pointing this out.

Fig S6 – why inconspicuous?

We edited the title of the figure.

Line 585 – 'neuronal fate determinants' - don't the authors mean more specifically Satb2-fate?

We edited this sentence according to the Reviewers suggestion.

eIF4A1 expression – does it change over corticogenesis?

We now provide the developmental expression pattern of eIF4A1 throughout corticogenesis (Fig. S9m) as well as progenitor expression of eIF4A1 using Western blotting in sorted apical progenitors (Fig. S9r-S9t).

Mechanism involving eEF-2 and eIF4A1 downstream of Ire1 α – should it be in interaction with?

In our work, we do not show the interaction of Ire1 and eIF4A1 or eEF-2, but a translational mechanism controlling the level of expression of both proteins.

Surprising finding that different protein synthesis rates are intrinsic features of distinct progenitor and differentiated neuron lineages – was this already implied previously by the single cell RNA sequencing analyses across corticogenesis? Is it indeed surprising?

In our experiments, we quantify the translation rates to show that different pools of apical progenitors are characterized by different protein translation dynamics, which in our opinion could not be predicted using RNA sequencing analyses, relying on quantification of transcripts in unfractionated cellular lysates. We also believe this message is one of the main strengths of our work.

Line 658: was it shown here that Ire1 α drives the expression of RNA binding proteins in general?

We removed this part of the sentence.

Line 676: 'The Ire1 α cKO partially mimics the Satb2 KO (8), with the loss of upper layer type neurons at the expense of CTIP2-expressing deeper layer cells' – wouldn't it be to the benefit of CTIP cells not at their expense?

We edited this sentence (page 7, line 185).

Line 681: 'Such progressive restriction of neuronal progenitor potency is abolished' – is it abolished, or is it just that CTIP2 is increasingly expressed?

We removed this sentence.

Abstract: Is there really a unique sensitivity of upper layer fate to translation rates?

We changed this sentence.

Reviewer #2 (Remarks to the Author):

The manuscript from Ambrozkiwicz et al., describes the functions of Ire1a in regulating the polarity of cortical neurons and neuronal identity presumably under the control of Stab2. The authors find Ire1a as a regulator of stab2 in a pharmacological screening performed with in vitro cultured cortical cells. Then they go ahead and characterize the functions of Ire1a during cortical differentiation using elegant transgenic and shRNA approaches. They explore several mechanisms and utilize many different experimental approaches.

This manuscript contains interesting data and accumulates results from an incredible amount of experimental work. It contains beautiful images and figures. However, the manuscript is complex, not focused, and not all the conclusions are fully supported. Particularly, the role of Ire1a in regulating neuronal identity is preliminary and possibly the data could offer other interpretations. An alternative interpretation is that the phenotypes could be reflecting changes in the time of transition from early fates to late phases, rather than changes per se. The role in neuronal polarity and migration is more clear, and although it has been explored by a previous group, it has not been investigated as deep and as in this manuscript. The work has great potential, but in its present form, I consider lacks focus and is not conclusive. I suggest rewriting with a sharper focus on migration, not necessarily including all experiments, and revising the conclusions.

We thank the Reviewer for their skepticism about the previous version of our work. We made sure to reconceptualize the story and restructure the way we present the data. Specifically, we have done a lot of work to understand the role of Ire1 α in regulating the neuronal diversity in the cortex (compare new figures 5, S3, S7, S10, S11 and 9). Let us explain the results of the new experiments below in point-by-point responses. Regarding the comment of the Reviewer on the migration, we decided against focusing on this aspect, given the presumptive lack of novelty of such manuscript (as the Reviewer correctly points out, there is a previous report describing the role of Ire1 α in neuronal migration already, Urra et al., Nat Cell Biol, 2019),

We have received the feedback from the Reviewer 1, who states that the part on axon and migration dilutes our message. We would like to share with the Reviewer our opinion that this manuscript is a thorough characterization of the role of Ire1 α and its function in regulating protein translation in the developing cortex, as well as the new results supporting the new evidence that RNA transcripts are not principal regulators of protein abundance in the developing cortex. We believe that these data will change the way the scientists approach some main concepts in developmental neuroscience, specifically in times of single-cell RNA sequencing and classifying neuronal diversity based on abundance of RNA. We encourage the Reviewer to also compare with our previous work (Ambrozkiwicz et al., 2018, Neuron and Harnett and Ambrozkiwicz et al., 2022, Nat Struct Mol Biol).

The mentioned that our recent work is also the reason, why the current manuscript focuses on Satb2. Among many neuronal subtype marker proteins and fate determinants, Satb2 was the one, whose translation was sharply up-regulated mid-gestation (Harnett and Ambrozkiwicz et al., 2022, Nat Struct Mol Biol). We report that in the case of Satb2, and its mRNA and protein levels are not correlated across the development of the cortex.

We would like to point out that in the course of the revision, we have generated two new mouse lines (crossing in a translation reporter MetRS* (Alvarez-Castelao et al., Nat Biotechnol, 2017) into our *Ire1a^{fl/fl}* line to study translation rates in vivo (Fig. S10), as well as cell cycle reporter Fucci2aR (Mort et al., Cell Cycle, 2014), to visualize cell cycle dynamic in vivo as well (Fig. S7s-S7t).

Finally, we would like to propose, why we believe that the part of our manuscript on axon formation belongs to this paper, which addresses the mechanisms of establishing cell diversity in the developing neocortex. Differentiation programs have been shown to be evolutionarily conserved between cortical lineages (Telley et al., Science, 2019). Axon formation programs belong to differentiation programs. We also show in our recent work that loss of Satb2 in the cortex engenders axon aberrances (Bessa et al, 2023, biorxiv). It is therefore conceivable that loss or disruption of expression of neuronal identity proteins, such as Satb2, might cause phenotypes related to morphology of a neuron (page 15, lines 521-531).

Major issues:

Issues about the role of Ire1a in regulating identity: The argument that the expression of Satb2 defines a particular neuronal identity is oversimplifying. The authors consider Satb2 as a determinant of the identity of the intra-telencephalic neurons and callosal neurons. This is an outdated concept that simplifies the literature. Satb2 is expressed in neurons of all layers and Satb2 positive cells are born during a protracted period of development that includes the time early-born neurons are generated (Paolino et al., Proc Natl Acad Sci U S A. 2020). If there is no confusion, the authors themselves show expression in early-born deep layer neurons, although they do not clarify this in the text. In early and late-born neurons Satb2 regulates different transcriptional networks. It is not only involved in callosals in upper layers but also in regulating subcortical branching. Thus, it should be best to revise the introduction and describe work posterior to 2008, such as work from McConnell, Chen, Suarez, Cremisi, and other labs, that clarifies the functions and expression of Satb2. It is increasingly clear that neurons differentiate along postmitotic life and that neuronal subtypes are best defined by subsets of transcription factors.

We thank this Reviewer for making this point. We revised the fragment on Satb2 in the introduction. By no means did we mean to ignore the seminal works on Satb2. Writing our manuscript was quite the challenge, given the magnitude of

the experimental dataset. In the previous version of the manuscript, we have tried to include the information crucial to understanding our rationale. We apologize for overlooking this. We amended the introduction, including some of the works listed by the Reviewer. We focused the conclusions specifically on the Satb2 expression and avoid the faulty "upper layer" and "deeper layer" generalization. We fully agree with the Reviewer that the neuronal fate is defined by a combinatorial expression of transcription factors (page 3, line 7). Additionally, we now show new data using more identity proteins, like Cux1, Sox5 and Tbr1 (Fig. S3i-S3j).

The data shows that Ire1a regulates the expression of Satb2 and Ctip2 in some early-born neurons (not in all by looking to cortical sections of the mutants), in late E14.5 neurons the effects of Ire1a are minor. Thus it cannot be assumed that these losses of Satb2 and gain of Ctip2 represent a switch of upper layer fate to early fates. Many WT early-born neurons express Satb2 or Ctip2 only. I would not agree that the data in the manuscript informs about the control of upper layer identity.

We are very thankful for the insightful comment of this Reviewer. We now focused our conclusions specifically on Satb2 in cortical lineages.

Moreover, the results with the Emx-Satb2 do not support the conclusion that Ire1a is required to induce Sab2 and repress Ctip2. This result is observed only when BrdU cells are analyzed but it seems to reflect asynchrony in the generation of successive waves. The images show clearly that there is not a marked reduction in the total number of Satb2 cells or a marked increase in Ctip2 in the KO. Some of these interpretations could be related to the screening. The screening is very elegant and good but the number of Satb2 cells within the selected gates in controls is not very high, which raises the possibility that Ire1a regulates levels of Stab2.

The initial BrdU experiments were performed at P2. We now demonstrate more timepoints, focusing on the generation of CTIP2 in the cortex (S7o-S7v). We also show the rescue experiment using the translational reporter for Satb2 and CTIP in E12.5 cortical progenitors using wild-type and helicase activity-deficient eIF4A1 (Fig. 9). In fact, we measure loss of Satb2 in neurons showing higher efficiency for CTIP2 reporter (Fig. 9e-9f). Additionally, we present new data from the screening, showing decreased number of Satb2 and increased number of CTIP2 cells, when Ire1 α is inhibited (Fig. 5a-5g). We have removed the claim of repressive activity of Satb2 from the manuscript.

Issues about the role in polarity and migration:

I consider that a role in neuronal polarity and migration is best supported by the data. However, there are a lot of confounding arguments in the manuscript. The role of Ire1a as a regulator of the rates of translation during cortical differentiation is well supported and interesting but it is mixing the regulation of translation of Satb2 and Ctip2 mRNAs

with defects in neuronal polarity and migration makes everything unnecessarily complex.

We agree with the Reviewer that the manuscript that we presented last time was quite complex. We have made significant efforts to focus the manuscript more by restructuring it, as well as by presenting more data to support the link between Ire1 α and Satb2. We appreciate the Reviewer's claim that by including polarity data, we make the manuscript more complex. We now included the polarity experiments in the supplementary and reduced the amount of text describing these phenotypes.

Other issues:

1-It is not fully clear to me that the phenotypes of aberrant polarity and aberrant migration are not equally observed in electroporations targeting early-born and late-born neurons. If the expression of ire1a is higher in early precursors, the effects of knockin-out ire1a on early-born E12-E13 neurons should be more marked than the effects on e14 neurons.

We would like to point out that the protein abundance in biological systems does not necessarily correlates with the activity of a given protein. In the new version of the manuscript, we present the data on the expression of Ire1 α in the developing cortex (Fig. 5j). We show a very restricted expression of S724-P form of Ire1 α at the ventricular zone (Fig. 5j). Additionally, we now show thorough quantifications of morphology and the laminar positioning of bipolar neurons and cells with disturbed polarity across the cortical plate (Fig. S3m-S3q, S4d and S6a-S6d). In both neuron types, the early progenitor (E12.5)- and the late progenitor (E14.5)-derived ones, we observe approximately half of neurons to have bipolar morphology. This itself indicates that even within a defined cortical lineage, there is some biological sensitivity to a given axon formation program (in this case mediated by Ire1 α). There are other molecular programs that a neuron employs to form an axon that do not converge on Ire1 α -regulated signaling.

2- If possible, clarify if cells with the abnormal polarity in reconstructed neurons are the ones that fail to migrate.

We now provide such quantification (Fig. S3o-S3q and S6b-S6d).

3-Represent the data from individual mice separately and analyze statistical differences between individual animals within conditions, do not pull all neurons from all animals for each condition.

We thank the Reviewer for this comment. We followed the format on this sort of data presentation, as we believe it allows for the better representation of

neuronal positioning, as published (Tabata, et al., 2022, Nat Commun). Its big disadvantage is the bias towards overrepresentation of effects coming from cortices with higher electroporation efficiency. We now provide the additional quantification using cortical bins, which allows for normalization to the total number of electroporated cells (Fig. S3e-S3f, S3o-S3q, S6a-S6d, S7d-S7g, S7i-S7m, S9h).

5-In the western blots, it is unclear if the controls and bands are from the same gels. In Figure S4a, it seems that the bands are from different gels. If so, please make it obvious and show control bands for each of them.

We thank the Reviewer for this comment. We modified the figure accordingly (now Fig. S9a), as well as provided original images for all Western blotting data used for the representative data across the manuscript (Fig. S14).

6-In figure 6, it is shown that late progenitors have a higher translation rate than early progenitors, but it is my understanding that the roles in Ire1a dependent translation that are investigated by this work are in neurons. There is not sufficient data supporting the role specifically in precursors. The in vivo data shown in Figure 8 supports very well a role in postmitotic neurons.

Let us briefly explain. On the previous Fig. 6 (now Fig. 2), we have measured the HPG incorporation, a proxy for translation rates in the wild-type neurons. On previous Fig. 4a-4b (now Fig. 8a-8b), the Reviewer can appreciate the measurements of the translation rates in Ki67-positive progenitors in control and *Ire1a* KO. We now also include new quantifications regarding the translation efficiency of Satb2 and CTIP2 reporter in E12.5 control and *Ire1a* cKO progenitors (Fig. 9m-9r). We also show the VZ-restricted expression of enzymatically active form of Ire1 α , S724-P (Fig. 5j). To further support our findings, we also generated a new mouse line by crossing *Ire1a*^{fl/fl}; *Emx1*^{Cre/+} with MetRS* (Alvarez-Castelao et al., Nat Biotechnol, 2017), which allowed us to show diminished translation rates in E14.5 cortex upon Ire1 α disruption (Fig. S10). Given our measurements on Fig. 8, we do not claim that the role of Ire1 α in regulating translation rates is progenitor-specific. This is also not true in the light of our data on Fig. 9.

7- Images showing Satb2 expression in vitro cells show very low expression levels. In figure 7, there are very few red cells and the morphology is not purely neuronal. Please revise.

In old Fig. 7 (new Fig. 1), we show the result of the immunocytochemistry using Satb2 antibody. As the Reviewer can appreciate on a), we find cells that express high levels of Satb2, as well as low levels, yet the transfected neurons are clearly positive for Satb2. In this experiment, we did not quantify the intensity of Satb2

fluorescence, but quantified the proportion of cells that express is (both low and high). Cell negative for *Satb2* contain absolutely no signal. Given sparsity of cells in this experiment due to low-density plating, it is challenging to find an image which includes the red and green cells together. We have now included smaller insets to better visualize the *Satb2* expression levels.

Regarding the comment of the Reviewer about “non-neuronal morphology of cells”, we have repeated the treatment regime and stained for NeuN as a marker for neurons (Fig. S1h and S1k).

8- Have the authors attempted to overexpress *ire1a* at later stages and analyze the translation rates?

In our manuscript, we present that *Ire1 α* , a major regulator of the UPR in a stressed cell, also has a non-canonical function in safeguarding of cellular translation. We show that the level of the well-established *Ire1 α* splicing client *Xbp1S* is unaltered in the cKO cortex (S9a-S9b). Overexpressing *Ire1 α* is an experimental way to trigger the UPR (Tirasophon et al., 2000, *Genes Dev*; Lee et al., 2002, *Genes Dev*; Acosta-Alvear, 2018, *eLife*), resulting in *eIF2 α* phosphorylation on Ser52 and global shutdown of canonical protein translation, rendering the ribosomes to initiate synthesis of pseudoORFs. In this way, by employing two downstream pathways, *Ire1 α* is a potent regulator of protein translation, but its overexpression would conceivably result in translation inhibition. To address the question of the Reviewer we overexpressed *Xbp1S* in a wild-type cortex and failed to observe any effect on the proportion of *Satb2*- or *CTIP2*-expressing neurons (Fig. S3k-S3l). Moreover, we mimicked UPR activation in the cortex by replacing the endogenous *eIF2 α* with its phospho-mimic and -deficient isoform and showed a generalized loss of neuronal differentiation and diminished cortical plate entry (Fig. S11e-S11g).

9- Increased translation rate is a hallmark of early-born, deeper layer postmitotic neurons in the developing cortex and then the reverse in postmitotic neurons, but it is unclear to me how this occurs and if it is related to *ire1a*.

We believe the Reviewer is asking about the mechanism of translation dynamics in the developing cortex and how *Ire1 α* is involved. We have now performed additional experiments *in vivo*, as well as further validated our previous findings regarding both points. Let us briefly explain. We indeed show that in cultured DIV1 cells, the translation rate is indeed higher in E12.5-derived cortices as compared to E14.5-derived ones (Fig. 4e-4h): We now provide the measurements of translation rates in the P2 cortex using *MetRS** reporter mouse (Fig. 4a-4c). Notably, we report that the translation rates are constant within early E12.5 progenitor-derived lineages. In E14.5 progenitors, we indeed see a spike of translation that then diminishes in their postmitotic progeny in cultures (Fig. 2a-2c). We now provide data on the translome of the late lineages

and show timed and developmental age specific synthesis of initiation factors and ribosome constituents (Fig. 3, Table S2).

Further, we show that *Ire1α* regulates translation in cortical progenitors (Fig. 8a-8b and Fig. 8e-8f) by regulating the speed of ribosome translocation (Fig. 8h) and the number of translation initiation sites (Fig. 8j-8l). Using the *MetRS^{*}; Ire1α^{flf}; Emx1^{Cre/+}* mouse, we also show diminished translation rates in E14.5 cortex. We demonstrate that *Ire1α* controls the types of transcripts being translated, and specifically affects the synthesis of polypeptides regulating protein synthesis, such as ribosome constituents, cytoplasmic translation and RNA processing (Fig. 6). We also show that *Ire1α* loss induces ribosome stalling in neurons, indicative of hampered translation (Fig. 7a-7b).

We do not claim that *Ire1α* is the only translation regulator in the developing cortex, however, our experiments unveil that it is an important molecule able to regulate the rate of protein synthesis in the developmental systems.

10-the authors conclude that the loss of *Ire1α* results in suppression of translation rates as an effect of slower elongating ribosomes and decreased translation sites, but the effects are the same at E12 or E14. It is unclear to me if it should be expected that the data is not the same in the two different stages.

We appreciate the Reviewer's comment. However, our data clearly show that the disruption of *Ire1α* in the developing cortex, as well as in murine embryonic fibroblasts, consistently causes diminished translation rates (Fig. 8). As we mentioned before, we do not believe that *Ire1α* is the sole translational regulator in the cortex, and do not ascribe the differences we detect in the wild-type (Fig. 1-4) solely to *Ire1α* activity.

11-There are some editing mistakes, for example in the intro: is required to from the corpus callosum.

We heavily edited the manuscript.

Reviewer #3 (Remarks to the Author):

In this manuscript Ambrozkiwicz and colleagues identify, using a small molecule screen, Ire1 α as an essential determinant of SatB2+ upper layer neuron identity. The authors then go on to characterize the expression pattern of Ire1 α and the mechanism by which it drives cortical upper layer neurogenesis. The experiments presented suggest that Ire1 α is a determinant of global translation rates by dynamically interacting with the ribosome and regulating the levels of eIF4A1 and eEF-2 expression. The global decrease in translation appears to be caused by increased ribosome stalling and reduced numbers of translation initiation sites. Intriguingly, as the levels of alternatively spliced Xbp1, a hallmark of canonical UPR, do not change, a non-canonical UPR-independent role of Ire1 α in the regulation of polysome levels is proposed. The authors put forward the idea that whilst eEF-2 is involved in cortical lamination, eIF4A1 regulates acquisition of upper layer identity downstream of Ire1 α in a mechanism of translational control dependent on 5'UTR-embedded structural elements in the Ctip2 and SatB2 mRNAs. The finding that developmental regulation of ribosome dynamics may be responsible for establishment of neuronal diversity and cortical layering is interesting and novel, as the developmental neuroscience field so far has focused predominantly on genomic and transcriptional events. The authors present a compelling body of work and the quality of the experiments is high. However, owing to the large amount of data presented and the non-uniform colour scheme in the figures and labels, the text is often very difficult to follow, especially for a reader not well acquainted with developmental neuroscience. We provide questions and suggestions for improvement below.

MAJOR

1. The manuscript starts with the observation that the Ire1 α inhibitor APY69 reduces Satb2 expression. The authors should specify (in the main text and with new supplementary figures) how many and which inhibitors were tested and how APY69 compares to all of them.

We thank the Reviewer for this remark. We now show the screening results as well as validation, indicating that APY69 was indeed one of the most potent molecules in the screening (Fig. 5a-5g, Table S3).

2. Fig. 3c (polysome RNA-seq):

- Line 286-288: "Given no gross changes on the RNA level (Fig. 3b), we conclude that the expression level changes for these factors are of translational nature". Genes that are regulated at the translational level should come up in their polysome RNA-seq differential expression analysis. Therefore, to back up the claim in line 286-288, the authors should show where those factors lay on the volcano plot in fig.3c. Additionally, the authors should clarify why they think the regulation of a post-translational modification (like eEF2 phosphorylation) is "of translational nature".

- Considering the observation in fig.3a (=increased polysomes in the cKO), one would expect that RNA sequenced from the polysome fractions of the cKO would be more

than in the CTR. In this case, the volcano plot should be skewed towards the cKO. Instead, from fig.3c one would infer that the two distributions are equally centered. Is this the case? The authors should expand on this and explain better their analysis pipeline in the methods section.

We thank the Reviewer for raising these important points. Let us begin by explaining, that during the revision, we have repeated the sequencing experiment. We added another batch of samples and now comprehensively describe the transcripts found in the polysomes in control and in the cKO cortex (Fig. 6, Table S4). We now compare Differential Gene Expression and Transcript Expression across polysome fractions. We also included marking of some selected proteins, according to their function (Fig. 6e).

Regarding the second point of the Reviewer, we also rewrote the methods section to better explain the pipeline. We also provided the code, written for this analysis. Sequencing data are available on NCBI under GSE172489. All code used in analysis of RNAseq can be found at https://github.com/qoldt/IRE1aKO_Polysome_RNAseq.

We would expect skewing in the case of a generalized stalling, or "unspecific" recruitment of all transcripts to the polysome fraction. In our analyses, we see that some polysome-bound transcripts are less and some more represented in the cKO cortex (Fig. 6b, Table S4). We would also like to point out, that we now present the results of the ribopuromylation assay to label stalled ribosome in neurons (Fig. 7a).

- Genes displaying a significant fold-change have different colors. It's not clear why. They should explain this in the figure legend.

We have included more replicates to our sequencing approaches and fully reworked the results and their representation (Fig. 6)

3. Fig. 3i-j: The authors claim the interaction of Ire1 α with the small but not with the large ribosomal subunit, based only on RPS6 and RPL7. The authors should test other RPS and RPL to confirm that the observation is subunit-specific, and not protein-specific.

The interaction between Ire1 α and the ribosome is well established and published (Acosta-Alvear et al., 2018, eLife). We have also attempted further IPs as requested by this Reviewer, however, not successfully. For this reason, we edited the sentence and now talk about a specific dynamic interaction between Ire1 α and RPS6 (page 11, line 376).

4. The authors could comment on their interpretation as to why they observe a phenotype for eIF4A1, but not of any other members of the initiation complex (fig.3-S4). In particular, the authors should also test eIF4A2 levels. It is in fact known

(Williams-Hill et al. 1996) that eIF4A1 mRNA is translated less efficiently in post-mitotic cells, while eIF4A2 mRNA show the opposite trend. As progenitors differentiate into post-mitotic neurons, the authors should test whether the observed reduction in eIF4A1 is accompanied by an increase in eIF4A2 levels.

This is a very interesting point. We now provide a couple of datasets addressing this point of the Reviewer. First, we show that the expression pattern of eIF4A2 in fact points at the cortical plate enriched expression at E15 (Fig. S9q), while eIF4A1 is consistently found throughout the section (also compare Fig. S9n). Moreover, we tested the progenitor-specific expression of Ire1 α , eIF4A1 and eIF4A2. Briefly, we purified FlashTag-labeled apical progenitors from E14.5 cortex using FACS and using Western blotting detected all three proteins to be expressed in the progenitor cells in vivo (Fig. S9r-S9t). Finally, we show no gross changes of eIF4A2 levels in Ire1 α cortical lysates (Fig. S11c).

We also now show that once Ire1 α is inhibited, eIF4A1 is destabilized, which points at Ire1 α -regulated eIF4A1 proteostasis (Fig. 7k-7m).

5. To better support their claim of a UPR-independent mechanism, the authors should expand on their observation, which is based exclusively on spliced Xbp1 levels, and systematically check in their omics data all the known UPR-target genes and by WB at least some more candidates (like Atf4, ...).

We thank the Reviewer for this comment. This inspired us to also analyze how mimicking UPR in the developing brain affects neuronal diversity. First, as mentioned above, we reworked the polysome sequencing datasets and now present a more comprehensive list of affected transcripts, majority of them related to translation, ribosome constituents and RNA processing (Fig. 6b and 6e-6f). Secondly, the Western blotting for UPR proteins in *Ire1 α* cKO cortex revealed no gross changes to ATF-4, ATF-6, CHOP, BiP, PERK or HSP90 (Fig. S11b and S11c). We also immunostained control and cKO cortex for JNK2 and its phosphorylated form and detected no gross changes. Finally, we replaced endogenous eIF2 α with its S52 phospho-deficient and phospho-mimic variant. During UPR, S52 phosphorylation acts as a master switch-off for protein translation. We now show that mimicking UPR leads to general neuronal differentiation defects and hinders cortical plate entry, rather than specifically affects generation of Satb2 neurons (Fig. S11a and S11e-S11g).

6. Line 452-453: "Altogether, these data indicate that deep layer-fated neurons have inherently higher translation rates compared to upper layer neurons". Increased translation in deep layer neurons may not be an intrinsic feature of upper layer neurons, rather a developmental artefact in their in vitro experiments, as the two populations will be doing different things at different times.

To support their claims the authors should measure protein synthesis in Ctip2+ and Satb2+ neurons in intact tissues of different developmental stages. We suggest an ex

vivo puromycilation assay (e.g. Biever et al 2019) or a FUNCAT assay on brain slices across developmental time-points. The authors could then perform immunostaining against Puro, Ctip2 and Satb2 and quantify the protein synthesis signal in each cell type.

Although quite challenging, this experiment requested by the Reviewer provided exciting new data. We now include the data on the ex vivo puromycilation using brains slices (Fig. 3a), corroborating the finding from the culture (Fig. 2). Additionally, we have crossed MetRS* mouse line with our *Emx1^{Crel+}* driver to metabolically label Satb2- and CTIP2-expressing neurons within the cortical plate at P0 (Fig. 4a-4c). Moreover, we also now show the translome of E15.5 cortex-derived neurons as compared to E12.5 ones (Fig. 3d), revealing that specific translational requirements are developmental stage-specific. We would also like to point out that we crossed MetRS* also with *Emx1^{Crel+}; Ire1a^{fl/fl}* and using BONCAT now corroborate diminished ANL incorporation in E14.5 cortex (Fig. S10).

7. The authors could comment on why CTIP2 neurons don't show a translational dependent phenotype in fig.7, when at E12 the treatment is applied. Perhaps if they anticipated the time of treatment, CHX would also affect Ctip2 identity specification. With regards to the effect that CHX has on Satb2 cell identity, the authors should clarify what is the alternative fate gained, as it is not Ctip2 according to their data. The authors should perform staining for the main neuronal identity markers (ie. FezF2, Cux1, Cux2, FoxP2 and Tbr1) to tackle this point.

We now show that CHX treatment seems to specifically affect Satb2 lineage (Fig. S1h-S1k) at DIV5. Unfortunately, we show the quantification using Tbr1, Brn2 and NeuN markers, due to technical issues with stainings using the other markers (particularly in cultures). We sincerely hope this is acceptable for the Reviewer. We believe the translational sensitivity is a specific feature of Satb2 neurons because of our previous proteomics findings (Harnett and Ambrozkiwicz, 2022, Nat Struct Mol Biol) that Satb2 translation is up-regulated mid-gestation, as well as quite a distinct enrichment of G-quadruplexes in Satb2 5'UTR, as compared to CTIP2 (Fig. S6b-S6e). We believe extending the time of CHX treatment could potentially affect the culture viability (Jareb and Banker, 1997, J Neurosci).

8. The model of differential 5'UTR recruitment is interesting. However, it needs further validation. For example, if the model is correct, one should see CTIP2 and Satb2 mRNA differentially regulated in their polysome RNAseq data set. Is that the case? Additionally, the authors should cluster the genes detected in their polysome RNAseq data set according to G-quadruplexes occurrence and test whether they are differentially regulated.

We thank the Reviewer for this comment. We tried such clustering, but the effect was quite minor. Our interpretation of this was that - according to our data on Fig. 6 - *Ire1α* specifically regulates translation of protein synthesis machinery, including ribosome constituents and RNPs. Increased polysomes in the cKO represent stalled ribosomes (Fig. 7a-7b), overall hindering cellular translation. G-quadruplexes in the case of *Ire1α* cKO are not a general hub of stalling of all transcripts, but rather represent a specific context of eIF4A1-mediated regulation of *Satb2*. We now present data on the number of G-quadruplexes in 5'UTRs of major neuronal type determinants (Fig. S8d-S8e). *Satb2* is clearly an outlier, regarding the number of G-quadruplexes found in its 5' UTR, making it uniquely sensitive to eIF4A1 and - indirectly - *Ire1α* loss.

We would like to point out, we also attempted RiboSeq from the control and cKO cortex, but the experiment, due to technical issues, unfortunately failed.

Another technicality that we think might be an issue here is the material needed for the polysome pulldowns. For the analysis at E18.5, we needed approximately 150 cortices. Such analysis at earlier timepoints would be very informative but would require significantly more input material.

We have now added a rescue experiment in E12.5 cortical progenitors of control and *Ire1α* cKO using wild-type eIF4A1 and its helicase-deficient mutant (Fig. 9m-9r) and show different translation efficiencies for *Satb2* and CTIP2 specifically in progenitors. We amended the conclusions about differential recruitment to regulation of translation of *Satb2*.

9. Line 293: the authors should comment on whether the effect of EEF2 is only on the distribution of neurons across the cortical wall or it also increases neurogenesis, as suggested by the overall increase in EGFP+ cells seen in figure S4e.

As we have explained to the Reviewer down in the answer to their point 18., all our quantifications are normalized to the total number of EGFP-expressing cells in a given section. This is the generally accepted way to compare between in utero electroporated brains. However, we also included the experimental data for cell cycle exit of neurons upon eEF-2 overexpression (Fig. S9o-S9p). It indeed seems to be a mixed effect and we have amended our conclusions.

10. Fig. S4: Nickases are more likely to result in small in frame indels. The authors must show that they effectively KO the gene by immunostaining.

Due to eIF4A1 staining pattern within the cortex, we have corroborated the targeting in culture (Fig. S9k-S9l). Additionally, our construct is commercially available and has been validated before (Addgene #122345 and #122346; Ochiai et al., Sci Adv., 2020).

11. Line 316-320: The authors should rephrase this paragraph to tone down their conclusions, the experiments are not so black and white.

We agree with the Reviewer and now edited the paragraph (page 11, line 383-384).

12. The authors make the claim that eIF4A1 and eEF-2 have very distinct roles, controlling upper layer specification and lamination, respectively. However, from their data (Fig. S4) it seems as though eEF-2 might also affect progenitor divisions and neuron production. The authors should perform a pH3 stain in CTRL and eEF-2 OE samples to address this.

We now provide new experimental evidence that this is indeed the case (Fig. S9o-S9p), using another mitotic marker Ki67. We find that this is a broader marker to cover most of the mitotic stages and identify proliferating cells (Uxa et al., Cell Death Diff, 2021).

13. The authors should use color palettes more consistently across (similar) experiments and prefer color-blind friendly tones.

We would like to respectfully explain why we use different palettes. MCA, who handled the manuscript editing and figure assembly, is color-blind, and representing staining results in general is challenging. Consistent color choices would in our opinion be more confusing, given the number of different panels in this work. We therefore decided to represent the results using somewhat varying color choices, always defining the staining represented by a given color directly on the figures.

MINOR

14. Fig.1b: the levels are normalized over DMSO for 0.2 μ M samples, but over the untreated for 1 μ M samples. The authors should use the same control across conditions.

All samples were normalized to the untreated condition (Table S3). DMSO did not change Satb2tdTom fraction.

15. Fig.1b: the authors should shortly describe how they calculate the “relative Satb2tdTom fraction”. Is it tdTomato+ (Q2+Q4) / EGFP+ (Q1+Q2) and then further normalized to DMSO (or Untreated)? Citing in the methods section ref.26 is not enough.

Yes, that is correct. We included this information in the supplementary methods, as requested by the Reviewer.

16. Fig.1d: the authors should show where VZ is in the zoom out

We now provide the zoom out image (Fig. S3c).

17. Fig. S2: from the images it seems that there are more CTIP2+ cells (especially in the deeper layers) in the cKO than in the CTR. This is not reflected by the quantification. The authors should clarify this.

We quantified the total numbers of Satb2 and CTIP2 cells in these sections and detected no differences. We, however, do report a change in the position of Satb2 and CTIP2 positive neurons within the cortical plate (Fig. S7n).

18. Fig. 1e: In the images shown the transfection efficiency is very different and this could explain the skew in the distribution of the neurons seen. More similar images should be shown the new panel should include single channel images of the EGFP signal, so that reader can better appreciate the increased branching.

The transfection efficiency during in utero electroporation is not an issue, because we normalize the number of EGFP-positive and marker positive neurons to the total number of EGFP-positive neurons/cells within a given section (please also compare our previous work, Ambrozkiwicz et al., 2018, Neuron; 2021, Mol Psychiatry; Kraushar et al., 2021, Mol Cell). This is the case for marker expression, as well as for quantification of laminar positioning within the cortical plate. As mentioned before, we now provide these quantifications as the relative abundance of neurons within a given cortical bin (Fig. S3e and S3f, S6a). Regarding the comment about changing the figure to a single channel - given a lot of data panels, we already had a major difficulty in putting together the figure panels as they are. For this reason, we provide the tracings of CP-neurons next to the images (Fig. S3m, S4a). Additionally, we now provide quantifications of laminar positioning of bipolar neurons and neurons with aberrant morphology across the CP in E12.5 and E14.5 progenitor-targeted electroporation (Fig. S3o-S3q, S6b-S6d) as well as the data after expansion microscopy to demonstrate the morphology of control and KO neurons (Fig. S4c).

19. Fig. 1g should include representative images.

We now provide the representative images for fate quantifications (Fig. S13) and Western blotting (Fig. S14).

20. Fig. 4h: The western blot and the quantifications in 4h do not mirror each other. In the blot it seems that the decay curve between Cre and control are different.

We are very grateful for this comment of the Reviewer. We have repeated the experiment with harringtonine and added new datapoints to the graph (Fig. 8h).

Suggested Changes to text.

We thank the Reviewer for taking the time for the text edits. Because we were advised to refocus the manuscript, we have rewritten parts of it and edited the text. We would like to ascertain the Reviewer, that we introduced all the suggested edits.

21. Line 113: the authors should add a line to explain the aim of the BrdU experiment (something like “to label proliferating cells”).

Page 9, line 290.

22. Line 118: the authors should spell out cortical plate “CP”

Page 8, line 220.

23. Line 184: “This contrasted with the majority of control neurons characterized with a single TP and lack of additional neurites emanating from the soma at this stage (Fig. 2d).” Perhaps the authors were referring to Fig. 2f instead of 2d? Or are they using “bipolar neurons” as a proxy for the described traits?

The latter is correct.

24. Fig. S4e: eIF4A1 KO has the wrong color palette (green instead of yellow?)

We apologize for this. MCA, who wrote the manuscript, assembled the figures and wrote the rebuttal is color blind.

25. Fig.3i legend: missing explanation for “H” (compared to “IP”)

Now Fig. 7j, we rephrased the figure legend.

26. The “y-axes” of Fig.3e and 3i look misplaced/unclear. They authors should clarify

We removed these.

27. Line 447: The sentence is a bit misleading. Maybe it’s worth editing it to something like “We detected higher HPG incorporation rates in e14 than e12 cells also at DIV5.”

We edited the text of the manuscript.

28. Figure 1g should include representative images

We added representative images on Fig. S13 and S14.

29. In the text Figures S1D, E, F are not referenced in the order they appear in the figures

Now Fig. S3r-S3w, we also now mention it in the text (page 8, line 235-240).

30. Line 651-2: sentence is obscure. Authors should rephrase it.

We edited the text of the manuscript.

31. Line 35: typo “determine”, not “determines”

We meant that Satb2 determines the corpus callosum. We edited the sentence so there is no confusion (page 3, line 12).

32. Line 112: “of THE dorsal telencephalon”

We edited the text of the manuscript.

33. Line 289: authors should cite fig. S4a-b.

We edited the text of the manuscript.

34. Line 296: “to THE developing cortex”

We edited the text of the manuscript.

35. Line 740: typo “by in part”.

We edited the text of the manuscript.

REVIEWER COMMENTS

Reviewer #1 (Remarks to the Author):

This is an improved version of this manuscript. It still contains a huge amount of data and is therefore quite dense. However conclusions seem more logical in this revised version.

Some improvements / clarifications can still be made:

Two key sentences seem almost contradictory and it would make sense to simplify these so that they appear more complementary: End of abstract: 'Here, we show that cortical neuron diversity is generated by mechanisms operating beyond the gene transcription, with Ire1 α -safeguarded proteostasis serving as an essential regulator of brain development.'

End of introduction 'Additionally, this study extends the function of Ire1 α beyond the regulation of proteostasis during cellular response to stress to an innate developmental requirement for neuronal specification in the cortex.'

Abstract 'unique sensitivity in the development of neurons expressing Satb2, a determinant of upper cortical layers, to translation rates'. I'm surprised that half-lives of proteins aren't mentioned in the manuscript? This seems missing.

Some exaggeration in the Results 'displayed profoundly higher translation rates as compared to their E12.5 predecessors' I would remove the word 'profoundly'

Beginning of page 6: 'biological pathways such as ER stress, RNA stability, processing and splicing, cytoplasmic translation and metabolic processes' – this description is not obvious from categories shown in 3d.

Fig 2c: Ire1 α is on the schema, whereas it hasn't yet been mentioned in the Results section?

Line 195: 'Notably, we detected robust VZ-restricted immunostaining of S724-phosphorylated Ire1 α ' – there seems to be labelling in superficial regions as well as E14.5?

Line 218-219: 'fewer Satb2-expressing neurons COMPENSATED BY MORE CTIP2-positive ones'

Lamination: Fig S3e: Ire1 α inactivation does not affect lamination (Satb2, Ctjp2, E16.5); Fig S4a it does affect lamination (Cux1 E18.5) – this is a bit hard to follow.

The Fucci conclusions are a little unclear: what is the interpretation of these results (Venus and mCherry)?

Line 410: This conclusion is hard to determine from the graph: 'Ire1 α KO cells also showed diminished response to the harringtonine treatment, reflecting slowly elongating ribosomes (Fig. 8h).'

Fig S8b- e may not be mentioned in the correct order in the manuscript Line 445: 'predicted G4s in Satb2 mRNA among classical neuronal fate determinants (Fig. S8b-S8e).'

This sentence already appears after Fig S11 has been cited.

Some typos / English grammar need correcting:

Abstract 'an enormous diversity' '5'-untranslated' 'beyond gene transcription'

Intro 'ventricular progenitor states' 'recently demonstrated such a mechanism' 'translational control of the 5'UTR of Satb2 itself'

Results 'development alters the type OF neurons' 'projected A tau-positive axon' 'reveals A critical translational window'

Line 137: 'We next took advantage of MetRS* mouse', could be 'We took advantage of the MetRS* mouse

Line 160 'Neurons derived from E14.5 cortex WERE enriched for Satb2 expression.'

Fig 6 legend: Polysome-enriched transcripts encode proteins... (not encode for)

Line 331: encoded for structural.. should be encoded structural

Line 334: including eIF4F (Fig. 3d), I think you mean Fig 6d?

Line 433: and diminished CP entry (Fig. 11f and S11g), should be Fig. S11f?

Line 434: 'These results demonstrate that cortical cell diversity... it would be better to say 'These results SUGGEST that cortical cell diversity

Line 448: 'We first constructed a fluorescence-based translational reporters... should be 'We first constructed fluorescence-based translational reporters...

Line 453: 'translation efficiency (Fig. 9a and 9b) and increased one for CTIP2 (Fig. 9c and 9d), should be 'translation efficiency (Fig. 9a and 9b) and increased CTIP2 translation efficiency (Fig. 9c and 9d),

Line 469: 'downstream of Ire1 α is embedded in THE 5'UTR of Satb2

Line 470: 'Representative pictures for neuronal fate quantification can be found IN Fig.

Line 498: 'According to our data, THE 5'UTR in Satb2 mRNA requires activity of translation

Line 504: 'THE CTIP2 reporter is translated more efficiently

Line 519: 'able to generate Satb2- and CTIP2-expressing neurons ARE present in the cortex

Reviewer #2 (Remarks to the Author):

The authors have improved significantly the structure and organization of the manuscript and I like the focus they now present. However, with this new and more clear focus, there is still an important issue that I would like the authors to clarify and that I already raised in my first review concerning the phenotypes of the cKO Ire1 α Flox; EmxCRE. To test in vivo the hypothesis of Ire1 α regulating Stab2 in vivo, the authors create a cortical cKO of Ire1 α by crossing a floxed Ire1 α allele with an emx1 Cre line. The immunofluorescences shown in FigS7 show that in cKO (Ire1 α Flox; EmxCRE) the expression of Stab2 is hardly affected. Still, in the main text, the authors use the KO only to describe the loss of Ire1 α upon cre expression in progenitors. They turn to in-utero electroporation of CAG-CRE plasmids to analyze the expression of Stab2 and Ctip2 expression without any explanation. They then conclude from immunofluorescence that Ire1 is required for Stab2 expression. This seems a contradiction that I cannot reconcile. I recommend showing an analysis of Stab2 and Ctip2 in cKO Ire1 α Flox; EmxCRE, where according to their conclusion, they should see important reductions in Stab2 expression and abnormalities in neuronal projections and morphologies. In sum, I consider that the article requires a more profound description of the mutant Ire1 α Flox; EmxCRE phenotypes and clarifications of these aspects. Also, I would still like to see changes in the abstract, which does not reflect the argumental line of the main text. Other minor: Fig 8. Modify the headline-it reads now "Loss of Ire1 α results in suppression of translation rates as an effect of slower elongating ribosomes and decreased translation sites", I consider it more correct "Loss of Ire1 α leads to diminished translation rates...." as in Fig S10

Reviewer #3 (Remarks to the Author):

Ambrozkiwicz and colleagues have resubmitted a more complete and substantially improved study. The authors have done most of the revision experiments requested, the majority of which are of sufficient quality and support their claims. We ask the reviewers to address the last few points that we have raised in the point-by-point rebuttal and to tone down claims that the revision experiments do not fully support. We understand that the color scheme choice was dictated by necessity, but in general we feel that the paper would benefit from simplification and encourage the authors to do so, both in terms of text and figure display. Provided the authors address the remaining considerations outlined below, we recommend acceptance of the manuscript for publication.

MAJOR

1. The manuscript starts with the observation that the Ire1 α inhibitor APY69 reduces Satb2 expression. The authors should specify (in the main text and with new supplementary figures) how many and which inhibitors were tested and how APY69 compares to all of them.

We thank the Reviewer for this remark. We now show the screening results as well as validation, indicating that APY69 was indeed one of the most potent molecules in the screening (Fig. 5a-5g, Table S3).

Reviewer response: We thank the authors for sharing the full screening results. Could they please include in the main text a comment as to why they chose to focus on APY69, instead of Dinaciclib and C71, both of which show a stronger effect at lower concentrations? What pathways do these compounds inhibit/activate?

We thank the Reviewer for raising these important points. Let us begin by explaining, that during the revision, we have repeated the sequencing experiment. We added another batch of samples and now comprehensively describe the transcripts found in the polysomes in control and in the cKO cortex (Fig. 6, Table S4). We now compare Differential Gene Expression and Transcript Expression across polysome fractions. We also included marking of some selected proteins, according to their function (Fig. 6e).

Reviewer response: How can the authors explain the presence of non-coding RNA in their polysome fractions? The authors should add a brief comment on this.

This is a very interesting point. We now provide a couple of datasets addressing this point of the Reviewer. First, we show that the expression pattern of eIF4A2 in fact points at the cortical plate enriched expression at E15 (Fig. S9q), while eIF4A1 is consistently found throughout the section (also compare Fig. S9n). Moreover, we tested the progenitor-specific expression of Ire1 α , eIF4A1 and eIF4A2. Briefly, we purified FlashTag-labeled apical progenitors from E14.5 cortex using FACS and using Western blotting detected all three proteins to be expressed in the progenitor cells in vivo (Fig. S9r-S9t). Finally, we show no gross changes of eIF4A2 levels in Ire1 α cortical lysates (Fig. S11c).

We also now show that once Ire1 α is inhibited, eIF4A1 is destabilized, which points at Ire1 α -regulated eIF4A1 proteostasis (Fig. 7k-7m).

Reviewer response: The blot in figure 7k is missing a loading control. It is not clear what is quantified in 7l. Are these the quantifications of the blots shown in 7m? In that case it would make more sense to swap the labels. Additionally, if the quantifications in 7l correspond to 7m eIF4A1, at t=0 the blot shows the opposite trend to what's reported in the quantifications. Could the authors provide an explanation?

Although quite challenging, this experiment requested by the Reviewer provided exciting new data. We now include the data on the ex vivo puromycylation using brains slices (Fig. 3a), corroborating the finding from the culture (Fig. 2). Additionally, we have crossed MetRS* mouse line with our Emx1Cre/+ driver to metabolically label Satb2- and CTIP2-expressing neurons within the cortical plate at P0 (Fig. 4a-4c). Moreover, we also now show the translome of E15.5 cortex-derived neurons as compared to E12.5 ones (Fig. 3d), revealing that specific translational requirements are developmental stage-specific. We would also like to point out that we crossed MetRS* also with Emx1Cre/+; Ire1 α f/f and using BONCAT now corroborate diminished ANL incorporation in E14.5 cortex (Fig. S10).

Reviewer response: I don't find the data presented 100% convincing. Radial glia cells have very

clear radial alignment and this should be reflected by high-intensity Puromycin-stained progenitors at E14. The blow up in figure 3b shows processes projecting incoherently and could be background staining of endothelial cells of the vasculature. Better quality images where progenitors are labelled by SOX2/Pax6 and Puromycin would better support these claims. Additionally, the line plots in figure 3a and 3c seem to be in contrast with the claims made, at E12.5 the profile reaches a grey value of 3000 in the VZ, while at E14.5 only 2500 and the prediction would be the opposite. In addition, it is very difficult to use such images for analyses as the stain appears to be uneven across the surface (e.g. 3A – left and right parts of the image) and the tiling gives obvious intensity artefacts at the edges of the individual tiles (3B). We ask the authors to provide better quality images to substantiate their claims. Additionally Fig.3c requires a loading control and relative quantification (with replicates).

Rebuttal Letter, Borisova et al., “Protein translation rate determines neocortical neuron fate”.

Reviewer Comments (Our responses in **bold**)

Reviewer #1 (Remarks to the Author):

This is an improved version of this manuscript. It still contains a huge amount of data and is therefore quite dense. However conclusions seem more logical in this revised version.

Some improvements / clarifications can still be made:

Two key sentences seem almost contradictory and it would make sense to simplify these so that they appear more complementary: End of abstract: ‘Here, we show that cortical neuron diversity is generated by mechanisms operating beyond the gene transcription, with Ire1 α -safeguarded proteostasis serving as an essential regulator of brain development.’

End of introduction ‘Additionally, this study extends the function of Ire1 α beyond the regulation of proteostasis during cellular response to stress to an innate developmental requirement for neuronal specification in the cortex.’

We thank the Reviewer for bringing this to our attention. Let us explain. In our work we uncover the regulation of protein translation as a mechanism to orchestrate cortical development by Ire1 α (abstract). On the other hand, we also show that Ire1a itself has an important role in regulating protein translation in cellular stress-independent context (introduction). We now amended these sections accordingly and only mention the post-transcriptional regulation by Ire1 α .

Abstract ‘unique sensitivity in the development of neurons expressing Satb2, a determinant of upper cortical layers, to translation rates’. I’m surprised that half-lives of proteins aren’t mentioned in the manuscript? This seems missing.

We included the half-lives in the abstract, as suggested.

Some exaggeration in the Results ‘displayed profoundly higher translation rates as compared to their E12.5 predecessors’ I would remove the word ‘profoundly’

We agreed with the Reviewer and removed the word.

Beginning of page 6: ‘biological pathways such as ER stress, RNA stability, processing and splicing, cytoplasmic translation and metabolic processes’ – this description is not obvious from categories shown in 3d.

We agreed with the Reviewer and changed the text (lines 138-142).

Fig 2c: Ire1a is on the schema, whereas it hasn't yet been mentioned in the Results section?

We agreed with the Reviewer and removed the Ire1a from the figure.

Line 195: 'Notably, we detected robust VZ-restricted immunostaining of S724-phosphorylated Ire1 α ' – there seems to be labelling in superficial regions as well as E14.5?

We changed the word "restricted" to "enriched" (line 204).

Line 218-219: 'fewer Satb2-expressing neurons COMPENSATED BY MORE CTIP2-positive ones'

We changed the phrasing as suggested by the Reviewer.

Lamination: Fig S3e: Ire1a inactivation does not affect lamination (Satb2, CtIP2, E16.5); Fig S4a it does affect lamination (Cux1 E18.5) – this is a bit hard to follow.

The quantification on Fig. S3e reflects the E12.5 progenitor lineage labeled using in utero electroporation, whereas the dataset on Fig. S4a the E14.5 progenitor lineage. We also specify this clearly in the text (line 224/225, "to deliver control expression vector to progenitors at E12.5"; line 260, "L2/3-destined cortical neurons").

The Fucci conclusions are a little unclear: what is the interpretation of these results (Venus and mCherry)?

We thank the Reviewer for bringing this up, we have now introduced Fucci reporter earlier in the manuscript with relevant explanation (paragraph starting from the line 120).

Line 410: This conclusion is hard to determine from the graph: 'Ire1 α KO cells also showed diminished response to the harringtonine treatment, reflecting slowly elongating ribosomes (Fig. 8h).'

The representative WB and the graph (Fig. 8) display the experiment with the harringtonine, followed by puromycilation. EGFP-expressing *Ire1 α ^{fl}* cells deplete the pool of mRNA in translation already after 8 minutes of harringtonine exposure, whereas the Cre-infected cells at this timepoint still exhibit puromycilation signal, indicative of ongoing translation at this timepoint. We softened the conclusion here (line 425-427).

Fig S8b- e may not be mentioned in the correct order in the manuscript Line 445: 'predicted G4s in Satb2 mRNA among classical neuronal fate determinants (Fig. S8b-S8e).' This sentence already appears after Fig S11 has been cited.

We now split this figure in two (Fig. S8 and S12).

Some typos / English grammar need correcting:

We thank the Reviewer for taking the time for these edits. We corrected the text accordingly.

Abstract 'an enormous diversity' '5'-untranslated' 'beyond gene transcription'

Intro 'ventricular progenitor states' 'recently demonstrated such a mechanism' 'translational control of the 5'UTR of Satb2 itself'

Results 'development alters the type OF neurons' 'projected A tau-positive axon' 'reveals A critical translational window'

Line 137: 'We next took advantage of MetRS* mouse', could be 'We took advantage of the MetRS* mouse'

Line 160 'Neurons derived from E14.5 cortex WERE enriched for Satb2 expression..'

Fig 6 legend: Polysome-enriched transcripts encode proteins... (not encode for)

Line 331: encoded for structural.. should be encoded structural

Line 334: including eIF4F (Fig. 3d), I think you mean Fig 6d?

Line 433: and diminished CP entry (Fig. 11f and S11g), should be Fig. S11f?

Line 434: 'These results demonstrate that cortical cell diversity... it would be better to say 'These results SUGGEST that cortical cell diversity'

Line 448: 'We first constructed a fluorescence-based translational reporters... should be 'We first constructed fluorescence-based translational reporters...'

Line 453: 'translation efficiency (Fig. 9a and 9b) and increased one for CTIP2 (Fig. 9c and 9d), should be 'translation efficiency (Fig. 9a and 9b) and increased CTIP2 translation efficiency (Fig. 9c and 9d),

Line 469: 'downstream of Ire1 α is embedded in THE 5'UTR of Satb2'

Line 470: 'Representative pictures for neuronal fate quantification can be found IN Fig.

Line 498: 'According to our data, THE 5'UTR in Satb2 mRNA requires activity of translation

Line 504: 'THE CTIP2 reporter is translated more efficiently

Line 519: 'able to generate Satb2- and CTIP2-expressing neurons ARE present in the cortex

We appreciate the Reviewer for their careful reading of our work. We amended the text exactly as suggested.

Reviewer #2 (Remarks to the Author):

The authors have improved significantly the structure and organization of the manuscript and I like the focus they now present. However, with this new and more clear focus, there is still an important issue that I would like the authors to clarify and that I already raised in my first review concerning the phenotypes of the cKO *Ire1α*^{Flox}; *Emx*CRE. To test in vivo the hypothesis of *Ire1α* regulating *Stab2* in vivo, the authors create a cortical cKO of *Ire1α* by crossing a floxed *Ire1α* allele with an *emx1* Cre line. The immunofluorescences shown in FigS7 show that in cKO (*Ire1α*^{Flox}; *Emx*CRE) the expression of *Stab2* is hardly affected. Still, in the main text, the authors use the KO only to describe the loss of *Ire1α* upon cre expression in progenitors. They turn to in-utero electroporation of CAG-CRE plasmids to analyze the expression of *Stab2* and *Ctip2* expression without any explanation. They then conclude from immunofluorescence that *Ire1* is required for *Stab2* expression. This seems a contradiction that I cannot reconcile. I recommend showing an analysis of *Stab2* and *Ctip2* in cKO *Ire1α*^{Flox}; *Emx*CRE, where according to their conclusion, they should see important reductions in *Stab2* expression and abnormalities in neuronal projections and morphologies. In sum, I consider that the article requires a more profound description of the mutant *Ire1α*^{Flox}; *Emx*CRE phenotypes and clarifications of these aspects.

We thank the Reviewer for raising this important point. Let us explain our rationale. The requested BrdU experiment, as well as analysis of the lamination of *Satb2* and CTIP2 neurons at P2 were performed in the first Revision (Fig. S7). As the Reviewer correctly points out, in the cKO P2 cortex we observe mild changes in the lamination of neurons. This is partially the reason, why we decided to use Cre electroporation in the *f/f* line to study the cell-autonomous roles of *Ire1α* during development. In our previous studies (Ambrozkiwicz et al., *Neuron*, 2018; Ambrozkiwicz et al., *Mol Psych*, 2021 among others), we did observe compensatory mechanisms in the *Emx1* cKO regarding cortical lamination, whilst reporting distinct phenotypes when acutely delivering Cre using in utero electroporation. The existence of such mechanisms is further substantiated by the recent publication providing evidence on tissue-wide effects overriding cell-intrinsic gene function and that acute Cre expression (for example using IUE) might circumvent such compensation in the cKO cortices during development (Hansen et al., *Oxford Open Neuroscience*, 2022).

However, we have taken the Reviewer's point seriously and analyzed the *Satb2* and CTIP2 neuronal counts during embryogenesis, as well as the morphology of cortical plate neurons in our *Ire1α* cKO (Fig. S8). We indeed corroborate the findings made before using electroporations in *f/f* line. We would like to point out that the reduction in *Satb2*-expressing neuron number does become compensated for postnatally in the cKO (Fig. S7). We also added the sentence explaining the usage of *f/f* line in the initial phenotyping (line 227-228).

Also, I would still like to see changes in the abstract, which does not reflect the argumental line of the main text.

We have restructured the abstract as suggested.

Other minor: Fig 8. Modify the headline-it reads now "Loss of Ire1 α results in suppression of translation rates as an effect of slower elongating ribosomes and decreased translation sites", I consider it more correct "Loss of Ire1 α leads to diminished translation rates...." as in Fig S10.

We modified the headline as requested.

Reviewer #3 (Remarks to the Author):

Ambrozkiewicz and colleagues have resubmitted a more complete and substantially improved study. The authors have done most of the revision experiments requested, the majority of which are of sufficient quality and support their claims. We ask the reviewers to address the last few points that we have raised in the point-by-point rebuttal and to tone down claims that the revision experiments do not fully support. We understand that the color scheme choice was dictated by necessity, but in general we feel that the paper would benefit from simplification and encourage the authors to do so, both in terms of text and figure display. Provided the authors address the remaining considerations outlined below, we recommend acceptance of the manuscript for publication.

MAJOR

1. The manuscript starts with the observation that the Ire1 α inhibitor APY69 reduces Satb2 expression. The authors should specify (in the main text and with new supplementary figures) how many and which inhibitors were tested and how APY69 compares to all of them.

We thank the Reviewer for this remark. We now show the screening results as well as validation, indicating that APY69 was indeed one of the most potent molecules in the screening (Fig. 5a-5g, Table S3).

Reviewer response: We thank the authors for sharing the full screening results. Could they please include in the main text a comment as to why they chose to focus on APY69, instead of Dinaciclib and C71, both of which show a stronger effect at lower concentrations? What pathways do these two compounds inhibit/activate?

We thank the Reviewer for pointing this out. We chose to focus on APY69 because of the result on Fig. 5f, showing APY69 as the most potent molecule able to promote Satb2-to-CTIP2 switch in neuronal cultures (line 192-195). We previously reported the effects of Dinaciclib, a CDK inhibitor, on neuronal cultures (Ambrozkiewicz et al., J Neurosci Methods, 2017). Compound 71 has anti-necrotic effect by inhibiting TNF pathway (Hofmans et al., J Med Chem, 2018).

We thank the Reviewer for raising these important points. Let us begin by explaining, that during the revision, we have repeated the sequencing experiment. We added another batch of samples and now comprehensively describe the transcripts found in the polysomes in control and in the cKO cortex (Fig. 6, Table S4). We now compare Differential Gene Expression and Transcript Expression across polysome fractions. We also included marking of some selected proteins, according to their function (Fig. 6e).

Reviewer response: How can the authors explain the presence of non-coding RNA in their polysome fractions? The authors should add a brief comment on this.

We believe it is not so surprising to see the non-coding RNA associating with polysomes. This has been reported before, for example Carlevaro-Fita et al., RNA, 2016; Booy, et al., JBC, 2021; Douka et al., RNA, 2021. Polysomal lncRNAs have been shown to display distinct molecular features. Moreover, it has been shown that inhibiting elongation might stabilize lncRNA-polysome associations. This last finding supports our hypothesis of Ire1 α involvement in regulating ribosome function. We mentioned the identification of non-coding RNAs in the line 339.

This is a very interesting point. We now provide a couple of datasets addressing this point of the Reviewer. First, we show that the expression pattern of eIF4A2 in fact points at the cortical plate enriched expression at E15 (Fig. S9q), while eIF4A1 is consistently found throughout the section (also compare Fig. S9n). Moreover, we tested the progenitor-specific expression of Ire1 α , eIF4A1 and eIF4A2. Briefly, we purified FlashTag-labeled apical progenitors from E14.5 cortex using FACS and using Western blotting detected all three proteins to be expressed in the progenitor cells in vivo (Fig. S9r-S9t). Finally, we show no gross changes of eIF4A2 levels in Ire1 α cortical lysates (Fig. S11c).

We also now show that once Ire1 α is inhibited, eIF4A1 is destabilized, which points at Ire1 α -regulated eIF4A1 proteostasis (Fig. 7k-7m).

Reviewer response: The blot in figure 7k is missing a loading control. It is not clear what is quantified in 7l. Are these the quantifications of the blots shown in 7m? In that case it would make more sense to swap the labels. Additionally, if the quantifications in 7l correspond to 7m eIF4A1, at t=0 the blot shows the opposite trend to what's reported in the quantifications. Could the authors provide an explanation?

We thank the Reviewer for pointing this out. We now provide a new edited Figure, with the loading control for Fig. 7k and changed lay-out of the panels. We also provide the Reviewer with the experimental data for this panel. We believe the representative panel corresponds to the quantification well. The panels are organized as DMSO/APY29 pairs, with 0h, 1h and 4h timepoints and not as DMSO 0h, DMSO 1h and DMSO 4h etc. We hope this clarifies the remark.

Although quite challenging, this experiment requested by the Reviewer provided exciting new data. We now include the data on the ex vivo puromycylation using brains slices (Fig. 3a), corroborating the finding from the culture (Fig. 2). Additionally, we have crossed MetRS* mouse line with our Emx1Cre/+ driver to metabolically label Satb2- and CTIP2-expressing neurons within the cortical plate at P0 (Fig. 4a-4c). Moreover, we also now show the translome of E15.5 cortex-derived neurons as compared to E12.5 ones (Fig. 3d), revealing that specific translational requirements are developmental stage-specific. We would also like to point out that we crossed MetRS* also with Emx1Cre/+; Ire1 α /f and using BONCAT now corroborate diminished ANL incorporation in E14.5 cortex (Fig. S10).

Reviewer response: I don't find the data presented 100% convincing. Radial glia cells have very clear radial alignment and this should be reflected by high-intensity Puromycin-stained progenitors at E14. The blow up in figure 3b shows processes projecting incoherently and could be background staining of endothelial cells of the vasculature. Better quality images where progenitors are labelled by SOX2/Pax6 and

Puromycin would better support these claims. Additionally, the line plots in figure 3a and 3c seem to be in contrast with the claims made, at E12.5 the profile reaches a grey value of 3000 in the VZ, while at E14.5 only 2500 and the prediction would be the opposite. In addition, it is very difficult to use such images for analyses as the stain appears to be uneven across the surface (e.g. 3A – left and right parts of the image) and the tiling gives obvious intensity artefacts at the edges of the individual tiles (3B). We ask the authors to provide better quality images to substantiate their claims. Additionally Fig.3c requires a loading control and relative quantification (with replicates).

We have fully agreed with the Reviewer and now provide images of improved quality for this experiment (Fig. 3a and 3b). Due to the cell density in the developing cortex and the nature of puromycin signal, we were unfortunately unable to dissect single cells and quantify the incorporation. For the experiment, we have used the Emx1-Cre:: Fucci26R reporter mouse to visualize Pax6-expressing (Fig. S2c and S2d) cycling Venus-labeled progenitors. Interestingly, we do observe a rather constant puromycin incorporation across the cortex at E12.5, in contrast to a gradient of puromycin incorporation at E14.5. Such pattern matches our quantification of HPG incorporation using electroporated progenitors on Fig. 2.

We now also provide the requested loading control and quantification of the culture experiment (Fig. 3c and 3d).